# DYSCA: A DYNAMIC AND SCALABLE BENCHMARK FOR EVALUATING PERCEPTION ABILITY OF LVLMS

**Jie Zhang**[12]**, Zhongqi Wang**[12]**, Mengqi Lei**[3]**, Zheng Yuan**[12]**,**
**Bei Yan**[12]**, Shiguang Shan**[12]**, Xilin Chen**[12]
[1] Key Laboratory of AI Safety of CAS, Institute of Computing Technology,
Chinese Academy of Sciences (CAS), Beijing, China
[2] University of Chinese Academy of Sciences, Beijing, China
[3] China University of Geosciences

## ABSTRACT

Currently many benchmarks have been proposed to evaluate the perception ability of the Large Vision-Language Models (LVLMs). However, most benchmarks conduct questions by selecting images from existing datasets, resulting in the potential data leakage. Besides, these benchmarks merely focus on evaluating LVLMs on the realistic style images and clean scenarios, leaving the multi-stylized images and noisy scenarios unexplored. In response to these challenges, we propose a dynamic and scalable benchmark named Dysca for evaluating LVLMs by leveraging synthesis images. Specifically, we leverage Stable Diffusion and design a rule-based method to dynamically generate novel images, questions and the corresponding answers. We consider 51 kinds of image styles and evaluate the perception capability in 20 subtasks. Moreover, we conduct evaluations under 4 scenarios (i.e., Clean, Corruption, Print Attacking and Adversarial Attacking) and 3 question types (i.e., Multi-choices, True-or-false and Free-form). Thanks to the generative paradigm, Dysca serves as a scalable benchmark for easily adding new subtasks and scenarios. A total of 24 advanced open-source LVLMs and 2 close-source LVLMs are evaluated on Dysca, revealing the drawbacks of current LVLMs. The benchmark is released at `https://github.com/Robin-WZQ/Dysca`.

## 1 INTRODUCTION

Recent years have witnessed the great success of the Large Vision-Language Models (LVLMs) (Li et al., 2023d; Zhu et al., 2023; Dai et al., 2023; Liu et al., 2023b; Li et al., 2023a; Chen et al., 2023b; Zhang et al., 2023; Su et al., 2023; Gong et al., 2023; Sun et al., 2023b). These models leverage the powerful Large Language Models (LLMs) (Chung et al., 2022; OpenAI, 2022; Touvron et al., 2023; OpenAI, 2023; FastChat, 2023) as their brain and incorporate the state-of-the-art visual encoders (Radford et al., 2021; Fang et al., 2023; Dosovitskiy et al., 2020) as their eyes. Thanks to the alignment of visual feature with textual space and the development of visual instruction tuning techniques (Liu et al., 2023b), LVLMs showcase the impressive capability in terms of visual scene comprehension and multimodal instruction-following.

In order to comprehensively evaluate the capabilities of LVLMs, many benchmarks have been purposed (Antol et al., 2015a; Singh et al., 2019; Xu et al., 2023; Shao et al., 2023; Li et al., 2023c;b; Fu et al., 2023; Bai et al., 2023b; Yu et al., 2023; Yang et al., 2023b; Chen et al., 2024), where we categorize the current benchmarks into three types (Fu et al., 2023). The first type is the classical benchmarks, such as COCO Caption (Chen et al., 2015) and VQA (Antol et al., 2015a; Goyal et al., 2017; Marino et al., 2019). Although these benchmarks provide high-quality evaluation data, they also have notable limitations. On the one hand, they are inadequate for measuring the fine-grained capabilities of current LVLMs, offering the limited insightful feedback for the future improvement. On the other hand, since these classical benchmarks have been available as the open-source test data for a long time, it is hard to prevent the data leakage problem. The second type of benchmarks evaluate the LVLMs through a subjective manner (Yang et al., 2023b; Wu et al., 2023). Although the benchmarks reveal the insightful drawbacks of current models, their data scale is limited (i.e., less than 200 annotations) and they require manual evaluation by experts. The third type is built

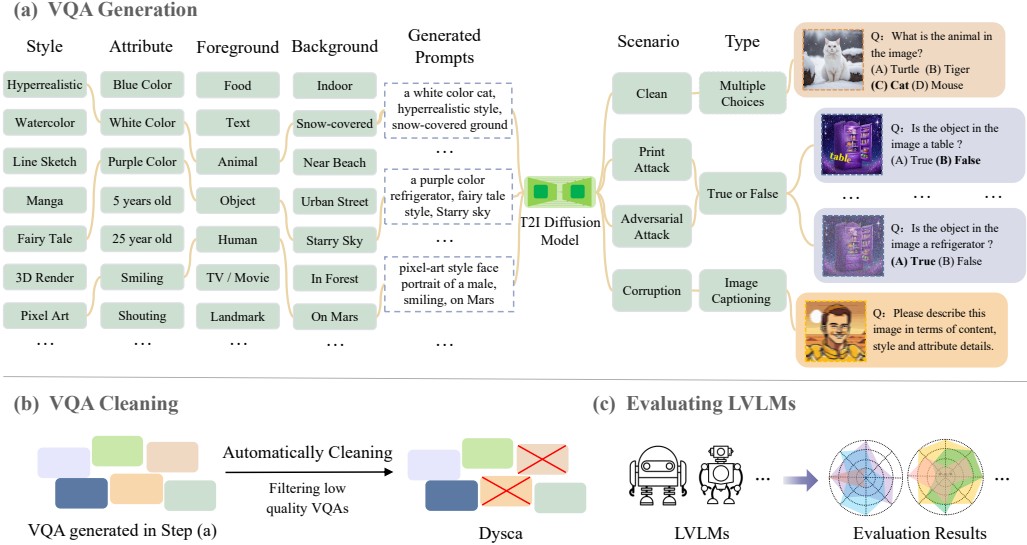

Figure 1: Overview of the automatic pipeline for generating Vision-language QAs, cleaning Vision-language QAs and evaluating LVLMs. (**a**) We first constructs prompts in terms of content, style and background, leveraging the Text-to-Image (T2I) diffusion model (e.g., SDXL (Podell et al., 2023)) to synthesis images to be asked. Then based on the scenarios and the question type, we post-process the synthesis images and generate the specific textual questions, respectively. (**b**) We further filter out low quality Vision-language QAs by utilizing trained models to form the final Dysca. (**c**) Finally, we evaluate LVLMs on our Dysca and feedback the fine-grained evaluation results.

for objectively evaluating current LVLMs and the comparison between them are shown in Tab. 1. They provide an objective and automatic evaluation manner, giving the fine-grained evaluation for the LVLMs. However, these benchmarks conduct Vision-language QAs by selecting images from existing dataset and annotate the textual questions. Although they claim that the questions are re-annotated, the previous work (Chen et al., 2024) has demonstrated that these benchmarks have Models unintentionally leaked into the training data of LLMs and LVLMs. Besides, most benchmarks focus on evaluating LVLMs in the realistic images and clean scenarios, leaving the multi-stylized images and noisy scenarios unexplored. While some works like MMCBench (Zhang et al., 2024b) and Typographic Dataset (Cheng et al., 2024) have investigated the robustness of LVLMs with corrupted and print-attacked images, respectively, they have not explored the effect of these noisy images on various perceptual tasks.

In this paper, aiming to address these challenges above, we propose Dysca which is a dynamic and scalable benchmark for evaluating the perception ability of LVLMs via various subtasks and scenarios. Inspired by the prior evaluation works for LLMs (Liang et al., 2023), we investigate on whether we could leverage the large-scale synthesized images for evaluating LVLMs. We display the overview of our pipeline in Fig. 1. Specifically, we leverage Stable Diffusion and design a rule-based method to dynamically generate novel images, questions and corresponding answers. We decouple the prompt into 4 part, i.e., attribute, foreground, style and background, and design pre-defined templates to dynamically generate prompts, as displayed in Fig. 3. Then we utilize the state-of-the-art text-to-image diffusion models (e.g., SDXL (Podell et al., 2023)) to generate the corresponding images. Since we already know the main information of the images through prompts, we easily generate question-answer textual pairs by the rule-based method. After that, in order to obtain the high quality Vision-language QAs, we employ CLIP (Radford et al., 2021) to perform data cleaning on the generated Vision-language QA pairs. Dysca focuses on assessing the fine-grained perception abilities, including recognizing human, animal, object, landmark, etc. Dysca evaluates LVLMs with **20 perceptual subtasks**, containing a total number of **51 different artistic styles**. Besides, to evaluate the robustness of the models across different scenarios and question types, we construct **4 testing scenarios** (clean, corruption, print attacking and adversarial attacking) and **3 question types** (multi-choices, true-or-false and free-form questions).

Compared to previous works in Tab. 1, we provide an end-to-end process from image to Vision-QA generation. This approach significantly reduces annotation costs compared to manually labeling

Table 1: Comparisons between existing LVLM benchmarks. '✓✗' indicates that the benchmarks include both newly collected images / annotations and images / annotations gathered from existing datasets. '*' The scale of our released benchmark is 617K, however Dysca is able to generate unlimited data to be tested.

| Benchmark | #Evaluation Data Scale | #Perceptual Tasks | Automatic Annotation | Novel Images & Novel Questions | Question Type | Automatic Evaluation |
|---|---|---|---|---|---|---|
| LLaVA-Bench | 0.15K | - | ✗ | ✓✗ | Free-form | ✓ |
| MME | 2.3K | 10 | ✗ | ✓✗ | True-or-false | ✓ |
| LVLM-eHub | - | 3 | ✓ | ✗ | Free-form | ✗ |
| tiny-LVLM-eHub | 2.1K | 3 | ✓ | ✗ | Free-form | ✓ |
| SEED-Bench | 19K | 8 | ✓✗ | ✗ | Multi-choices | ✓ |
| MMBench | 2.9K | 12 | ✗ | ✓✗ | Multi-choices | ✓ |
| TouchStone | 0.9K | 10 | ✗ | ✓ | Free-form | ✓ |
| REFORM-EVAL | 50K | 7 | ✓ | ✗ | Multi-choices | ✓ |
| MM-BigBench | 30K | 6 | ✓ | ✗ | Multi-choices | ✓ |
| MM-VET | 0.2K | 4 | ✓✗ | ✓✗ | Free-form | ✓ |
| MLLM-Bench | 0.42K | 7 | ✗ | ✓✗ | Free-form | ✓ |
| SEED-Bench2 | 24K | 10 | ✓✗ | ✗ | Multi-choices | ✓ |
| BenchLMM | 2.4K | 15 | ✗ | ✗ | Free-form | ✓ |
| JourneyDB | 5.4K | 2 | ✓ | ✓ | Free-form Multi-choices | ✓ |
| Dysca (Ours) | 617K* | 20 | ✓ | ✓ | Free-form Multi-choices True-or-false | ✓ |

images (e.g., MME (Fu et al., 2023)) while achieving the correctness for evaluating LVLMs. It also avoids the risk of hallucinate annotations that may occur when using ChatGPT for labeling based on image prompts (e.g., JourneyDB (Sun et al., 2023a)). This novel pipeline enables us to create a benchmark that is easily scalable and adaptable for incorporating new subtasks and scenarios. Thanks to the generative paradigm, Dysca can be customized to meet the specific requirements of the evaluator for testing purposes.

In summary, our work makes the following key contributions:

- **Dynamic and Scalable Benchmark:** We propose Dysca, a benchmark that is able to dynamically generate the test data that users need and is easily to scale up to to new subtasks and scenarios.

- **Multi-grained Perceptual Subtasks and Multi-scenarios:** Dysca aims to testing LVLMs' performance on diverse styles, 4 image scenarios (i.e., clean, corruption, print attacking and adversarial attacking) and 3 question types (i.e., multi-choices, true-or-false and free-form questions), reporting the 20 perceptual subtasks performance of 26 mainstream LVLMs, including GPT-4o (Ope) and Gemini-1.5-Pro (Team et al., 2024).

- **Analysis and Observations:** We demonstrate for the first time that evaluating LVLMs using large-scale synthetic data is valid. Experiments show the strong correlation coefficient between our evaluation rankings and the rankings obtained from non-synthetic benchmarks. The evaluation results also reveal the weakness of current LVLMs when facing different question types, image styles and image scenarios.

## 2 RELATED WORKS

### 2.1 LARGE VISION-LANGUAGE MODELS

The landscape of Large Vision-Language Models (LVLMs) has been significantly shaped by the pioneering success of Large Language Models (LLMs) such as GPTs (Radford et al., 2019; Brown et al., 2020; Ouyang et al., 2022) and LLaMA (Touvron et al., 2023), catalyzing advancements in multimodal content understanding and generation (Zhang et al., 2024a), including intricate tasks like image-text comprehension. At the forefront of these developments, BLIP-2 (Li et al., 2023d) introduces a lightweight Q-Former (Li et al., 2023d) that facilitates alignment between textual and

visual representations through a cross-attention mechanism (Li et al., 2023d). InstructBLIP (Dai et al., 2023) takes a step further by incorporating textual instructions into the Q-Former, which significantly improves the zero-shot performance. LLAVA (Liu et al., 2023b) employs GPT-4 (OpenAI, 2023) to transform data into multimodal instruction-following data and uses CLIP (Radford et al., 2021) and LLAMA (Touvron et al., 2023) for fine-tuning instructions, achieving advanced multimodal chat abilities. LLAVA-1.5 (Liu et al., 2023a) extends this paradigm by integrating MLP projection and introducing academic task-specific Vision-language QA data. Recently, models like Otter (Li et al., 2023a), MiniGPT-4 (Zhu et al., 2023), Qwen-VL-Chat (Bai et al., 2023a) and XComposer-VL (Zhang et al., 2023) further unleash the cross-modal understanding capabilities of LVLMs. Besides, many powerful closed-source LVLMs, including Gemini-1.5-Pro (Team et al., 2024) and GPT-4o (Ope), have publicly released their APIs, promoting the development of downstream applications.

## 2.2 BENCHMARKS FOR LVLMs

The great progress of LVLMs triggers the development of benchmarks for evaluating these models, where we divide previous benchmarks into three categories. The first type is the classical benchmarks which focuses on evaluating LVLMs abilities via image caption (Lin et al., 2014) and VQA (Antol et al., 2015b;a). However, these benchmarks cannot provide the fine-grained feedback on how to improve the models. Besides, since these benchmarks have been the public resources for a long time, it is hardly to guarantee that the LVLMs have not use them for training. The second type subjectively evaluates LVLMs by experts (Yang et al., 2023b; Wu et al., 2023). Although these benchmarks reveal the insightful feedback of the LVLMs, their scale is limited (i.e., less than 200 annotations). The subjective manner also makes the evaluation expensive and hardly to expand the scale.

The third type (Liu et al., 2023b; Fu et al., 2023; Xu et al., 2023; Shao et al., 2023; Li et al., 2023c;b; Liu et al., 2023c; Bai et al., 2023b; Li et al., 2023f; Yang et al., 2023a; Yu et al., 2023; Ge et al., 2023; Cai et al., 2023; Chen et al., 2024; Liu et al., 2024) focuses on evaluating LVLMs in an objective and large-scaled manner, where we list the detailed information of them in the Tab. 1. Some of them have been adopted by the community (Contributors, 2023) as the standard benchmarks for evaluating LVLMs (OpenAI, 2022; Zhang et al., 2023), like MME (Fu et al., 2023) and MMBench (Liu et al., 2023c). These benchmarks evaluate models through the objective answer types and most of them leverage the automatic annotation and evaluation manner for revealing the fine-grained drawbacks of current LVLMs. However, the previous benchmarks primarily concentrate on evaluating LVLMs using realistic images under clean scenario, leaving multi-stylized images and noisy scenarios unexplored. Moreover, many of them conduct QA by selecting images from publicly available datasets (e.g., (Lin et al., 2014; Russakovsky et al., 2014)). While they state that the questions have been re-annotated, they cannot guarantee that the LVLMs have not seen the image during training stage. The previous work (Chen et al., 2024) has proved that these benchmarks have unintentionally leaked into the training data of LLMs and LVLMs. One possible way to solve data leakage is using novel but synthesis images, where JourneyDB (Sun et al., 2023a) is the first work aiming to leverage synthesis images to evaluate current LVLMs. The prompts together with the corresponding images are downloaded from Midjourney (mid) and ChatGPT (OpenAI, 2022) is leveraged to label the images. However, JourneyDB is a top-down framework where the number of images is fixed. Besides, the ChatGPT labeling may cause hallucinate annotations, leading to the unreliable evaluation results. In contrast, Dysca serves as a bottom-up framework which enables dynamic and scalable generation of both images and evaluation questions, while also supporting varied evaluation tasks. The rule-based question generation method also makes the annotations more accuracy.

## 3 DYSCA

### 3.1 OVERVIEW OF OUR PIPELINE

The overview of our pipeline is shown in Fig. 1, containing data generation, data cleaning and LVLMs evaluation. For the data generation, our Dysca benchmark consists of four dimensions, i.e., $(M, P, I, Q)$, where $M$ means "Metadata", $P$ means "Prompt", $I$ means "Image" and $Q$ means "Question-answer pair". We further decouple the metadata $M$ into 4 parts, i.e., "style", "attribute", "foreground" and "background", and the combination of the four parts constitute the image prompts $P$. Then, given the prompt $P$ and the selected scenario, we leverage the Text-to-Image (T2I) diffusion model (e.g., SDXL (Podell et al., 2023)) to synthesis image $I$ and add the specific perturbation to the image $I$. After that, since the prompt already includes the question angle and the corresponding

Table 2: Key statistics of Dysca.

| Statistic | #Number |
|---|---|
| Total questions | 617K |
| - Clean | 156K (25.2%) |
| - Print attacking | 149K (24.1%) |
| - Adversarial attacking | 156K (25.2%) |
| - Corruption | 156K (25.2%) |
| Question type | |
| - Multi-choices | 251K (40.6%) |
| - True-or-false | 250K (40.5%) |
| - Free-form | 116K (18.8%) |
| Image resolution | 1024*1024 |
| Unique number of images | 289K |
| Unique number of questions | 162K |
| Unique number of answers | 31K |
| Average question length | 37.8 |
| Average answer length | 2.7 |
| Average choice number | 3.0 |

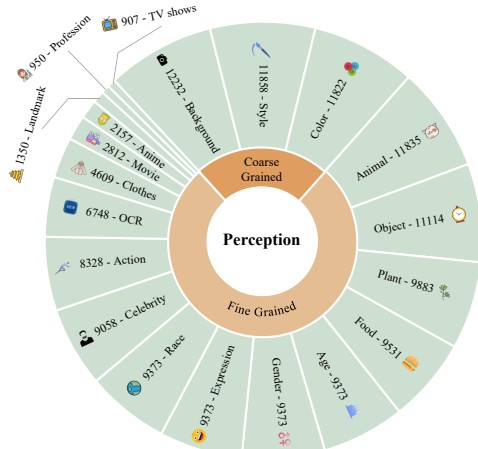

Figure 2: Overview of the dataset distribution of 20 perceptual tasks. The number in each subtask shows the corresponding amount of their annotation.

answer, we construct a rule-based approach to generate the $Q$. Three types of questions are considered, i.e., multi-choice, true-or-false and free-form. Multi-choice and true-or-false questions utilize a closed-ended manner to assess LVLMs, while free-form questions employ an open-ended manner through image captioning for evaluation. For the data cleaning, considering that the T2I diffusion model may generate unsuccessful outcomes, we then use CLIP (Radford et al., 2021) and PP-OCRv3 (Li et al., 2022) to automatically clean the whole dataset to obtain the final Dysca. Finally, we evaluate 14 open-sourced LVLMs and 2 closed-source LVLMs on our proposed Dysca.

## 3.2 PERCEPTUAL TASKS

**Evaluation dimensions.** Perception is one of the most fundamental capabilities of LVLMs and previous works (Fu et al., 2023) have shown that the lack of perceptual ability may result in hallucination (Li et al., 2023e). In order to comprehensively evaluate LVLMs' perception capability, we collect and organize existing sub-dimensions from current benchmarks, resulting in 20 assessment dimensions where we show all the subtasks and the corresponding amount of their annotation in the Fig. 2. We investigate on two types of perception dimensions, i.e., coarse-grained and fine-grained perception. Coarse-grained perception involves recognizing the style, background and color of images. Fine-grained perception involves recognizing the animal, object, plant, food, age, gender, expression, race, celebrity, action, text, clothes, movie, anime, landmark, profession and TV shows.

**Data sources.** For each perceptual subtask, we collect the textual data first to construct the metadata $M$. For the TV shows, Anime and Movie, we select the titles from the rating list of IMDb[1] based on the number of user reviews. For the styles, we utilize the style lists collected from the community[2] and remove those which have strong reflect on the image content like "architectural style" and "Pokemon style". Note that the style list does not include the style prompt associated with a particular artist's name. Besides, for the remaining contents, we select them from the label of current dataset (e.g., ImageNet (Russakovsky et al., 2014)). All the selected textual data above constitute the metadata $M$. We provide the detailed information of the metadata in the Appendix E.

## 3.3 CONSTRUCTION OF QUESTIONS & ANSWERS

Recall that the data generation for Dysca benchmark consists of four dimensions, i.e., $(M, P, I, Q)$, denoting the metadata $(M)$, prompt $(P)$, image $(I)$ and question-answer pairs $(Q)$, respectively. The relationships between these parts and the process of constructing Dysca are shown in Fig. 3. The metadata M is the core of the whole Dysca, containing all the information for generating $P$, $I$ and $Q$. The metadata $M$ consists of foreground, attribute, background and style, which guides the generation of the prompt $(P)$ through pre-designed templates. Then, we utilize the T2I diffusion model to generate the corresponding image using the prompt $P$. For generating the image with a specific text on it for the OCR subtask, we leverage TextDiffusion2 (Chen et al., 2023a), which is the state-of-

---

[1]https://www.imdb.com/

[2]https://stable-diffusion-art.com/sdxl-styles/

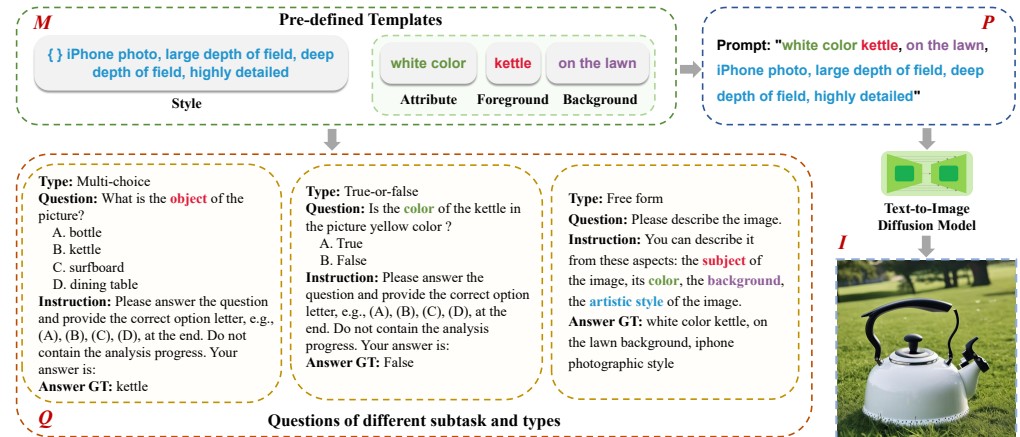

Figure 3: The process of generating the prompt (P), image (I) and QA pairs (Q) from metadata (M).

the-art text rendering method. For the rest of images, we leverage Stable Diffusion XL (Podell et al., 2023). Subsequently, based on the different question types we select, i.e., multi-choices, true-or-false and free-form, we generate the corresponding VQA pairs in Dysca.

Besides, in order to evaluate the model performance under various scenarios, we conduct experiments on 4 scenarios, i.e., clean, corruption, print attacking and adversarial attacking. For the print attacking, followed by (Cheng et al., 2024), we add the deceptive text on the image, where the text is a wrong option. Besides, to comprehensively evaluate the performance of LVLMs under corruption scenario, we add more typographic factors to original settings (i.e., different font orientations and font positions). For the adversarial attacking, we leverage PGD (Madry et al., 2017) to generate the adversarial image. We use InstructBLIP (Dai et al., 2023) as the proxy model and regard others as the black box models. The reason why we choose InstructBLIP is that it has shown superior performance in clean scenario. Besides, the black-box setting better reflects the robustness of the models when they face the real-world adversarial attacks. For the corruption, we leverage the image corruption methods collected from (Zhang et al., 2024b). We remove some hard corruptions as they significantly impact the quality of the image, leading to human failure in judging the style and content of the image. The detailed examples are shown in Appendix F.

**Data Clean.** To ensure the quality of Dysca, four steps are adopt: 1) First, we manually remove difficult-to-generate foregrounds and attributes, along with backgrounds and styles that could heavily affect image content. We believe this process can serve as a coarse-grained method to eliminate samples that are highly likely to be generated incorrectly. 2) Then, we leverage the off-the-shelf models, i.e., PP-OCRv3 (Li et al., 2022) and CLIP-L-14 (Radford et al., 2021), to clean the data. PP-OCRv3 (Li et al., 2022) is leveraged as the filter to exclude the failure image that TextDiffusion2 (Chen et al., 2023a) generates the wrong text on the image. For the other images, we use CLIP-L-14 (Radford et al., 2021) with a threshold of 0.75 to filter out the images with low text-image consistency. We find that using 0.75 as the threshold achieves a good balance between image correctness and data scale. 3) After that, We select the top six performing models and eliminate any question-answer pairs where the models either answer incorrectly or indicate that the answer was not included among the options. We observe that nearly 100% of the samples filtered out by these models are incorrect. 4) Finally, we analyze the patterns in these incorrect samples, removing the associated vocabulary from our metadata and discarding all related samples. By meticulously refining the metadata manually and utilizing automated tools to assist in question filtering, Dysca ensures high-quality data synthesis. In the end, we filter out nearly 40% of low quality samples. The final statistics of our released Dysca are shown in Tab. 2. Note that the OCR subtask does not involve print attacking scenario as misidentifying adversarial text does not indicate poor OCR robustness of the LVLMs. Therefore, there are 7K fewer questions in the print attacking scenario. Besides, for the free-form question type, since it allows to assess the model's perception abilities across multiple subtasks at the same time, we reduce the number of free-form questions for achieving a balanced data distribution.

## 3.4 EVALUATION STRATEGY

**Instruction Design.** We design two types of instructions to improve the instruction-following result of LVLMs. For the multi-choices and true-or-false questions, we design the questions followed by

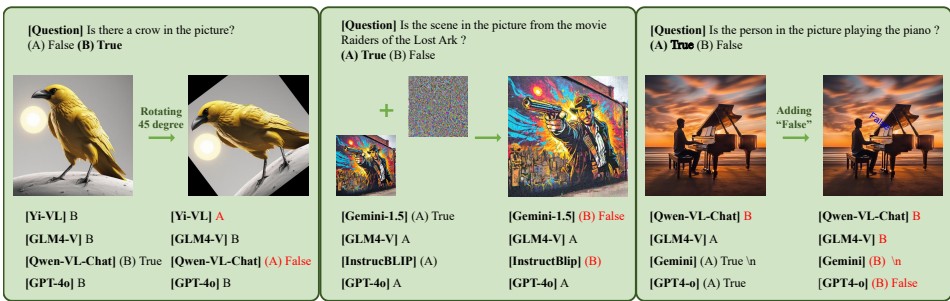

Figure 4: The failure cases for the noisy scenarios. From left to right are: corruption scenario, adversarial attacking scenario, and print attacking scenario.

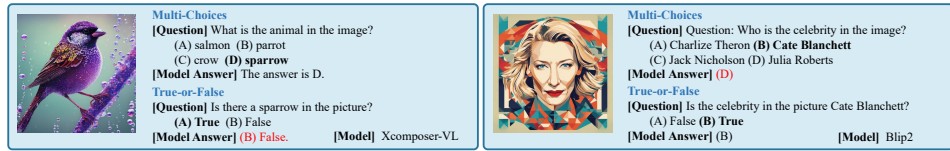

Figure 5: Models exhibit different performance when facing the same image but different question types.

the description "Please answer the question and provide the correct option letter, e.g., (A), (B), (C), (D), at the end. Do not contain the analysis progress. Your answer is: ". For the free-form questions, recalling that the prompt $P$ contains four part, i.e., the style, attribute, foreground and background, we instruct the model to caption these four dimensions by "Please describe the image. You can describe it from these aspects: {}", where "{}" includes the specific template we design for each part. We display the sample in the Fig. 3 and more examples can be found in the Appendix H.

**Evaluation Metrics.** For the multi-choices and true-or-false questions, we use accuracy as the evaluation metric. We randomly shuffle the order of choices to prevent evaluation results from being influenced by the model's tendency towards specific choices (Zong et al., 2023). The random accuracy of the two types are equal to 25% and 50%, respectively. We use regular expressions to extract the model's answer choices. For cases where the extraction is fail, we calculate the Levenshtein distance between the answer string and the choice string, and select the option with the minimum distance as the model's answer. For the free-form questions, we test the model's image caption capability where the ground truth is the prompt of the image. Followed by (Xu et al., 2023), we use SentenceTransformer (Thakur et al., 2021) to compute the text similarity with prompt $P$ and the caption output of the LVLMs. The final score of each question type is the average score of subtasks.

Besides, thanks to generative evaluation framework of Dysca, we are able to effectively control variables and conduct a detailed analysis of the model's fine-grained capabilities. Specifically, we introduce two novel metrics to measure the sensitivity of LVLMs on question types and covariate shift (i.e., image style). The sensitivity to question type aims to evaluate whether LVLMs exhibit inconsistent performance when facing different types of question types (i.e., multi-choice vs. true-or-false). We first normalize the score of true-or-false by $TFSQ = \frac{S-50}{100-50} * 100\%$ and multi-choices by $MCSQ = \frac{S-25}{100-25} * 100\%$, where $S$ denotes the score of the models in each question type. The sensitivity of LVLMs to question types is defined by:

$$SQ = \frac{(TFSQ - SQ_{Avg})^2 + (MCSQ - SQ_{Avg})^2}{2}, \quad (1)$$

where $SQ_{Avg} = \frac{TFSQ + MCSQ}{2}$.

The sensitivity to covariate shift aims to evaluate whether LVLMs exhibit inconsistent performance when facing the same content and question format, but with variations in image covariates. It is defined by:

$$SC = \frac{\sum_{i=1}^{N}(S_i - SC_{Avg})^2}{N}, \quad (2)$$

Table 3: Evaluation results of 26 LVLMs, where the darker colors represent better performance. The top 1 result on each column are **bolded** and the value in brackets is the relative values with respect to the ones in the clean scenario. "PrintAtt" and "AdverAtt" means "Print Attacking" and "Adversarial Attacking", respectively. "*": the model is under white-box setting.

| Model | LLM | Visual Encoder | Blind | Clean ↑ | Corruption ↑ | AdverAtt ↑ | PrintAtt ↑ | SQ ↓ | SC ↓ |
|---|---|---|---|---|---|---|---|---|---|
| MiniGPT-4 | Vicuna-7B | EVA-CLIP ViT-G | 35.37 | 41.38 | 42.30 (+0.92) | 34.42 (-6.96) | 42.71 (+1.33) | 193.6 | 1.7 |
| MiniGPT-4 | Vicuna-13B | EVA-CLIP ViT-G | 35.21 | 50.17 | 49.63 (-0.54) | 31.77 (-18.40) | 47.55 (-2.62) | 758.0 | 2.5 |
| MiniGPT-4 | LLaMA2-7B | EVA-CLIP ViT-G | 34.77 | 56.61 | 55.70 (-0.91) | 33.55 (-23.06) | 49.78 (-6.83) | 1344.5 | 3.0 |
| MiniGPT-2 | LLaMA2-7B | EVA-CLIP ViT-G | 35.28 | 58.46 | 58.06 (-0.40) | 56.62 (-1.84) | 52.96 (-5.50) | 1512.1 | 4.3 |
| BLIP2 | Flan-T5-XL | EVA-CLIP ViT-G | 35.35 | 65.30 | 66.09 (+0.79) | 32.55 (-32.75) | 57.01 (-8.29) | 165.5 | 8.5 |
| BLIP2 | OPT-3B | EVA-CLIP ViT-G | 34.99 | 39.54 | 40.29 (+0.75) | 30.62 (-8.92) | 37.26 (-2.28) | 110.7 | 2.3 |
| BLIP2 | OPT-7B | EVA-CLIP ViT-G | 35.21 | 39.55 | 41.12 (+1.57) | 31.76 (-7.79) | 38.82 (-0.73) | 50.2 | 1.8 |
| InstructBLIP | Vicuna-7B | EVA-CLIP ViT-G | 35.14 | 67.54 | 67.01 (-0.53) | 34.42 (-33.12) | 52.58 (-14.96) | 1287.1 | 2.8 |
| InstructBLIP | Vicuna-13B | EVA-CLIP ViT-G | 34.37 | 64.89 | 64.68 (-0.21) | 31.77 (-33.12) | 53.53 (-11.36) | 1252.4 | 2.5 |
| InstructBLIP | Flan-T5-XL | EVA-CLIP ViT-G | 34.51 | 66.54 | 67.58 (+1.04) | 32.95 (-33.59)* | 52.09 (-14.45) | 271.3 | 6.3 |
| InstructBLIP | Flan-T5-XXL | EVA-CLIP ViT-G | 34.82 | 68.65 | 69.79 (+1.14) | 32.95 (-35.70) | 57.73 (-10.92) | 215.0 | 6.8 |
| LLava-1.5 | Vicuna-7B | CLIP ViT-L | 34.63 | 51.27 | 51.70 (+0.43) | 49.62 (-1.65) | 47.27 (-4.00) | 788.1 | 3.2 |
| LLava-1.5 | Vicuna-13B | CLIP ViT-L | 35.21 | 59.23 | 59.58 (+0.35) | 56.87 (-2.36) | 51.69 (-7.54) | 912.5 | 7.7 |
| Otter | LLaMA-7B | CLIP ViT-L | 35.19 | 54.90 | 56.02 (+1.12) | 51.42 (-3.48) | 37.78 (-17.12) | 427.4 | 6.0 |
| Shikra | LLaMA-7B | CLIP ViT-L | 34.96 | 62.24 | 63.06 (+0.82) | 58.78 (-3.46) | 49.56 (-12.68) | 440.6 | 5.0 |
| Xcomposer-VL | InternLM-7B | EVA-CLIP ViT-G | 32.33 | 71.40 | 72.08 (+0.68) | 30.28 (-41.12) | 64.71 (-6.69) | 147.9 | 6.1 |
| Xcomposer2-VL | InternLM2-7B | CLIP ViT-L | 32.76 | 79.13 | 78.64 (-0.49) | 76.60 (-2.53) | 66.34 (-12.79) | **20.9** | 1.9 |
| Qwen-VL-Chat | Qwen-7B | OpenClip ViT-bigG | 33.06 | 62.18 | 61.05 (-1.13) | 59.85 (-2.33) | 51.94 (-10.24) | 1885.4 | 6.7 |
| Emu2-Chat | LLaMA-33B | EVA2-CLIP-E | 35.14 | 63.64 | 62.81 (-0.83) | 61.90 (-1.74) | 54.82 (-8.82) | 2497.9 | 1.6 |
| GLM-4V | GLM-4-9B-Chat | EVA2-CLIP-E | 35.08 | **82.09** | **81.95** (-0.14) | **80.72** (-1.37) | 52.09 (-30.00) | 25.5 | **0.8** |
| MiniCPM-V2.5 | Llama3-Instruct 8B | SigLIP SoViT-400m | 34.99 | 78.75 | 77.41 (-1.34) | 75.44 (-3.31) | 60.77 (-17.98) | 38.3 | 1.7 |
| Yi-VL | Yi-6B-Chat | OpenClip ViT-H | 35.01 | 75.71 | 74.94 (-0.77) | 72.53 (-3.18) | 64.97 (-10.74) | 233.1 | 1.9 |
| mPLUG-Owl-2 | LLaMA2-7B | CLIP ViT-L | 35.03 | 74.09 | 72.85 (-1.24) | 69.76 (-4.33) | **72.85** (-1.24) | 180.9 | 2.2 |
| Phi-3-Vision | Phi-3 | CLIP ViT-L | 34.74 | 73.23 | 72.11 (-1.12) | 69.66 (-3.57) | 57.78 (-15.45) | 292.7 | 1.5 |
| GPT-4o | / | / | 35.02 | 75.69 | 75.52 (-0.17) | 73.47 (-2.22) | 56.34 (-19.35) | 67.4 | 1.9 |
| Gemini-1.5-Pro | / | / | 34.55 | 77.79 | 77.12 (-0.67) | 75.89 (-1.90) | 61.05 (-16.74) | 439.6 | 1.6 |

where $SC_{Avg} = \frac{\sum_{i=1}^{N} S_i}{N}$ and $S_i$ denotes the score of the models in each style. Since there are 51 image styles in Dysca, we set $N = 51$.

## 4 RESULTS AND ANALYSIS

In this section, we report the evaluation results and make insightful analysis. A total of 26 LVLMs are evaluated on Dysca, including BLIP2 (Li et al., 2023d), InstructBLIP (Dai et al., 2023), LLaVA (Liu et al., 2023a), MiniGPT-4 (Zhu et al., 2023), Otter (Li et al., 2023a), XComposer-VL (Zhang et al., 2023), Qwen-VL-chat (Bai et al., 2023a), Shikra (Chen et al., 2023b), Emu2-Chat (Sun et al., 2024), GLM-4V (GLM et al., 2024), MiniCPM-v2.5 (Yao et al., 2024), Yi-VL (AI et al., 2024), mPLUG-Owl-2 (Ye et al., 2023), Phi-3-Vision (Abdin et al., 2024), GPT-4o (OpenAI, 2023), Gemini-1.5-pro (Team et al., 2024). Each model is evaluated with all the 20 perception subtasks under 4 scenarios. The detailed rankings for each subtask can be found in the Appendix A.

### 4.1 MAIN RESULTS

**Blind Setting.** We first evaluate LVLMs when only textual questions are provided. As shown in the "Blind" column of Tab. 3, all LVLMs yield consistent results on the Dysca and perform comparable to random guessing. This outcome demonstrates that the generated paradigm employed by Dysca effectively mitigates the potential impact of data leakage (Chen et al., 2024), thereby enhancing the fairness of the evaluation results. Additional comparisons can be found in Appendix D.

**Clean Scenario.** The evaluation results of various LVLMs in different perceptual subtasks under clean scenarios are presented in the "clean" column of Tab. 3. We calculate the average score of 3 question types. As can be seen, GLM-4V (GLM et al., 2024) outperforms other LVLMs, achieving top-1 performance. MiniCPM-v2.5 (Yao et al., 2024), Xcomposer2-VL (Dong et al., 2024) and Gemini-1.5-pro (Team et al., 2024) also perform well. It is evident that for the latest large models, their scores remain below 85. The results highlight that all existing LVLMs still struggle to provide accurate responses to questions formulated by Dysca.

**Noisy Scenarios.** The evaluation results of various LVLMs under noisy scenarios (i.e., corruption, print attacking and adversarial attacking) are presented in last 3 columns in Tab. 3. The value in the brackets shows the relative values with respect to the ones in the clean scenario. As can be seen, GLM-4V (GLM et al., 2024) still takes a lead on corruption and adversarial attacking scenarios. For the print attacking scenario, mPLUG-Owl-2 (Ye et al., 2023) performs the best. Here, we present a failure case sample for each of the three different scenarios on Fig. 4.

## 4.2 KEY OBSERVATIONS

**(1) For LVLMs, the capacity of the language model plays a crucial role.** When using the same visual encoder, models that utilize a language model with a larger parameter sizes (e.g., MiniGPT-4 achieves an improvement score of 8.79 in the clean scenario when using Vicuna-13B compared to Vicuna-7B) or with a stronger capability (e.g., GLM-4V that uses the GLM-4-9B-Chat language model shows an improvement score of 18.45 in the clean scenario compared to Emu2, which uses the LLaMA-33B language model) tend to achieve a better performance.

**(2) Each model shows robustness in the corruption scenario, but experiences significant degradation in both attack scenarios.** In the image corruption scenario, all models demonstrate minimal score variations, i.e., less than 1%. However, under print attacks, performance drops are notable. For instance, two closed-source models exhibit significant performance degradation: Gemini-1.5-pro declines by 21.5%, resulting in a score reduction from 77.79 to 61.05, while GPT-4o shows a 25.8% decrease, dropping its score from 75.69 to 56.10. Among the leading open-source models, GLM-4V experiences a sharp 36.5% performance drop, lowering its score from 82.09 to 52.09, and Phi-3-Vision records a 21.39% decline, reducing its score from 73.23 to 57.78. Notably, mPLUG-Owl-2 demonstrates the highest robustness, with only a 1.7% reduction. The XComposer-VL series also exhibits strong resilience against print-based attacks. In the adversarial attack scenario, where the attack algorithm directly targets the image encoder, LVLMs employing a shared encoder architecture (e.g., Blip2, InstructBLIP, and XComposer-VL, all of which utilize EVA-CLIP (Fang et al., 2022) as their image encoder) suffer substantial performance declines, with some models performing even worse than random chance. For example, XComposer-VL experiences a 57.6% drop, reducing its score from 71.40 to 30.28. Models with different image encoders also experience performance degradation ranging from 1% to 5%, showing a higher impact than that of corruption noise. More detailed results are available in Appendix F.

**(3) Models exhibit varying sensitivity to different question types and covariate shifts.** For the sensitivity to question types, we present two examples in Fig. 5. As can be seen, the XComposer-VL (Zhang et al., 2023) recognizes the sparrow in the image under a multiple-choice setting but fails to identify the sparrow in the same image under a true-or-false setting. The quantity results are shown in Tab. 3. Xcomposer2-VL achieves the best result with a score of 20.9. However, we observe that the perception ability of LVLM does not show a positive correlation with sensitivity to question types. For instance, while the GLM-4V achieves the highest performance in evaluation tasks, it exhibits higher sensitivity to question types than Xcomposer2-VL. One of the factors influencing the sensitivity to question types may be the inherent biases of the language model. Using the same base language model may result in similar outcomes for this metric. it is also noted that Gemini-1.5-pro performs bad in this metric, revealing its preference for certain question types. For the sensitivity to covariate shifts, as shown in the last column of Tab. 3, GLM-4V achieves the best result with a score of 0.8. However, we also observe that the perception ability of LVLM does not show a positive correlation with covariate shifts. For example, InstructBLIP-Flan-T5-XXL outperforms InstructBLIP-Flan-T5-XL in terms of performance but shows higher sensitivity to covariate shifts.

## 4.3 ANALYSIS ON INTER-TASK AND INTRA-TASK

**Analysis on Inter-task.** In order to investigate the inter-relationships across evaluation dimensions, we conduct hierarchical clustering based on the Euclidean distance of the scores across 20 dimensions for 26 models. We observe that models show varied consistency of performance across the dimensions. LVLMs tend to perform better in dimensions that involve well-defined image perception, such as landmark recognition and object recognition. However, dimensions likes style recognition and movie recognition exhibit greater challenges for LVLMs, which may be attributed to the limited training resources in these specific domains. Besides, commercial models exhibit poor performance in tasks related to "people" (i.e., "race", "age", and "gender"). This is likely due to the additional safety training incorporated into closed-source models, which results in more conservative responses on related questions. Detailed results are provided in Appendix A.

**Analysis on Intra-task.** Thanks to the diverse metadata design, Dysca enables highly granular analysis within a single task. Taking animal categories as an example, we analyze the performance of LVLMs on different animals. Here, we perform the same hierarchical clustering methods across 51 animal categories. LVLMs performance vary significantly across these categories. We find that models tend to perform poorly when applied to marine life. This could be caused by the challenges

in collecting ocean-related data. This observation highlights the need to direct the model's focus to oceanic domains.

## 4.4 THE VALIDITY OF DYSCA

In this section, we investigate on the evaluation gap between Dysca and non-synthesis benchmarks. We calculate the Spearman's rank correlation coefficient (Spearman, 1904) $\rho$ and the Kendall rank correlation coefficient (KENDALL, 1938) $\tau$ between the evaluation ranking of Dysca under clean scenario with the non-synthesis benchmark's evaluation

Table 4: The correlation results on three benchmarks, where $\rho \in [-1, 1]$ and $\tau \in [-1, 1]$.

| Style | Method | MMBench | OCRBench | SeedBench-2 |
|-------|--------|---------|----------|-------------|
| All | $\rho$ | 0.70 | 0.90 | 0.46 |
| | $\tau$ | 0.60 | 0.80 | 0.43 |
| Realistic | $\rho$ | 0.70 | 1.00 | 0.64 |
| | $\tau$ | 0.60 | 1.00 | 0.62 |

ranking, i.e., MMBench (Liu et al., 2023c), OCRBench (Liu et al., 2024) and SeedBench-2 (Li et al., 2023b). Both coefficient generate a score in the range of [-1,1], where 1 represents a perfect positive correlation, -1 represents a perfect negative correlation, and 0 represents no correlation. These coefficients are typical tools for measuring the correlation between variables in statistics. When the absolute value of either coefficient exceeds 0.6, it is considered to indicate a significant correlation (Akoglu, 2018). Specifically, we intersect our Dysca with current benchmarks based on the perceptual subtasks, evaluation models and evaluation question types. We then calculate the correlation of model evaluation rankings within this intersection. The results are shown in the first row of Tab. 4. For the MMbench and OCRBench, both metrics show the high correlation, with $\rho$ and $\tau$ higher than 0.6. However, the correlation for SeedBench-2 is not as strong. Considering that SeedBench-2 only contains realistic images, we conduct additional experiments using the evaluation ranks on our realistic style images only. As shown in the second row of Tab. 4, the correlation results of SeedBench-2 significantly improve (i.e., 0.46 vs. 0.64 for $\rho$ and 0.43 vs. 0.62 for $\tau$). The correlation with OCRBench also improves to 1, demonstrating the validity of using synthetic datasets for evaluation LVLMs.

Besides, we calculate the data distribution distance between each benchmark to prove the low distance distribution between Dysca and non-synthesis benchmarks. We select CCBench (Liu et al., 2023c), COCO-Val (Lin et al., 2014), MMVet (Yu et al., 2023), MMBench (Liu et al., 2023c), MME (Fu et al., 2023), MM-Star (Chen et al., 2024), OCRBench (Liu et al., 2024) and ScienceQA (Lu et al., 2022). The reason why we choose these benchmark is that they have been widely used in evaluating LVLMs. We use Kernel Maximum Mean Discrepancy (KMMD) (Schölkopf et al., 2007) to measure the distribution distance. Specifically, we randomly sample 3,000 images from each benchmark (if the scale of the benchmark less than 3000, we use all that data) and utilize CLIP (Radford et al., 2021) to encode these images. Then, we calculate the KMMD value using an RBF kernel between each pair. The results are shown in Fig. 6. Each row represents the value of KMMD between two benchmarks. The value of last row denotes the average value of KMMD. As can be seen, the distribution distance between Dysca and real-image benchmarks

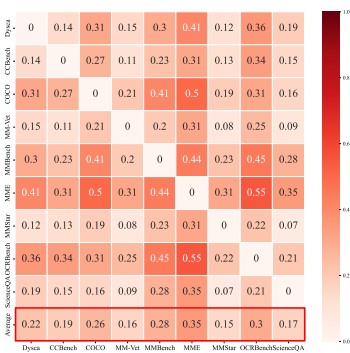

Figure 6: The KMMD distance between each benchmarks, with darker colors indicating larger distances.

ranks in the middle compared to all other benchmarks, indicating that the evaluation results can effectively reflect the model's performance in real-world scenarios.

## 5 CONCLUSION

In this paper, we purpose Dysca, a dynamic and scalable benchmark for evaluating perception ability of Large Vision Language Models (LVLMs). Dysca consists of 617K Vision-language QA pairs, covering 20 perceptual subtasks, 4 image scenarios and 3 question types. We conduct the experiment on 24 advanced open-source LVLMs and 2 closed-source LVLMs, revealing the insightful weakness of current LVLMs when facing different question types, image styles and image conditions. Experiments demonstrate the validity on evaluating LVLMs by using synthesis images.

ACKNOWLEDGEMENT

This work is partially supported by Strategic Priority Research Program of the Chinese Academy of Sciences (No. XDB0680202), Beijing Nova Program (20230484368), Suzhou Frontier Technology Research Project (No. SYG202325), and Youth Innovation Promotion Association CAS.

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
