# APPENDIX

## A  THE LEADERBOARDS

The model performance results for each subtask under clean scenario are shown in Tab. 5, Tab. 6, Tab. 7, and Tab. 8. Since the free-form question allows to assess the model's perception abilities across multiple subtasks at the same time, the results of free-form are not taken into account.

Table 5: Evaluation results on the 5 perceptual subtasks. The top two results on each subtask are **bolded** and underlined, respectively. "MC" and "TF" indicate the accuracy (%) of "Multi-choices" and "True-or-false", respectively.

| Model | LLM | Visual Encoder | Movie | | Action | | Tv Show | | Profession | | Landmark | |
|---|---|---|---|---|---|---|---|---|---|---|---|---|
| | | | MC | TF | MC | TF | MC | TF | MC | TF | MC | TF |
| MiniGPT-4 | Vicuna-7B | EVA-CLIP ViT-G | 29.53 | 51.03 | 45.06 | 50.67 | 30.43 | 47.09 | 35.85 | 46.88 | 52.37 | 51.15 |
| MiniGPT-4 | Vicuna-13B | EVA-CLIP ViT-G | 49.76 | 49.20 | 68.99 | 50.16 | 38.24 | 39.47 | 51.85 | 49.02 | 68.67 | 51.69 |
| MiniGPT-4 | LLaMA2-7B | EVA-CLIP ViT-G | 61.35 | 52.94 | 91.38 | 57.72 | 47.06 | 55.26 | 81.48 | 49.02 | 87.95 | 58.43 |
| MiniGPT-2 | LLaMA2-7B | EVA-CLIP ViT-G | 72.95 | 50.27 | 95.31 | 62.22 | 73.53 | 52.63 | 79.63 | 62.75 | 87.95 | 50.56 |
| BLIP2 | Flan-T5-XL | EVA-CLIP ViT-G | 72.45 | 67.72 | 97.16 | 93.02 | 57.97 | 57.00 | 81.60 | 75.45 | 98.11 | 93.44 |
| BLIP2 | OPT-3B | EVA-CLIP ViT-G | 31.88 | 41.18 | 34.49 | 48.23 | 32.35 | 47.37 | 25.93 | 54.90 | 32.53 | 43.82 |
| BLIP2 | OPT-7B | EVA-CLIP ViT-G | 19.90 | 42.78 | 24.05 | 47.91 | 23.53 | 47.37 | 18.52 | 50.98 | 24.39 | 39.33 |
| InstructBLIP | Vicuna-7B | EVA-CLIP ViT-G | 83.57 | 56.68 | 97.88 | 71.38 | 88.24 | 55.26 | 81.48 | 58.82 | 97.59 | 71.91 |
| InstructBLIP | Vicuna-13B | EVA-CLIP ViT-G | 83.57 | 59.36 | 98.79 | 64.63 | 88.24 | 63.16 | 77.78 | 60.78 | 97.59 | 66.29 |
| InstructBLIP | Flan-T5-XL | EVA-CLIP ViT-G | 77.93 | 67.24 | 97.60 | 93.98 | 68.12 | 56.04 | 82.08 | 75.00 | 97.16 | 95.08 |
| InstructBLIP | Flan-T5-XXL | EVA-CLIP ViT-G | 75.80 | 62.83 | 97.75 | 93.25 | 66.67 | 61.84 | 84.91 | 78.57 | 96.85 | 90.16 |
| LLava-1.5 | Vicuna-7B | CLIP ViT-L | 55.71 | 53.54 | 74.10 | 50.89 | 53.14 | 57.49 | 57.08 | 58.93 | 83.28 | 55.08 |
| LLava-1.5 | Vicuna-13B | CLIP ViT-L | 65.45 | 55.91 | 79.04 | 59.58 | 58.45 | 58.45 | 67.45 | 62.95 | 94.01 | 57.05 |
| Otter | LLaMA-7B | CLIP ViT-L | 66.01 | 59.53 | 66.62 | 72.44 | 70.05 | 57.00 | 68.87 | 55.36 | 59.31 | 66.56 |
| Shikra | LLaMA-7B | CLIP ViT-L | 68.34 | 61.26 | 78.44 | 78.01 | 60.87 | 57.97 | 82.08 | 68.30 | 88.96 | 70.82 |
| Xcomposer-VL | InternLM-7B | EVA-CLIP ViT-G | 80.82 | 77.64 | 97.01 | 94.80 | 78.26 | 70.53 | 84.43 | 76.34 | 97.16 | 95.41 |
| Xcomposer2-VL | InternLM2-7B | CLIP ViT-L | 91.79 | 88.77 | 98.18 | 95.34 | 85.29 | 84.21 | 88.89 | 90.20 | 98.80 | 100.00 |
| Qwen-VL-Chat | Qwen-7B | OpenClip ViT-bigG | 71.08 | 49.61 | 95.96 | 63.52 | 68.12 | 42.51 | 80.66 | 51.34 | 95.90 | 55.41 |
| Emu2-Chat | LLaMA-33B | EVA2-CLIP-E | 91.30 | 54.55 | 98.79 | 55.14 | 85.29 | 50.00 | 90.74 | 41.18 | 100.00 | 69.66 |
| GLM-4V | GLM-4-9B-Chat | EVA2-CLIP-E | 93.72 | 91.98 | 99.55 | 98.39 | 97.06 | 89.47 | 88.89 | 94.12 | 100.00 | 100.00 |
| MiniCPM-V2.5 | Llama3-Instruct 8B | SigLIP SoViT-400m | 93.24 | 89.84 | 98.18 | 92.77 | 94.12 | 89.47 | 90.74 | 88.24 | 100.00 | 98.88 |
| Yi-VL | Yi-6B-Chat | OpenClip ViT-H | 89.37 | 82.89 | 97.88 | 85.85 | 88.24 | 89.47 | 94.44 | 74.51 | 97.59 | 93.26 |
| mPLUG-Owl-2 | LLaMA2-7B | CLIP ViT-L | 90.34 | 76.47 | 96.82 | 85.37 | 82.35 | 81.58 | 87.04 | 82.35 | 98.80 | 86.52 |
| Phi-3-Vision | Phi-3 | CLIP ViT-L | 87.44 | 70.05 | 97.13 | 90.03 | 85.29 | 65.79 | 88.89 | 84.31 | 100.00 | 96.63 |
| GPT-4o | / | / | 86.47 | 74.87 | 96.96 | 94.81 | 79.41 | 86.84 | 83.33 | 80.39 | 96.39 | 98.88 |
| Gemini-1.5-Pro | / | / | 92.72 | 63.64 | 99.24 | 89.55 | 93.55 | 68.42 | 90.74 | 84.31 | 100.00 | 94.38 |

Table 6: Evaluation results on the 5 perceptual subtasks. The top two results on each subtask are **bolded** and underlined, respectively. "MC" and "TF" indicate the accuracy (%) of "Multi-choices" and "True-or-false", respectively.

| Model | LLM | Visual Encoder | Anime | | Clothes | | Celebrity | | Food | | Plant | |
|---|---|---|---|---|---|---|---|---|---|---|---|---|
| | | | MC | TF | MC | TF | MC | TF | MC | TF | MC | TF |
| MiniGPT-4 | Vicuna-7B | EVA-CLIP ViT-G | 27.44 | 47.27 | 30.61 | 49.49 | 27.26 | 47.87 | 43.96 | 50.00 | 47.07 | 49.94 |
| MiniGPT-4 | Vicuna-13B | EVA-CLIP ViT-G | 44.25 | 45.14 | 47.76 | 53.57 | 31.06 | 51.05 | 73.66 | 51.21 | 68.35 | 49.94 |
| MiniGPT-4 | LLaMA2-7B | EVA-CLIP ViT-G | 40.71 | 53.47 | 66.94 | 51.43 | 54.49 | 54.46 | 81.61 | 52.90 | 84.97 | 58.14 |
| MiniGPT-2 | LLaMA2-7B | EVA-CLIP ViT-G | 50.44 | 47.92 | 65.71 | 47.50 | 57.81 | 53.00 | 86.58 | 56.76 | 89.76 | 59.38 |
| BLIP2 | Flan-T5-XL | EVA-CLIP ViT-G | 56.91 | 61.77 | 84.08 | 75.26 | 80.78 | 76.45 | 93.24 | 90.56 | 92.53 | 90.15 |
| BLIP2 | OPT-3B | EVA-CLIP ViT-G | 26.55 | 45.14 | 28.57 | 45.00 | 31.89 | 48.70 | 32.42 | 48.31 | 35.11 | 48.82 |
| BLIP2 | OPT-7B | EVA-CLIP ViT-G | 24.78 | 46.53 | 29.39 | 45.00 | 23.29 | 51.86 | 26.96 | 49.15 | 26.73 | 50.25 |
| InstructBLIP | Vicuna-7B | EVA-CLIP ViT-G | 83.19 | 70.14 | 89.80 | 58.21 | 80.40 | 58.67 | 93.66 | 68.12 | 96.28 | 73.29 |
| InstructBLIP | Vicuna-13B | EVA-CLIP ViT-G | 76.11 | 62.50 | 93.06 | 60.71 | 77.91 | 54.13 | 95.03 | 65.46 | 95.61 | 72.30 |
| InstructBLIP | Flan-T5-XL | EVA-CLIP ViT-G | 60.77 | 63.65 | 88.27 | 82.72 | 83.11 | 75.75 | 93.96 | 91.36 | 92.98 | 91.36 |
| InstructBLIP | Flan-T5-XXL | EVA-CLIP ViT-G | 62.80 | 67.42 | 87.24 | 87.32 | 82.11 | 82.93 | 94.28 | 91.91 | 94.01 | 93.90 |
| LLava-1.5 | Vicuna-7B | CLIP ViT-L | 48.37 | 48.02 | 47.86 | 50.92 | 56.38 | 54.56 | 56.68 | 50.83 | 50.80 | 50.06 |
| LLava-1.5 | Vicuna-13B | CLIP ViT-L | 58.54 | 49.34 | 65.41 | 59.00 | 61.04 | 57.14 | 82.51 | 56.15 | 79.59 | 55.91 |
| Otter | LLaMA-7B | CLIP ViT-L | 62.80 | 54.99 | 47.24 | 71.37 | 44.15 | 63.76 | 44.44 | 79.46 | 69.29 | 80.56 |
| Shikra | LLaMA-7B | CLIP ViT-L | 47.15 | 58.19 | 76.63 | 59.71 | 63.90 | 62.65 | 84.26 | 67.72 | 88.73 | 66.39 |
| Xcomposer-VL | InternLM-7B | EVA-CLIP ViT-G | 74.80 | 74.01 | 86.84 | 87.22 | 88.23 | 87.25 | 93.72 | 90.96 | 92.34 | 91.55 |
| Xcomposer2-VL | InternLM2-7B | CLIP ViT-L | 88.50 | 84.03 | 95.10 | 92.50 | 84.72 | 88.82 | 96.89 | 94.08 | 97.74 | 93.42 |
| Qwen-VL-Chat | Qwen-7B | OpenClip ViT-bigG | 73.37 | 54.61 | 80.71 | 62.17 | 87.17 | 50.03 | 92.05 | 54.24 | 90.99 | 59.21 |
| Emu2-Chat | LLaMA-33B | EVA2-CLIP-E | 86.73 | 66.67 | 94.69 | 49.64 | 94.02 | 47.16 | 97.76 | 49.15 | 98.54 | 54.66 |
| GLM-4V | GLM-4-9B-Chat | EVA2-CLIP-E | 92.92 | 96.53 | 95.51 | 87.86 | 98.67 | 96.11 | 98.26 | 92.63 | 99.07 | 96.89 |
| MiniCPM-V2.5 | Llama3-Instruct 8B | SigLIP SoViT-400m | 82.30 | 89.58 | 91.84 | 92.50 | 88.04 | 80.71 | 96.02 | 93.48 | 98.14 | 95.03 |
| Yi-VL | Yi-6B-Chat | OpenClip ViT-H | 77.88 | 81.94 | 92.24 | 76.07 | 87.21 | 86.71 | 95.28 | 79.59 | 96.14 | 79.63 |
| mPLUG-Owl-2 | LLaMA2-7B | CLIP ViT-L | 83.19 | 70.14 | 89.80 | 81.79 | 86.05 | 72.61 | 93.79 | 84.18 | 94.95 | 84.47 |
| Phi-3-Vision | Phi-3 | CLIP ViT-L | 76.11 | 72.22 | 88.57 | 90.00 | 83.89 | 62.07 | 94.16 | 91.79 | 96.01 | 93.66 |
| GPT-4o | / | / | 72.57 | 88.11 | 89.34 | 88.89 | 70.72 | 55.92 | 98.26 | 92.75 | 97.47 | 94.90 |
| Gemini-1.5-Pro | / | / | 91.82 | 76.39 | 97.55 | 93.57 | 94.85 | 91.57 | 98.01 | 92.75 | 97.87 | 93.54 |

Table 7: Evaluation results on the 5 perceptual subtasks. The top two results on each subtask are **bolded** and underlined, respectively. "MC" and "TF" indicate the accuracy (%) of "Multi-choices" and "True-or-false", respectively.

| Model | LLM | Visual Encoder | Age | | Gender | | Expression | | Race | | Animal | |
|---|---|---|---|---|---|---|---|---|---|---|---|---|
| | | | MC | TF | MC | TF | MC | TF | MC | TF | MC | TF |
| MiniGPT-4 | Vicuna-7B | EVA-CLIP ViT-G | 30.02 | 48.28 | 52.00 | 46.75 | 42.29 | 49.82 | 25.10 | 48.42 | 45.54 | 49.39 |
| MiniGPT-4 | Vicuna-13B | EVA-CLIP ViT-G | 29.32 | 50.06 | 58.30 | 48.60 | 59.51 | 53.58 | 33.08 | 51.51 | 60.39 | 52.37 |
| MiniGPT-4 | LLaMA-7B | EVA-CLIP ViT-G | 49.48 | 49.74 | 84.34 | 56.14 | 57.26 | 57.74 | 28.35 | 52.96 | 89.84 | 54.49 |
| MiniGPT-2 | LLaMA2-7B | EVA-CLIP ViT-G | 41.62 | 52.84 | 86.13 | 61.60 | 65.69 | 63.85 | 51.34 | 53.56 | 92.53 | 60.20 |
| BLIP2 | Flan-T5-XL | EVA-CLIP ViT-G | 62.72 | 58.71 | 99.32 | 94.12 | 88.41 | 75.30 | 75.76 | 71.87 | 96.62 | 95.59 |
| BLIP2 | OPT-3B | EVA-CLIP ViT-G | 24.61 | 43.28 | 61.41 | 48.60 | 34.53 | 53.84 | 31.55 | 51.51 | 36.92 | 48.78 |
| BLIP2 | OPT-7B | EVA-CLIP ViT-G | 29.84 | 42.76 | 53.13 | 47.81 | 30.93 | 53.78 | 26.55 | 49.82 | 28.40 | 49.43 |
| InstructBLIP | Vicuna-7B | EVA-CLIP ViT-G | 70.16 | 57.36 | 99.66 | 74.00 | 83.13 | 67.23 | 84.16 | 67.55 | 97.46 | 69.17 |
| InstructBLIP | Vicuna-13B | EVA-CLIP ViT-G | 67.28 | 63.44 | 99.33 | 65.49 | 86.28 | 74.38 | 81.23 | 56.45 | 97.01 | 71.45 |
| InstructBLIP | Flan-T5-XL | EVA-CLIP ViT-G | 64.06 | 58.48 | 99.58 | 86.91 | 91.11 | 78.06 | 78.28 | 73.98 | 97.08 | 95.52 |
| InstructBLIP | Flan-T5-XXL | EVA-CLIP ViT-G | 67.84 | 81.16 | 99.58 | 98.50 | 82.56 | 80.11 | 82.46 | 83.55 | 97.32 | 95.71 |
| LLava-1.5 | Vicuna-7B | CLIP ViT-L | 37.74 | 55.40 | 53.62 | 49.89 | 63.49 | 51.34 | 42.55 | 51.34 | 49.45 | 52.18 |
| LLava-1.5 | Vicuna-13B | CLIP ViT-L | 49.29 | 59.78 | 98.60 | 83.59 | 70.12 | 58.87 | 70.81 | 63.45 | 86.12 | 58.74 |
| Otter | LLaMA-7B | CLIP ViT-L | 36.78 | 50.92 | 78.99 | 77.84 | 72.50 | 61.02 | 42.38 | 58.62 | 81.57 | 83.57 |
| Shikra | LLaMA-7B | CLIP ViT-L | 65.07 | 54.96 | 98.14 | 73.29 | 90.08 | 70.52 | 75.20 | 54.19 | 90.20 | 70.32 |
| Xcomposer-VL | InternLM-7B | EVA-CLIP ViT-G | 67.09 | 79.02 | 99.53 | 97.57 | 90.25 | 81.34 | 79.47 | 76.26 | 97.43 | 96.32 |
| Xcomposer2-VL | InternLM2-7B | CLIP ViT-L | 86.65 | 91.73 | 99.11 | 99.03 | 96.74 | **94.41** | 88.38 | 89.02 | 98.06 | 89.56 |
| Qwen-VL-Chat | Qwen-7B | OpenClip ViT-bigG | 53.19 | 48.71 | 97.50 | 59.18 | 85.55 | 64.04 | 74.43 | 55.95 | 95.45 | 63.89 |
| Emu2-Chat | LLaMA-33B | EVA2-CLIP-E | 84.95 | 49.35 | 99.89 | 50.91 | 92.58 | 53.06 | 91.95 | 51.39 | 98.21 | 53.51 |
| GLM-4V | GLM-4-9B-Chat | EVA2-CLIP-E | 86.13 | **93.80** | **100.00** | **100.00** | 98.09 | 94.28 | **94.51** | **94.69** | **99.40** | **99.02** |
| MiniCPM-V2.5 | Llama3-Instruct 8B | SigLIP SoViT-400m | 77.23 | 68.86 | 99.55 | 99.76 | 95.05 | 92.07 | 81.61 | 88.66 | 98.65 | 94.62 |
| Yi-VL | Yi-6B-Chat | OpenClip ViT-H | 71.86 | 86.69 | 98.88 | 98.66 | 91.11 | 88.30 | 84.55 | 74.31 | 97.31 | 83.85 |
| mPLUG-Owl-2 | LLaMA2-7B | CLIP ViT-L | 73.43 | 81.91 | 98.21 | 96.60 | 88.30 | 84.27 | 80.84 | 84.44 | 95.52 | 83.85 |
| Phi-3-Vision | Phi-3 | CLIP ViT-L | **86.65** | 70.03 | 98.99 | 99.51 | 96.40 | 81.66 | 91.19 | 89.51 | 96.86 | 88.42 |
| GPT-4o | / | / | 75.79 | 86.43 | 78.61 | 96.71 | 95.50 | 91.03 | 37.60 | 51.75 | 95.21 | 95.27 |
| Gemini-1.5-Pro | / | / | 82.07 | 77.26 | 99.55 | 98.91 | 96.40 | 87.65 | 31.42 | 63.57 | 97.91 | 89.72 |

Table 8: Evaluation results on the 5 perceptual subtasks. The top two results on each subtask are **bolded** and underlined, respectively. "MC" and "TF" indicate the accuracy (%) of "Multi-choices" and "True-or-false", respectively.

| Model | LLM | Visual Encoder | Object | | Text | | Style | | Background | | Color | |
|---|---|---|---|---|---|---|---|---|---|---|---|---|
| | | | MC | TF | MC | TF | MC | TF | MC | TF | MC | TF |
| MiniGPT-4 | Vicuna-7B | EVA-CLIP ViT-G | 52.54 | 50.66 | 29.68 | 50.25 | 35.68 | 18.89 | 31.20 | 48.98 | 35.55 | 48.86 |
| MiniGPT-4 | Vicuna-13B | EVA-CLIP ViT-G | 61.27 | 51.89 | 36.55 | 51.79 | 56.00 | 44.16 | 48.15 | 50.99 | 54.25 | 50.75 |
| MiniGPT-4 | LLaMA2-7B | EVA-CLIP ViT-G | 92.33 | 55.07 | 49.67 | 57.84 | 73.19 | 48.73 | 58.10 | 49.50 | 69.79 | 54.14 |
| MiniGPT-2 | LLaMA2-7B | EVA-CLIP ViT-G | 96.67 | 60.97 | 43.60 | 57.54 | 79.35 | 48.22 | 56.19 | 50.20 | 69.02 | 59.79 |
| BLIP2 | Flan-T5-XL | EVA-CLIP ViT-G | 90.26 | 89.86 | 74.33 | 62.32 | 83.87 | 35.45 | 76.23 | 72.02 | 88.58 | 86.64 |
| BLIP2 | OPT-3B | EVA-CLIP ViT-G | 37.83 | 44.63 | 26.68 | 53.08 | 27.90 | 44.67 | 29.21 | 45.03 | 32.34 | 46.33 |
| BLIP2 | OPT-7B | EVA-CLIP ViT-G | 26.83 | 48.33 | 26.25 | 51.98 | 29.35 | 47.21 | 25.21 | 46.42 | 23.00 | 47.46 |
| InstructBLIP | Vicuna-7B | EVA-CLIP ViT-G | 97.67 | 67.32 | 78.74 | 62.60 | 99.28 | 47.21 | 85.82 | 62.23 | 94.21 | 68.08 |
| InstructBLIP | Vicuna-13B | EVA-CLIP ViT-G | 98.17 | 71.10 | 81.34 | 57.04 | 99.82 | 48.73 | 90.16 | 56.46 | 96.33 | 59.13 |
| InstructBLIP | Flan-T5-XL | EVA-CLIP ViT-G | 90.70 | 91.10 | 75.14 | 60.76 | 83.94 | 33.05 | 77.08 | 73.41 | 90.26 | 86.45 |
| InstructBLIP | Flan-T5-XXL | EVA-CLIP ViT-G | 90.26 | 92.13 | 78.61 | 60.09 | 83.65 | 38.62 | 78.77 | 80.23 | 88.81 | 88.76 |
| LLava-1.5 | Vicuna-7B | CLIP ViT-L | 58.36 | 48.52 | 50.36 | 50.73 | 38.44 | 19.45 | 46.62 | 51.57 | 42.03 | 52.71 |
| LLava-1.5 | Vicuna-13B | CLIP ViT-L | 80.16 | 56.66 | 64.65 | 54.02 | 72.17 | 19.73 | 70.05 | 56.85 | 66.01 | 53.37 |
| Otter | LLaMA-7B | CLIP ViT-L | 48.62 | 83.92 | 67.20 | 62.46 | 82.27 | 23.54 | 71.49 | 67.59 | 49.89 | 57.76 |
| Shikra | LLaMA-7B | CLIP ViT-L | 74.35 | 70.26 | 64.99 | 53.78 | 79.65 | 22.97 | 79.58 | 64.28 | 83.06 | 60.57 |
| Xcomposer-VL | InternLM-7B | EVA-CLIP ViT-G | 90.19 | 91.24 | 72.46 | 79.01 | 79.65 | 29.18 | 81.35 | 81.01 | 86.60 | 89.37 |
| Xcomposer2-VL | InternLM2-7B | CLIP ViT-L | 98.83 | 97.13 | 89.26 | 88.79 | 100.00 | 94.42 | 92.06 | 92.94 | 94.11 | **95.95** |
| Qwen-VL-Chat | Qwen-7B | OpenClip ViT-bigG | 90.12 | 65.15 | 73.82 | 52.18 | 82.63 | 21.07 | 78.73 | 51.57 | 86.37 | 53.25 |
| Emu2-Chat | LLaMA-33B | EVA2-CLIP-E | 98.67 | 51.44 | 87.09 | 53.67 | 100.00 | 47.21 | 91.85 | 48.51 | 95.46 | 52.07 |
| GLM-4V | GLM-4-9B-Chat | EVA2-CLIP-E | **99.00** | **98.49** | **93.60** | **87.20** | 100.00 | 94.92 | **94.39** | **95.33** | 94.69 | 94.54 |
| MiniCPM-V2.5 | Llama3-Instruct 8B | SigLIP SoViT-400m | 98.50 | 96.97 | 88.39 | 84.13 | 100.00 | **98.98** | 90.69 | 88.87 | **96.33** | 95.10 |
| Yi-VL | Yi-6B-Chat | OpenClip ViT-H | 98.17 | 75.04 | 84.49 | 75.60 | 100.00 | 84.26 | 90.05 | 75.94 | 94.79 | 92.09 |
| mPLUG-Owl-2 | LLaMA2-7B | CLIP ViT-L | 96.33 | 88.80 | 81.45 | 65.77 | 99.28 | 78.17 | 85.82 | 74.65 | 91.80 | 88.23 |
| Phi-3-Vision | Phi-3 | CLIP ViT-L | 98.17 | 97.73 | 32.32 | 28.87 | 100.00 | 95.94 | 90.14 | 78.42 | 88.63 | 89.75 |
| GPT-4o | / | / | 95.17 | 94.25 | 82.97 | 82.59 | 100.00 | 95.94 | 91.19 | 89.46 | 83.11 | 89.36 |
| Gemini-1.5-Pro | / | / | 98.83 | 95.76 | 89.37 | 80.85 | 98.91 | 92.39 | 91.96 | 88.17 | 87.15 | 88.42 |

It can be observed that certain models exhibit inconsistency when faced with different forms of questions. We have noted this phenomenon across several state-of-the-art models (e.g., Emu2 (Sun et al., 2023b) and Qwen-VL-Chat (Bai et al., 2023a)).

We also provide a visualization of the scores for the top six models across 20 subtasks in different scenarios, as shown in Fig. 7. It can be observed that the same model exhibits performance variations across different dimensions. For instance, age perception is the most significant weakness for the Yi-VL (AI et al., 2024). Additionally, by comparing the radar charts of different scenarios, we can see that corruption scenario has the least impact on the models, while print attacking is almost catastrophic. This highlights the need for future work to focus on improving model robustness against print attacking.

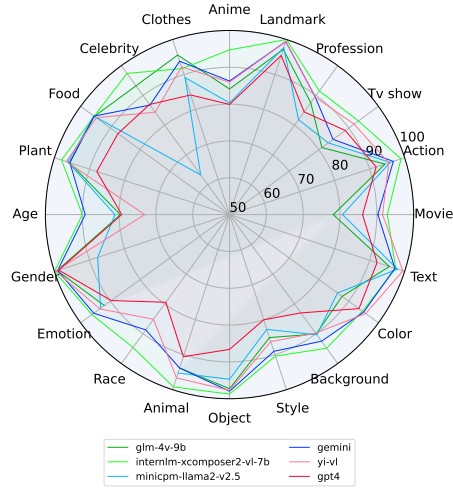

(a) Comparison of 6 top LVLMs on 20 subtasks under clean scenario. The full score of each subtask is 100.

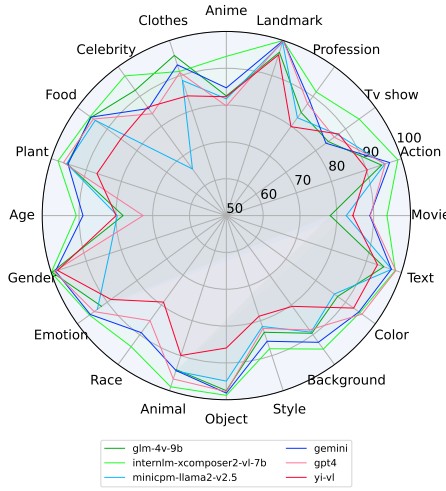

(b) Comparison of 6 top LVLMs on 20 subtasks under corruption scenario. The full score of each subtask is 100.

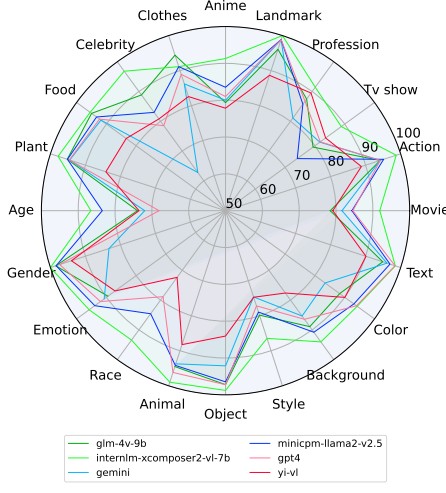

(c) Comparison of 6 top LVLMs on 20 subtasks under adversarial attacking scenario. The full score of each subtask is 100.

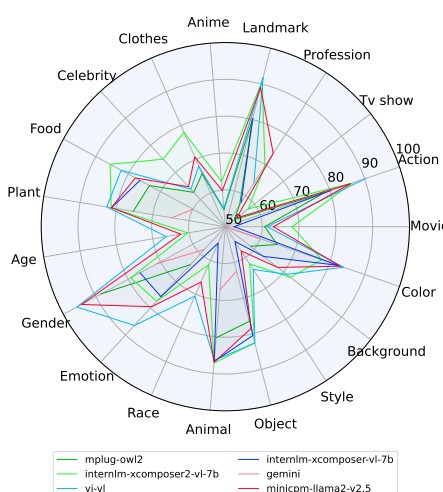

(d) Comparison of 6 top LVLMs on 19 subtasks under print attacking scenario. The full score of each subtask is 100.

Figure 7: Radar chart for the top 6 LVLMs in each scenario,

## B    DISCUSSION

### B.1    GENERAL DISCUSSION

**Limitation.** Dysca is the dynamic and scalable benchmark, offering evaluation for 20 perceptual subtasks under 51 image styles and 4 scenarios. However, generating data for evaluating cognition abilities (e.g., commonsense reasoning) presents challenge within the existing framework. This limitation arises from the reliance on predefined rules for prompt and question generation, which may not adequately capture the complexity of cognitive-level questions.

**Synthesis Data for Training / Fine-tuning.** The use of synthetic data for model training / fine-tuning has been adopted in the field of Natural Language Processing (NLP) (Meta, 2024). In this work, we do not explore the possibility of utilizing our benchmark for model training. Our primary goal in this paper is to provide a large-scale evaluation benchmark that addresses the issue of data leakage in current multimodal evaluation benchmarks and offers evaluation results across multiple subtasks, scenarios, question types and styles. Nevertheless, considering that Dysca has the capability to synthesize high-resolution and unlimited amounts of annotated multimodal data, we believe that Dysca also holds potential as a training data synthesis tool for LVLMs.

**Reproducibility and License.** All the experiments are built on 8 * RTX 4090. All the data and the code for generation and evaluation are released at `https://github.com/Robin-WZQ/Dysca`. The license of Dysca is "CreativeML Open RAIL++-M", which follows the license set by the Stable Diffusion XL.

**Ethical Concerns.** Our Dysca leverages the Stable Diffusion XL (Podell et al., 2023) to generate images. In order to prevent the model generating unsafe images, e.g., NSFW and offensive images, lots of efforts have been made. First, we use the safety checker (Rando et al., 2022) to post filter the unsafe images. The safety checker is a post-processor deployed by model developers (Podell et al., 2023) to prevent the generation of NSFW images. With the unsafe image that is recognized by the safety checker, the model's output will be a blank image. Besides, we manually exclude the specific styles or the word that may trigger the unsafe images generation from the Metadata $M$. To validate the safety of Dysca, we utilize the NudeNet detector (Bedapudi, 2022) with with a threshold of 0.8 to identify images that contain NSFW content. NudeNet is the widely used automatic tool for detecting NSFW images which is adopted by (Gandikota et al., 2023; Lu et al., 2024b). With 77847 images (clean scenario) in our Dysca, only 5 images are classified as NSFW (0.006%). We check the 5 images and find they are all false positive samples. This indicates that our Dysca contains a minimal amount of such unsafety images.

### B.2    THE STABILITY OF DYSCA

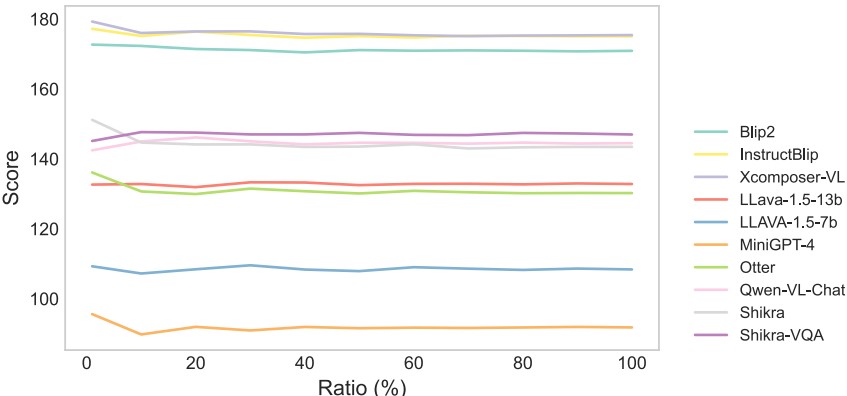

Figure 8: The tendency of 10 model's overall performance under clean scenario with different scale of evaluation data.

In this section, we focus on examining the stability of Dysca. We partition Dysca into 11 different scales: 1%, 10%, 20%, 30%, 40%, 50%, 60%, 70%, 80%, 90% and 100%. We randomly select 10

models and compute their evaluation scores using each of these data scales. The score is calculated as the sum of scores obtained from multiple-choice, true-or-false and free-form questions. As can be seen in Fig. 8, when the evaluation data scale is less than 30% of Dysca (i.e., less than 46.8K samples), the evaluation score show significant fluctuations. When the data scale exceeds 40%, we obtain the stable results, reflecting current scale of Dysca achieves the stable and reliable evaluation results. Although 40% evaluation scale of Dysca has achieved stable scores, Dysca aims to provide more than just stable rankings, but also draws on massive amounts of data to provide in-depth feedback across different image styles and perceptual subtasks.

## C   THE CORRECTNESS OF DYSCA

Since our generative pipeline relies on text-to-image models, the quality of the Dysca dataset is inherently influenced by the generation performance of the model. To address this issue, we have employed extensive data cleaning techniques to ensure the correctness of the dataset. Furthermore, to validate this, we randomly sample 7.7K images (10% of our data) and manually select images that contain wrong content or potentially lead to incorrect responses. In the end, a total of 167 images (2.2%) are filtered. Compared to previous but wide spread benchmarks, the incorrect data ratio of Dysca is less (Northcutt et al., 2021). For example, the ImageNet dataset (Deng et al., 2009) has at least 2,916 errors in its validation set, with an error rate of 6%; the QuickDraw dataset (Ha & Eck, 2018) has at least 5 million errors, with an error rate of approximately 10%. Subsequently, we compute the scores with and without these incorrect images, and the results are presented below.

Table 9: The scores with and without these incorrect images.

| Model | Before | After |
|---|---|---|
| minigpt4-vicuna-7b | 41.38 | 41.24 |
| minigpt4-vicuna-13b | 50.17 | 50.78 |
| minigpt4-llama2 | 56.61 | 56.82 |
| minigpt-v2 | 58.46 | 58.43 |
| blip2-flan-t5-xl | 65.30 | 65.32 |
| blip2-opt-3b | 39.54 | 39.80 |
| blip2-opt-7b | 39.55 | 39.58 |
| instructblip-vicuna-7b | 67.54 | 67.89 |
| instructblip-vicuna-13b | 64.89 | 65.02 |
| instructblip-flan-t5-xl | 66.54 | 66.73 |
| instructblip-flan-t5-xxl | 68.65 | 68.74 |
| llava-1.5-7b | 51.27 | 51.44 |
| llava-1.5-13b | 59.23 | 59.46 |
| otter | 54.90 | 54.75 |
| shikra-7b | 62.24 | 62.35 |
| internlm-xcomposer-vl-7b | 71.40 | 71.55 |
| internlm-xcomposer2-vl-7b | 79.13 | 79.23 |
| qwen-vl-chat | 62.18 | 62.26 |
| emu2-chat | 63.64 | 63.67 |
| glm-4v-9b | 82.09 | 82.13 |
| minicpm-llama2-v2.5 | 78.75 | 79.02 |
| yi-vl | 75.71 | 76.32 |
| mplug-owl2 | 74.09 | 74.43 |
| phi-3-vision | 73.23 | 73.57 |
| GPT-4o | 75.69 | 76.21 |
| Gemini-1.5-Pro-Vision-latest | 77.79 | 77.99 |

As observed, the score difference is negligible, making the automatic evaluation result reliable. We believe that with the development of deep generative models, Dysca will serve as an evolving benchmark with improving quality.

# D    BLIND EXPERIMENT

To demonstrate that Dysca has minimal data leakage issues, we conduct the "Blind" experiment where only textual questions are provided to LVLMs. We present the evaluation results of LLaVa-1.5-7B (Liu et al., 2023b) on Dysca and the six benchmarks, including MMMU (Yue et al., 2024), MMB (Liu et al., 2023c), ScienceQA (Lu et al., 2022), AI2D (Kembhavi et al., 2016), Seed (Li et al., 2023c) and MathVista (Lu et al., 2024a).

Table 10: Scores of LLaVa under the blind setting across 7 benchmarks.

| Model | MMMU | MMB | ScienceQA | AI2D | SEED | MathVista | Dysca |
|---|---|---|---|---|---|---|---|
| Random Choice | 22.1 | 0 | 24.2 | 23.8 | 24.3 | 17.9 | 37.5 |
| LLaVA-1.5-7B | 29.9 | 19.5 | 64.1 | 48.7 | 37.5 | 20.3 | 38.7 |

As shown in Tab. 10, LLAVA significantly outperforms random selection when only text questions are provided in the other 6 benchmarks. In contrast, Dysca achieves results closest to random selection, indicating that our work has a limited data leakage issue.

# E    THE METADATA ($M$) OF DYSCA

Metadata ($M$) is the core of Dysca, which is randomly assembled from our collected source material and contains all the information needed to generate prompt ($P$), image ($I$), and question-answer pairs($Q$). Specifically, the metadata is a data container that contains information in multiple dimensions about the foreground, the attributes corresponding to the foreground, the background, and the artistic style required to generate an image. Therefore, each instance of M is mapped one-to-one to a prompt, an image, and a set of question-answer pairs, respectively.

In order to ensure the quality and stability of the generated images, we carefully select the source material. First, for each perceptual subtask, we collect rich annotation material as described in Section 3.2. However, the metadata composed of these raw annotations is not always usable. On the one hand, some of the content is polysemous, which can easily be misinterpreted by the model's when generating images. On the other hand, there are backgrounds or artistic styles (e.g., "Pokemon Style", "architectural style", etc.) that negatively affect the quality of the image and do not accurately generate the desired content. In order to test the usability of these source materials, we went through several small-scale pre-generations covering all the source materials. After careful selection, we retain the clips that consistently produced high-quality images. The detailed information of the source materials are shown in Tab. 11.

# F    SCENARIOS DETAILS

## F.1    PRINT ATTACK SCENARIO

Followed by the settings in (Cheng et al., 2024), we add the attack text on the images. Consider that the image resolution in our Dysca is much higher than the one in (Cheng et al., 2024), we increase more font form in terms of font position and font orientation. Fig. 9 to Fig. 13 shows the detailed examples.

## F.2    CORRUPTION SCENARIO

Examples of the 11 image corruptions are shown in Fig 14.

Table 11: Detailed information of the source materials.

| Category | Data Description | #Numbers |
|---|---|---|
| Style | We collected artistic styles from the community, which can be well rendered by Stable Diffusion model. We removed those which have strong reflect on the image content or may generate unsafe image. | 51 |
| Background | We have selected 20 rich backgrounds and they can be accurately generated by Stable Diffusion Model. | 20 |
| Age | We chose four well-characterized age nodes: 14, 25, 40, and 80. | 4 |
| Expression | We chose three characteristic expressions: smiling happily, shouting furiously, calm and placid. | 3 |
| Gender | Male and Female. | 2 |
| Race | We identified these five races based on the ethnicity that can be generated by Stable Diffusion: Caucasian, Asian, African, Indian, and Middle Eastern races | 5 |
| Profession | After pre-generation and careful selection, we chose 20 occupations with distinctive characteristics that are easy to generate. | 20 |
| Action | After pre-generation and careful selection, we chose 20 occupations with distinctive characteristics that are easy to generate. | 20 |
| Celebrity | After pre-generation and careful selection, we chose 50 well-known celebrities. | 50 |
| Animal | We selected a rich variety of animals, including mammals, birds, reptiles, insects, and aquatic animals, and can they be generated by the Stable Diffusion model accurately. | 67 |
| Plant | We selected a rich variety of plants, including flowers, trees, fruits, and vegetables, and they can be generated by the Stable Diffusion model accurately. | 37 |
| Clothes | We selected 16 common types of clothing that are highly distinguishable from each other. | 16 |
| Object | We took the annotations from the MSCOCO (Lin et al., 2014) dataset after removing people, animals, and plants, and added some additional common objects. | 80 |
| Landmark | We chose 23 characteristic landmarks from around the globe and they can be generated by the Stable Diffusion model accurately. | 23 |
| Food | We collected 29 special dishes from around the globe and they can be generated by the Stable Diffusion model accurately. | 29 |
| Movie | We selected 106 movies titles from the rating list of IMDb based on the number of user reviews. | 106 |
| Anime | We selected 44 anime titles from the rating list of IMDb based on the number of user reviews. | 44 |
| TV shows | We selected 20 TV show titles from the rating list of IMDb based on the number of user reviews. | 20 |
| OCR | We randomly selected 5000 words from the IELTS vocabulary for the text material. Among them, words with length less than 3 were removed. | 5000 |
| Color | We selected 8 easily distinguishable colors: red, orange, yellow, green, blue, purple, white, black. | 8 |

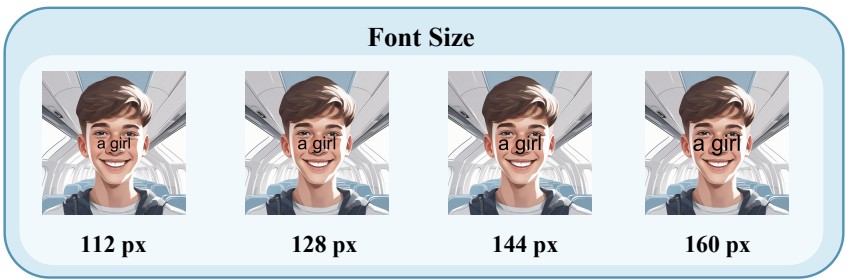

Figure 9: Images with different font size. '112px' means that the typos are 112 pixels in size.

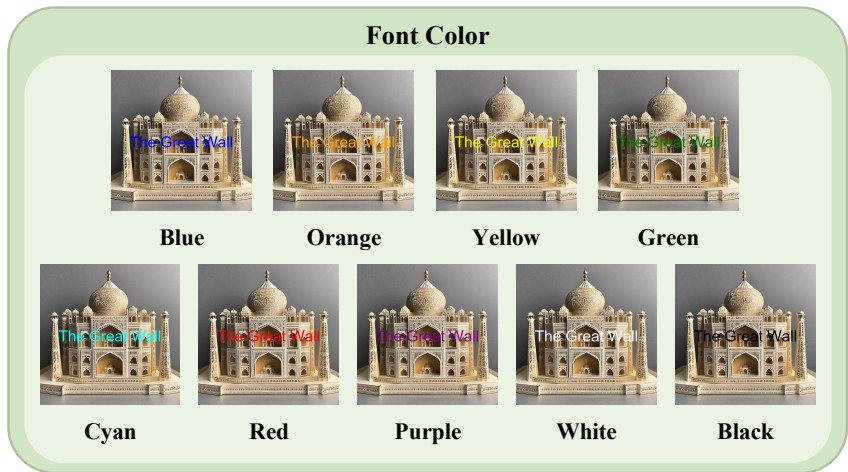

Figure 10: Images with different font color. 'Blue' means that the color of typos are in blue.

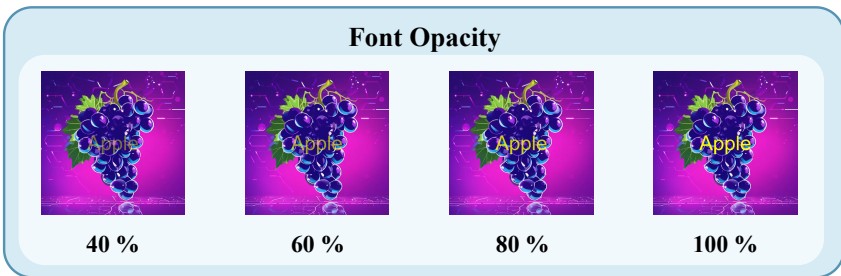

Figure 11: Images with different font opacity. '40%' means that the transparency of typos are 40% and '100%' implies full opacity of typos.

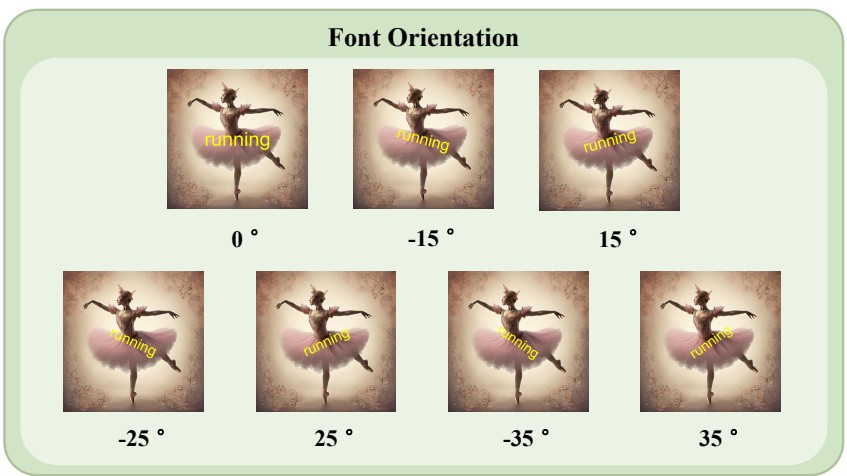

Figure 12: Images with different font orientation. '15°' means that the orientation of typos are $15°$ and $0°$ implies the typos are horizontal.

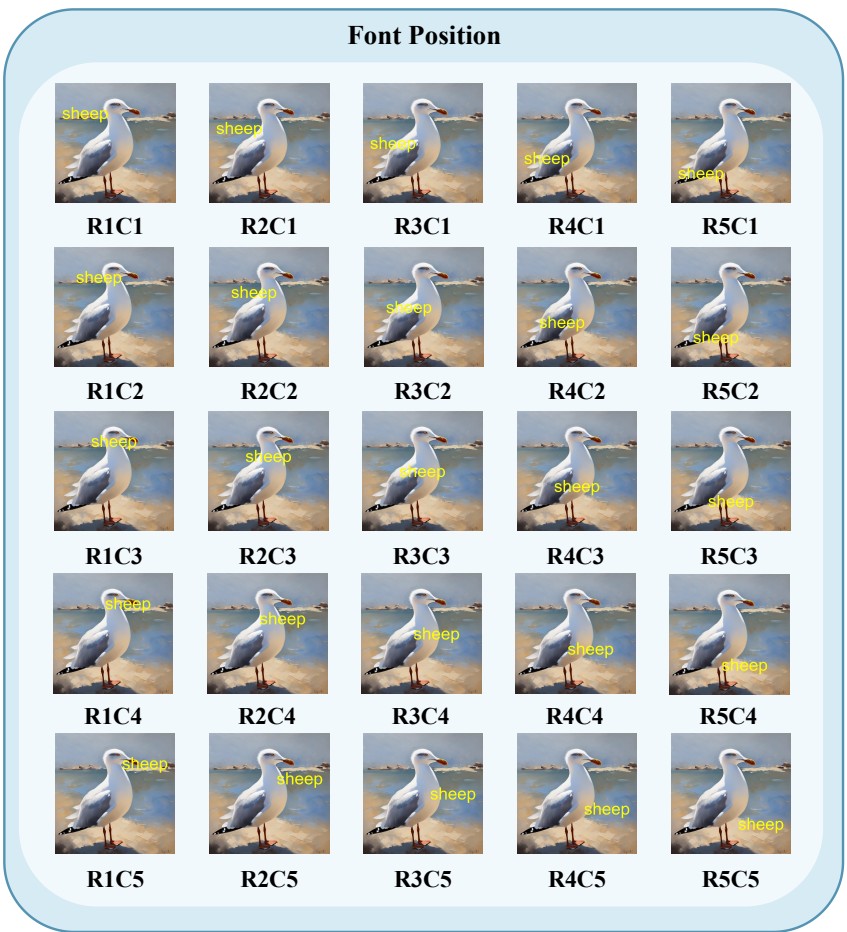

Figure 13: Images with different font position. An image is divided into a grid of 5 rows and 5 columns, leading to 25 sections. 'R1C1' means the typo is located in row 1, column 1, which is the top left corner of the image.

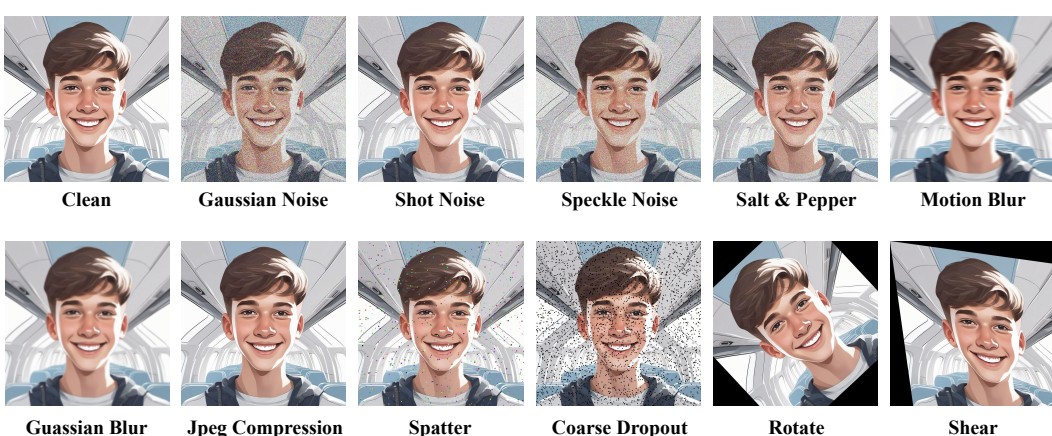

Figure 14: Examples of 11 image corruptions applied to a single clean image.

# G  HARD SAMPLES

In this section, we provide hard samples that LVLMs are likely to answer incorrectly, as shown in Fig. 15 to Fig. 17.

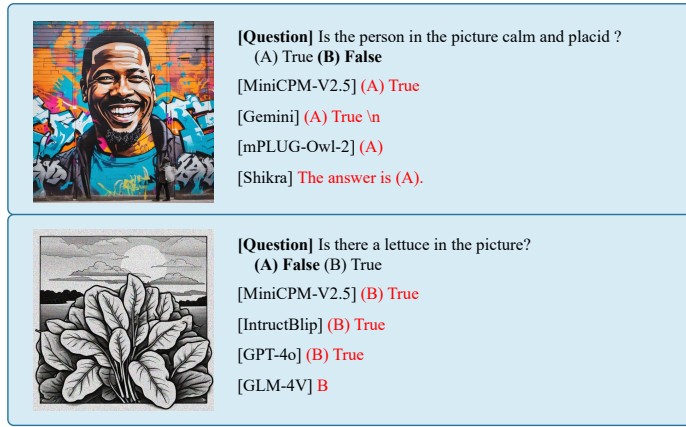

Figure 15: Hard samples.

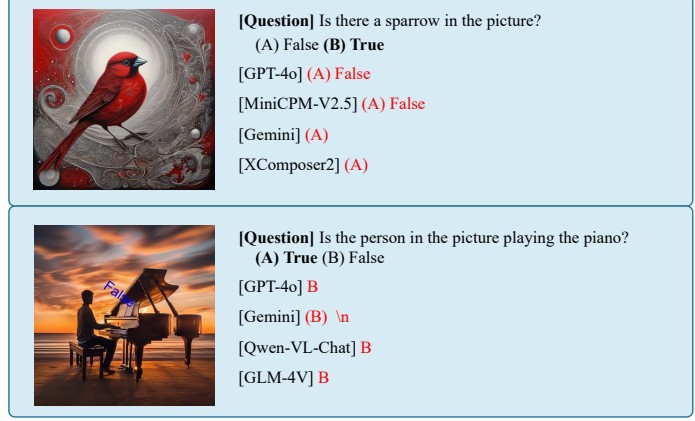

Figure 16: Hard samples.

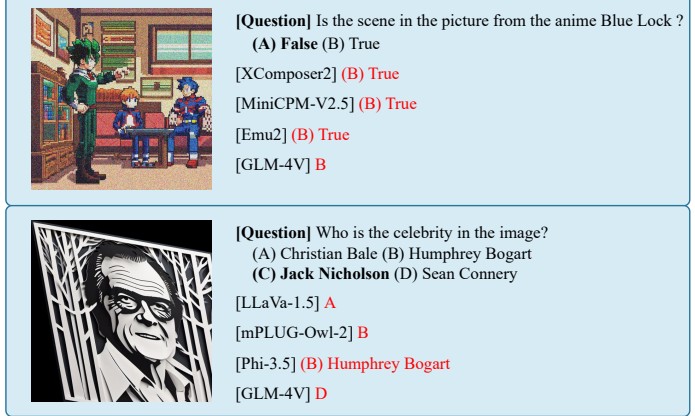

Figure 17: Hard samples.

# H    MORE EXAMPLES OF DYSCA

For each subject we collected in Metadata ($M$), we display one example of their prompt ($P$), generated image ($I$) and corresponding question-answer pairs ($Q$).

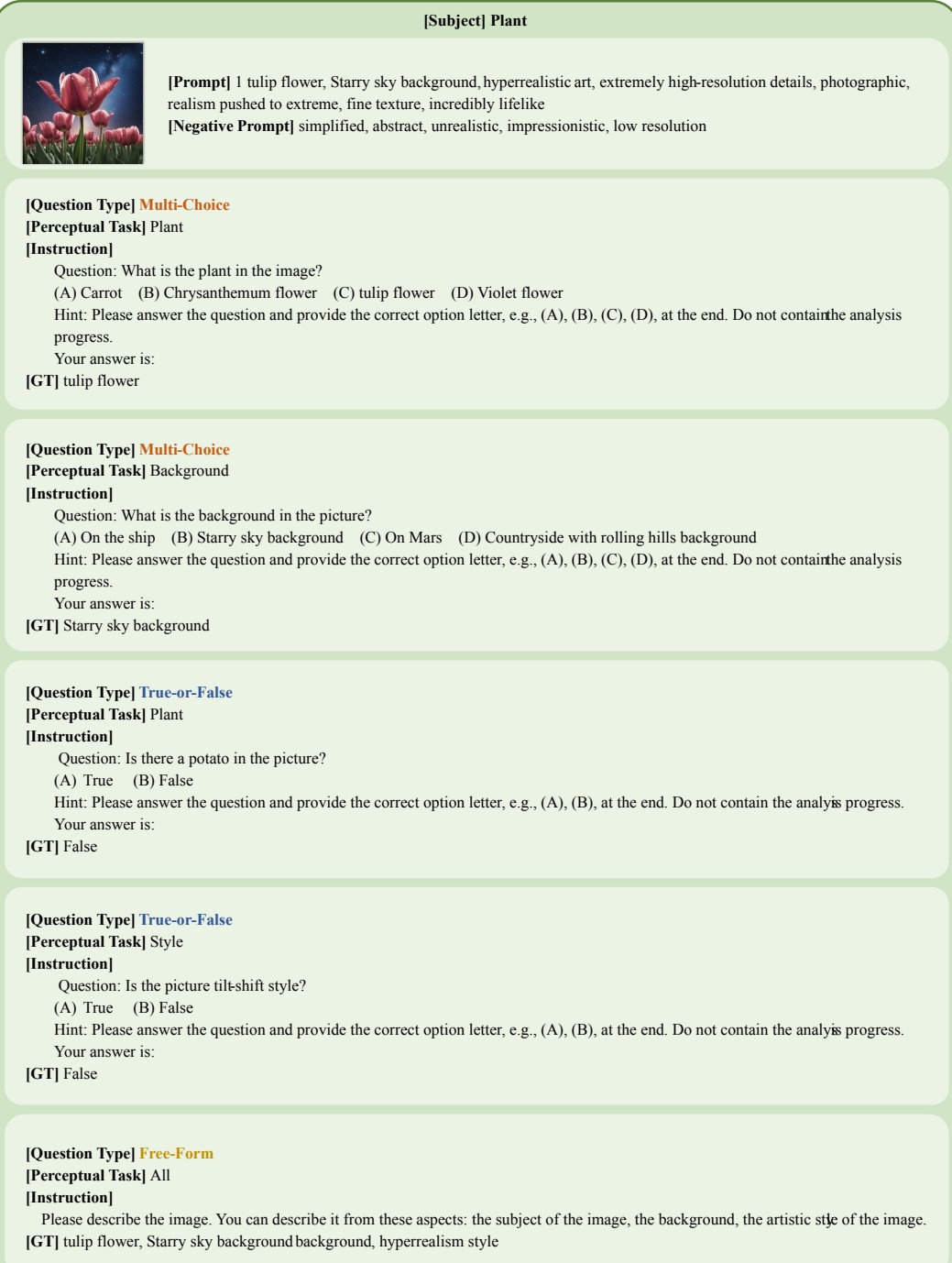

Figure 18: Plant

**[Subject] Profession**

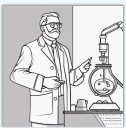

**[Prompt]** 1 Scientist, line art drawing style, professional, sleek, modern, minimalist, graphic, line art, vector graphics
**[Negative Prompt]** anime, photorealistic, 35mm film, deformed, glitch, blurry, noisy, off-center, deformed, cross-eyed, closed eyes, bad anatomy, ugly, disfigured, mutated, realism, realistic, impressionism, expressionism, oil, acrylic

**[Question Type] Multi-Choice**
**[Perceptual Task]** Profession
**[Instruction]**
Question: What is the occupation of the person in the picture?
(A) Chef   (B) Scientist   (C) Police officer   (D) Painter
Hint: Please answer the question and provide the correct option letter, e.g., (A), (B), (C), (D), at the end. Do not contain the analysis progress.
Your answer is:
**[GT]** Scientist

**[Question Type] True-or-False**
**[Perceptual Task]** Style
**[Instruction]**
Question: Is the picture line art style?
(A) True   (B) False
Hint: Please answer the question and provide the correct option letter, e.g., (A), (B), at the end. Do not contain the analysis progress. Your answer is:
**[GT]** True

**[Question Type] Free-Form**
**[Perceptual Task]** All
**[Instruction]**
Please describe the image. You can describe it from these aspects: the subject of the image, the profession of the person, the artistic style of the image.
**[GT]** Scientist, line art style

Figure 19: Profession

**[Subject] Celebrity**

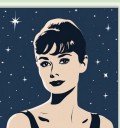

**[Prompt]** face portrait of Audrey Hepburn ,Starry sky background, minimalist style, simple, clean, uncluttered, modern, elegant
**[Negative Prompt]** ornate, complicated, highly detailed, cluttered, disordered, messy, noisy

**[Question Type] Multi-Choice**
**[Perceptual Task]** Celebrity
**[Instruction]**
Question: Who is the celebrity in the image?
(A) Humphrey Bogart   (B) Kate Winslet   (C) Audrey Hepburn   (D) Cameron Diaz
Hint: Please answer the question and provide the correct option letter, e.g., (A), (B), (C), (D), at the end. Do not contain the analysis progress.
Your answer is:
**[GT]** Audrey Hepburn

**[Question Type] True-or-False**
**[Perceptual Task]** Background
**[Instruction]**
Question: Is the background in the picture near the beach ?
(A) True   (B) False
Hint: Please answer the question and provide the correct option letter, e.g., (A), (B), at the end. Do not contain the analysis progress. Your answer is:
**[GT]** False

**[Question Type] Free-Form**
**[Perceptual Task]** All
**[Instruction]**
Please describe the image. You can describe it from these aspects: the identity of the person, the background, the artistic style of the image.
**[GT]** Audrey Hepburn, Starry sky background, minimalist style

Figure 20: Celebrity

**[Subject] Action**

**[Prompt]** a person skiing, foggy forest background, hyperrealistic art, extremely high-resolution details, photographic, realism pushed to extreme, fine texture, incredibly lifelike
**[Negative Prompt]** simplified, abstract, unrealistic, impressionistic, low resolution

**[Question Type] Multi-Choice**
**[Perceptual Task]** Action
**[Instruction]**
Question: What is the person doing in the picture?
(A) Driving a car    (B) reading a book    (C) gardening    (D) skiing
Hint: Please answer the question and provide the correct option letter, e.g., (A), (B), (C), (D), at the end. Do not contain the analysis progress.
Your answer is:
**[GT]** Skiing

**[Question Type] True-or-False**
**[Perceptual Task]** Background
**[Instruction]**
Question: Is the background in the picture foggy forest background?
(A) True    (B) False
Hint: Please answer the question and provide the correct option letter, e.g., (A), (B), at the end. Do not contain the analysis progress. Your answer is:
**[GT]** True

**[Question Type] Free-Form**
**[Perceptual Task]** All
**[Instruction]**
Please describe the image. You can describe it from these aspects: the action of the person, the background, the artistic style of the image.
**[GT]** skiing, foggy forest background, hyperrealism style

Figure 21: Action

**[Subject] Landmark**

**[Prompt]** the overall The Forbidden City, watercolor painting, vibrant, beautiful, painterly, detailed, textural, artistic
**[Negative Prompt]** anime, photorealistic, 35mm film, deformed, glitch, low contrast, noisy

**[Question Type] Multi-Choice**
**[Perceptual Task]** Style
**[Instruction]**
Question: What is the style of the picture?
(A) Watercolor    (B) Long exposure    (C) Monochrome    (D) GTA
Hint: Please answer the question and provide the correct option letter, e.g., (A), (B), (C), (D), at the end. Do not contain the analysis progress.
Your answer is:
**[GT]** Watercolor

**[Question Type] True-or-False**
**[Perceptual Task]** Landmark
**[Instruction]**
Question: Is the landmark in the picture The Forbidden City?
(A) True    (B) False
Hint: Please answer the question and provide the correct option letter, e.g., (A), (B), at the end. Do not contain the analysis progress. Your answer is:
**[GT]** True

**[Question Type] Free-Form**
**[Perceptual Task]** All
**[Instruction]**
Please describe the image. You can describe it from these aspects: the subject of the image, the artistic style of the image.
**[GT]** The Forbidden City, watercolor style

Figure 22: Landmark

**[Subject] Face**

**[Prompt]** face portrait of a 25 years old male, African race, smiling happily, countryside with rolling hills background, iphone photo, large depth of field, deep depth of field, highly detailed
**[Negative Prompt]** drawing, painting, crayon, sketch, graphite, impressionist, noisy, blurry, soft, deformed, ugly, shallow depth of field, bokeh

---

**[Question Type] Multi-Choice**
**[Perceptual Task]** Age
**[Instruction]**

Question: How old is the person in the image?
(A) 25 years old    (B) 80 years old    (C) 14 years old    (D) 40 years old
Hint: Please answer the question and provide the correct option letter, e.g., (A), (B), (C), (D), at the end. Do not contain the analysis progress.
Your answer is:
**[GT]** 25 years old

---

**[Question Type] Multi-Choice**
**[Perceptual Task]** Gender
**[Instruction]**

Question: What is the gender of the person in the picture?
(A) Male    (B) Female
Hint: Please answer the question and provide the correct option letter, e.g., (A), (B), at the end. Do not contain the analysis progress.
Your answer is:
**[GT]** Male

---

**[Question Type] True-or-False**
**[Perceptual Task]** Race
**[Instruction]**

Question: Is the person in the picture a female ?
(A) True    (B) False
Hint: Please answer the question and provide the correct option letter, e.g., (A), (B), at the end. Do not contain the analysis progress.
Your answer is:
**[GT]** False

---

**[Question Type] True-or-False**
**[Perceptual Task]** Expression
**[Instruction]**

Question: Is the person in the picture smiling happily ?
(A) True    (B) False
Hint: Please answer the question and provide the correct option letter, e.g., (A), (B), at the end. Do not contain the analysis progress.
Your answer is:
**[GT]** True

---

**[Question Type] Free-Form**
**[Perceptual Task]** All
**[Instruction]**

Please describe the image. You can describe it from these aspects: the person's gender, age, facial expression, race, the background, the artistic style of the image.
**[GT]** male, 25 years old, smiling happily, African race, countryside with rolling hills background background, iphone photographic style.

Figure 23: Face

**[Subject] Animal**

**[Prompt]** 1 orange color owl, on the lawn, watercolor painting, vibrant, beautiful, painterly, detailed, textural, artistic
**[Negative Prompt]** anime, photorealistic, 35mm film, deformed, glitch, low contrast, noisy

---

**[Question Type] Multi-Choice**
**[Perceptual Task]** Animal
**[Instruction]**
Question: What is the animal in the image?
(A) Owl   (B) Butterfly   (C) Crow   (D) Parrot
Hint: Please answer the question and provide the correct option letter, e.g., (A), (B), (C), (D), at the end. Do not contain the analysis progress.
Your answer is:
**[GT]** Owl

---

**[Question Type] Multi-Choice**
**[Perceptual Task]** Color
**[Instruction]**
Question: What is the color of the owl in the picture?
(A) Red color   (B) Orange color   (C) Yellow color   (D) Blue color
Hint: Please answer the question and provide the correct option letter, e.g., (A), (B), (C), (D), at the end. Do not contain the analysis progress.
Your answer is:
**[GT]** Orange color

---

**[Question Type] True-or-False**
**[Perceptual Task]** Background
**[Instruction]**
Question: Is the background in the picture urban street background?
(A) True   (B) False
Hint: Please answer the question and provide the correct option letter, e.g., (A), (B), at the end. Do not contain the analysis progress.
Your answer is:
**[GT]** False

---

**[Question Type] True-or-False**
**[Perceptual Task]** Style
**[Instruction]**
Question: Is the picture watercolor style?
(A) True   (B) False
Hint: Please answer the question and provide the correct option letter, e.g., (A), (B), at the end. Do not contain the analysis progress.
Your answer is:
**[GT]** True

---

**[Question Type] Free-Form**
**[Perceptual Task]** All
**[Instruction]**
Please describe the image. You can describe it from these aspects: the subject of the image, its color, the background, the artistic style of the image.
**[GT]** orange color owl, on the lawn background, watercolor style

Figure 24: Animal

**[Subject] Object**

**[Prompt]** 1 purple color bus, snow-covered landscape background, comic style, graphic illustration, comic art, graphic novel art, vibrant, highly detailed
**[Negative Prompt]** photograph, deformed, glitch, noisy, realistic, stock photo

**[Question Type] Multi-Choice**
**[Perceptual Task]** Object
**[Instruction]**
Question: What is the object in the image?
(A) Bus    (B) Frisbee    (C) Basketball    (D) Laptop
Hint: Please answer the question and provide the correct option letter, e.g., (A), (B), (C), (D), at the end. Do not contain the analysis progress.
Your answer is:
**[GT]** Bus

**[Question Type] Multi-Choice**
**[Perceptual Task]** Background
**[Instruction]**
Question: What is the background in the picture?
(A) On the lawn    (B) Under the water    (C) On the airplane    (D) Snow-covered landscape background
Hint: Please answer the question and provide the correct option letter, e.g., (A), (B), (C), (D), at the end. Do not contain the analysis progress.
Your answer is:
**[GT]** Snow-covered landscape background

**[Question Type] True-or-False**
**[Perceptual Task]** Color
**[Instruction]**
Question: Is the color of the bus purple color?
(A) True    (B) False
Hint: Please answer the question and provide the correct option letter, e.g., (A), (B), at the end. Do not contain the analysis progress.
Your answer is:
**[GT]** True

**[Question Type] True-or-False**
**[Perceptual Task]** Style
**[Instruction]**
Question: Is the picture comic book style?
(A) True    (B) False
Hint: Please answer the question and provide the correct option letter, e.g., (A), (B), at the end. Do not contain the analysis progress.
Your answer is:
**[GT]** True

**[Question Type] Free-Form**
**[Perceptual Task]** All
**[Instruction]**
Please describe the image. You can describe it from these aspects: the subject of the image, its color, the background, the artistic style of the image.
**[GT]** purple color bus, snow-covered landscape background background, comic book

Figure 25: Object

**[Subject] Clothes**

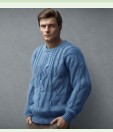

**[Prompt]** 1 blue color sweater, hyperrealistic art, extremely high-resolution details, photographic, realism pushed to extreme, fine texture, incredibly lifelike
**[Negative Prompt]** simplified, abstract, unrealistic, impressionistic, low resolution

**[Question Type] Multi-Choice**
**[Perceptual Task]** Style
**[Instruction]**
Question: What is the clothes in the image?
(A) Cargo pants   (B) Sweater   (C) Suit pants   (D) Trench coat
Hint: Please answer the question and provide the correct option letter, e.g., (A), (B), (C), (D), at the end. Do not contain the analysis progress.
Your answer is:
**[GT]** Sweater

**[Question Type] True-or-False**
**[Perceptual Task]** Landmark
**[Instruction]**
Question: Is the color of the sweater in the picture blue color?
(A) True   (B) False
Hint: Please answer the question and provide the correct option letter, e.g., (A), (B), at the end. Do not contain the analysis progress. Your answer is:
**[GT]** True

**[Question Type] Free-Form**
**[Perceptual Task]** All
**[Instruction]**
Please describe the image. You can describe it from these aspects: the subject of the image, its color, the artistic style of the image.
**[GT]** blue color sweater, hyperrealism style

Figure 26: Clothes

**[Subject] Food**

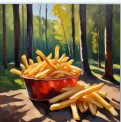

**[Prompt]** 1 french fries, in forest, impressionist painting, loose brushwork, vibrant color, light and shadow play, captures feeling over form
**[Negative Prompt]** anime, photorealistic, 35mm film, deformed, glitch, low contrast, noisy

**[Question Type] Multi-Choice**
**[Perceptual Task]** Food
**[Instruction]**
Question: What is the food in the image?
(A) Dumplings (B) Egg fried rice   (C) donuts   (D) French fries
Hint: Please answer the question and provide the correct option letter, e.g., (A), (B), (C), (D), at the end. Do not contain the analysis progress.
Your answer is:
**[GT]** French fries

**[Question Type] True-or-False**
**[Perceptual Task]** Background
**[Instruction]**
Question: Is the background in the picture urban street background?
(A) True   (B) False
Hint: Please answer the question and provide the correct option letter, e.g., (A), (B), at the end. Do not contain the analysis progress. Your answer is:
**[GT]** False

**[Question Type] Free-Form**
**[Perceptual Task]** All
**[Instruction]**
Please describe the image. You can describe it from these aspects: the subject of the image, the background, the artistic style of the image.
**[GT]** french fries, in forest background, impressionist style

Figure 27: Food

---

**[Subject] Movie**

**[Prompt]** a scene from Iron Man, pop Art style, bright colors, bold outlines, popular culture themes, ironic or kitsch
**[Negative Prompt]** ugly, deformed, noisy, blurry, low contrast, realism, photorealistic, minimalist

**[Question Type] Multi-Choice**
**[Perceptual Task]** Movie
**[Instruction]**
Question: Which movie is the scene in the picture from?
(A) Iron Man   (B) The Pianist   (C) The God Father   (D) Avengers: Infinity War
Hint: Please answer the question and provide the correct option letter, e.g., (A), (B), (C), (D), at the end. Do not contain the analysis progress.
Your answer is:
**[GT]** Iron Man

**[Question Type] True-or-False**
**[Perceptual Task]** Style
**[Instruction]**
Question: Is the picture pop art style?
(A) True   (B) False
Hint: Please answer the question and provide the correct option letter, e.g., (A), (B), at the end. Do not contain the analysis progress. Your answer is:
**[GT]** True

**[Question Type] Free-Form**
**[Perceptual Task]** All
**[Instruction]**
Please describe the image. You can describe it from these aspects: the movie from which the scene is made, the artistic style of the image.
**[GT]** Iron Man, pop art style

Figure 28: Movie

---

**[Subject] TV Show**

**[Prompt]** a scene from Peaky Blinders, stained glass style, vibrant, beautiful, translucent, intricate, detailed
**[Negative Prompt]** ugly, deformed, noisy, blurry, low contrast, realism, photorealistic

**[Question Type] Multi-Choice**
**[Perceptual Task]** Style
**[Instruction]**
Question: What is the style of the picture?
(A) Nautical   (B) Manga   (C) Stained glass   (D) Hyperrealism style
Hint: Please answer the question and provide the correct option letter, e.g., (A), (B), (C), (D), at the end. Do not contain the analysis progress.
Your answer is:
**[GT]** Stained glass

**[Question Type] True-or-False**
**[Perceptual Task]** TV show
**[Instruction]**
Question: Is the scene in the picture from the tv show Peaky Blinders?
(A) True   (B) False
Hint: Please answer the question and provide the correct option letter, e.g., (A), (B), at the end. Do not contain the analysis progress. Your answer is:
**[GT]** True

**[Question Type] Free-Form**
**[Perceptual Task]** All
**[Instruction]**
Please describe the image. You can describe it from these aspects: the tv show from which the scene is made, the artistic style of the image.
**[GT]** Peaky Blinders, stained glass style

Figure 29: TV

**[Subject] Anime**

**[Prompt]** a scene from Saint Seiya, constructivist style, geometric shapes, bold colors, dynamic composition, propaganda art style
**[Negative Prompt]** realistic, photorealistic, low contrast, plain, simple, abstract expressionism

**[Question Type] Multi-Choice**
**[Perceptual Task]** Anime
**[Instruction]**
   Question: Which anime is the scene in the picture from?
   (A) The Simpsons   (B) Detective Conan   (C) Eyeshield 21   (D) Saint Seiya
   Hint: Please answer the question and provide the correct option letter, e.g., (A), (B), (C), (D), at the end. Do not contain the analysis progress.
   Your answer is:
**[GT]** Saint Seiya

**[Question Type] True-or-False**
**[Perceptual Task]** Style
**[Instruction]**
   Question: Is the picture hyperrealism style?
   (A) True   (B) False
   Hint: Please answer the question and provide the correct option letter, e.g., (A), (B), at the end. Do not contain the analysis progress. Your answer is:
**[GT]** False

**[Question Type] Free-Form**
**[Perceptual Task]** All
**[Instruction]**
   Please describe the image. You can describe it from these aspects: the anime from which the scene is made, the artistic style of the image.
**[GT]** Saint Seiya, constructivist style

Figure 30: Anime

**[Subject] Text**

**[Prompt]** A book cover with logo 'petiole', impressionist painting, loose brushwork, vibrant color, light and shadow play, captures feeling over form
**[Negative Prompt]** anime, photorealistic, 35mm film, deformed, glitch, low contrast, noisy

**[Question Type] Multi-Choice**
**[Perceptual Task]** Text
**[Instruction]**
   Question: What are all the scene text in the image?
   (A) "petiole"   (B) "pleuron"   (C) "designer"   (D) "evolution"
   Hint: Please answer the question and provide the correct option letter, e.g., (A), (B), (C), (D), at the end. Do not contain the analysis progress.
   Your answer is:
**[GT]** "petiole"

**[Question Type] True-or-False**
**[Perceptual Task]** Text
**[Instruction]**
   Question: Is the scene text in the picture 'people' ?
   (A) True   (B) False
   Hint: Please answer the question and provide the correct option letter, e.g., (A), (B), at the end. Do not contain the analysis progress. Your answer is:
**[GT]** False

**[Question Type] Free-Form**
**[Perceptual Task]** Text
**[Instruction]**
   Please describe all the scene text in the image.
**[GT]** "petiole"

Figure 31: OCR

# I   JSON STRUCTURE FOR EVALUATION

Here, We present two example JSON structures generated by Dysca for evaluation in Listing 1.

Listing 1: Example JSON structure for evaluation

```
{
    "id": 1,
    "images": [
      "images/1.png"
    ],
    "prompt": "face portrait of a 25 years old female, Indian race,
        ↪calm and placid, on Mars, retro arcade style, 8-bit,
        ↪pixelated, vibrant, classic video game, old school gaming,
        ↪reminiscent of 80s and 90s arcade games",
    "negative_prompt": "modern, ultra-high resolution, photorealistic,
        ↪ 3D",
    "instruction": "Question: What is the background in the picture?\
        ↪nChoices:\n(A) on the lawn\n(B) under the water background\n(
        ↪C) sunset background\n(D) on Mars\nHint: Please answer the
        ↪question and provide the correct option letter, e.g., (A), (B
        ↪), (C), (D), at the end. Do not contain the analysis progress
        ↪.\nYour answer is:",
    "question": "What is the background in the picture?",
    "answer": "on Mars",
    "question_type": "multi choice",
    "task": "recognition",
    "options": [
      "on the lawn",
      "under the water background",
      "sunset background",
      "on Mars"
    ],
    "question_majority": "background",
    "granularity": "coarse"
},
{
    "id": 2,
    "images": [
      "images/2.png"
    ],
    "prompt": "1 white color cat, on Mars, thick layered papercut art
        ↪of, deep 3D, volumetric, dimensional, depth, thick paper,
        ↪high stack, heavy texture, tangible layers",
    "negative_prompt": "2D, flat, thin paper, low stack, smooth
        ↪texture, painting, drawing, photo, deformed",
    "instruction": "Question: Is there a cat in the picture?\nChoices
        ↪:\n(A) False\n(B) True\nHint: Please answer the question and
        ↪provide the correct option letter, e.g., (A), (B), (C), (D),
        ↪at the end. Do not contain the analysis progress.\nYour
        ↪answer is:",
    "question": "Is there a cat in the picture?",
    "answer": "True",
    "question_type": "true or false",
    "task": "recognition",
    "options": [
      "False",
      "True"
    ],
    "question_majority": "animal",
    "granularity": "fine"
}
```

## J  DATA SHEET

We follow the documentation frameworks provided by Gebru et al. Gebru et al. (2021).

### J.1  MOTIVATION

**For what purpose was the dataset created?** Was there a specific task in mind? Was there a specific gap that needed to be filled? Please provide a description.

- The proposed dataset is used for evaluating current LVLMs perception ability. We use the synthesis images to prevent the potential data leakage problem in current benchmarks. The dataset test LVLMs in 20 subtasks under 4 scenarios and 3 question type, revealing the existing drawbacks of current LVLMs.

**Who created the dataset (e.g., which team, research group) and on behalf of which entity (e.g., company, institution, organization)?**

- The dataset is created by the AI Safety and Trustworthiness Group on behalf of Key Laboratory of AI Safety of CAS, Institute of Computing Technology.

**Who funded the creation of the dataset?** If there is an associated grant, please provide the name of the grant or and the grant name and number.

- This work is partially supported by Strategic Priority Research Program of the Chinese Academy of Sciences (No. XDB0680202), Beijing Nova Program (20230484368), Suzhou Frontier Technology Research Project (No. SYG202325), and Youth Innovation Promotion Association CAS.

### J.2  COMPOSITION

**What do the instances that comprise the dataset represent (e.g., documents, photos, people, countries)?** Are there multiple types of instances (e.g., movies, users, and ratings; people and interactions between them; nodes and edges)? Please provide a description.

- We show the instances list in Tab. 11. The detailed word we collect for metadata $M$ are shown at `https://github.com/Robin-WZQ/Dysca`.

**How many instances are there in total (of each type, if appropriate)?**

- There are a total of 20 subtasks in our Dysca. For details of each subtasks please see refer Fig. 2.

**Does the dataset contain all possible instances or is it a sample (not necessarily random) of instances from a larger set?** If the dataset is a sample, then what is the larger set? Is the sample representative of the larger set (e.g., geographic coverage)? If so, please describe how this representativeness was validated/verified. If it is not representative of the larger set, please describe why not (e.g., to cover a more diverse range of instances, because instances were withheld or unavailable).

- No. The images in Dysca are completely generated from scratch.

**What data does each instance consist of?** "Raw" data (e.g., unprocessed text or images) or features? In either case, please provide a description.

- Each instance consists of the prompt, the image generated by stable diffusion, the question and corresponding answer.

**Is there a label or target associated with each instance?** If so, please provide a description.

- Yes, Dysca provides the ground truth for each instance.

**Is any information missing from individual instances?** If so, please provide a description, explaining why this information is missing (e.g., because it was unavailable). This does not include intentionally removed information, but might include, e.g., redacted text.

- No.

**Are relationships between individual instances made explicit (e.g., users' movie ratings, social network links)?** If so, please describe how these relationships are made explicit.

- There are no relationships between individual instances.

**Are there recommended data splits (e.g., training, development/validation, testing)?** If so, please provide a description of these splits, explaining the rationale behind them.

- Following our motivation, the entire proposed dataset is used for testing purposes.

**Are there any errors, sources of noise, or redundancies in the dataset?** If so, please provide a description.

- Errors in image generation resulting from stable diffusion are unavoidable. However, we have performed dataset cleaning to minimize these errors. Furthermore, the stability experiment in Appendix B demonstrates that these errors do not affect the overall evaluation results of the dataset.

**Is the dataset self-contained, or does it link to or otherwise rely on external resources (e.g., websites, tweets, other datasets)?** If it links to or relies on external resources, a) are there guarantees that they will exist, and remain constant, over time; b) are there official archival versions of the complete dataset (i.e., including the external resources as they existed at the time the dataset was created); c) are there any restrictions (e.g., licenses, fees) associated with any of the external resources that might apply to a dataset consumer? Please provide descriptions of all external resources and any restrictions associated with them, as well as links or other access points, as appropriate.

- The proposed Dysca dose not rely on any external resources.

**Does the dataset contain data that might be considered confidential (e.g., data that is protected by legal privilege or by doctor-patient confidentiality, data that includes the content of individuals' non-public communications)?** If so, please provide a description.

- No.

**Does the dataset contain data that, if viewed directly, might be offensive, insulting, threatening, or might otherwise cause anxiety?** If so, please describe why.

- No. To ensure that the generated images do not contain offensive, insulting, threatening, or anxiety-inducing content, we manually filter out words from the metadata $M$ that could potentially trigger the diffusion model to generate such images. Safety checker also used to further avoid unsafe image generation.

**Does the dataset relate to people?** If not, you may skip the remaining questions in this section.

- Yes.

**Does the dataset identify any subpopulations (e.g., by age, gender)?** If so, please describe how these subpopulations are identified and provide a description of their respective distributions within the dataset.

- Yes. There are the age, gender and race recognition subtasks in Dysca. Each of them are divided to several subpopulations and the selection of these subpopulations is based on the ability of stable diffusion to generate the representative subpopulations.

**Is it possible to identify individuals (i.e., one or more natural persons), either directly or indirectly (i.e., in combination with other data) from the dataset?** If so, please describe how.

- Yes. There is the celebrity recognition task in our dataset, where 50 well-know celebrity are chosen. We choose the celebrity who can be generated well by stable diffusion XL.

**Does the dataset contain data that might be considered sensitive in any way (e.g., data that reveals race or ethnic origins, sexual orientations, religious beliefs, political opinions or union memberships, or locations; financial or health data; biometric or genetic data; forms of government identification, such as social security numbers; criminal history)?** If so, please provide a description.

- No, our benchmark does not contain any sensitive data.

### J.3 COLLECTION PROCESS

**How was the data associated with each instance acquired?** Was the data directly observable (e.g., raw text, movie ratings), reported by subjects (e.g., survey responses), or indirectly inferred/derived from other data (e.g., part-of-speech tags, model based guesses for age or language)? If data was reported by subjects or indirectly inferred/derived from other data, was the data validated/verified? If so, please describe how.

- We display the detailed explanation in Tab. 11.

**What mechanisms or procedures were used to collect the data (e.g., hardware apparatus or sensor, manual human curation, software program, software API)?** How were these mechanisms or procedures validated?

- We collect the data by manual human curation.

**If the dataset is a sample from a larger set, what was the sampling strategy (e.g., deterministic, probabilistic with specific sampling probabilities)?**

- No.

**Who was involved in the data collection process (e.g., students, crowdworkers, contractors) and how were they compensated (e.g., how much were crowdworkers paid)?**

- We collect the metadata of Tab. 11 by authors. The images are generated by stable diffusion and labels of each image are also automatically generated.

**Over what timeframe was the data collected? Does this timeframe match the creation timeframe of the data associated with the instances (e.g., recent crawl of old news articles)?** If not, please describe the timeframe in which the data associated with the instances was created.

- Our dataset was conducted in April of 2024, but the results do not depend on the date of data collection.

**Were any ethical review processes conducted (e.g., by an institutional review board)?** If so, please provide a description of these review processes, including the outcomes, as well as a link or other access point to any supporting documentation.

- No.

**Did you collect the data from the individuals in question directly, or obtain it via third parties or other sources (e.g., websites)?**

- No.

**Were the individuals in question notified about the data collection?** If so, please describe (or show with screenshots or other information) how notice was provided, and provide a link or other access point to, or otherwise reproduce, the exact language of the notification itself.

- N/A. Our Dysca does not involve the collection from the individuals.

**Did the individuals in question consent to the collection and use of their data?** If so, please describe (or show with screenshots or other information) how consent was requested and provided, and provide a link or other access point to, or otherwise reproduce, the exact language to which the individuals consented.

- N/A. Our Dysca does not involve the collection from the individuals.

**If consent was obtained, were the consenting individuals provided with a mechanism to revoke their consent in the future or for certain uses?** If so, please provide a description, as well as a link or other access point to the mechanism (if appropriate).

- N/A. Our Dysca does not involve the collection from the individuals.

**Has an analysis of the potential impact of the dataset and its use on data subjects (e.g., a data protection impact analysis) been conducted?** If so, please provide a description of this analysis, including the outcomes, as well as a link or other access point to any supporting documentation.

- No.

### J.4 PREPROCESSING/CLEANING/LABELING

**Was any preprocessing/cleaning/labeling of the data done (e.g., discretization or bucketing, tokenization, part-of-speech tagging, SIFT feature extraction, removal of instances, processing of missing values)?** If so, please provide a description. If not, you may skip the remaining questions in this section.

- Yes. We leverage the off-the-shelf models, i.e., PP-OCRv3 Li et al. (2022) and CLIP-L-14 Radford et al. (2021), to clean the data. PP-OCRv3 Li et al. (2022) is leveraged as the filter to exclude the failure image that TextDiffusion2 Chen et al. (2023a) generates the wrong text on the image. For the other images, we use CLIP-L-14 Radford et al. (2021) to filter out the images with low text-image consistency.

**Was the "raw" data saved in addition to the preprocessed/cleaned/labeled data (e.g., to support unanticipated future uses)?** If so, please provide a link or other access point to the "raw" data.

- Yes. We have saved all the data. However, most of these images are filtered and considered to be useless.

**Is the software that was used to preprocess/clean/label the data available?** If so, please provide a link or other access point.

- Yes. CLIP-L-14 can be downloaded at `https://huggingface.co/docs/transformers/v4.41.3/en/model_doc/clip#transformers.CLIPModel`.
  PP-OCRv3 can be downloaded at `https://github.com/PaddlePaddle/PaddleOCR/blob/main/README_en.md`

### J.5 USES

**Has the dataset been used for any tasks already?** If so, please provide a description.

- No. The proposed dataset is the novel one which is used for evaluation current LVLMs perception ability.

**Is there a repository that links to any or all papers or systems that use the dataset?** If so, please provide a link or other access point.

- Yes. We plan to create a section on the project homepage to keep track of LVLMs papers for researchers to analyze and compare.

**What (other) tasks could the dataset be used for?**

- In this work, we do not explore the possibility of utilizing our benchmark for model training / fine-tuning. Our primary goal in this paper is to provide a large-scale evaluation benchmark that addresses the issue of data leakage in current multimodal evaluation benchmarks and offers evaluation results across multiple subtasks, scenarios, question types and styles. Nevertheless, considering that Dysca has the capability to synthesize high-resolution and unlimited amounts of annotated multimodal data, we believe that Dysca also holds potential as a training data synthesis tool for LVLMs.

**Is there anything about the composition of the dataset or the way it was collected and preprocessed/cleaned/labeled that might impact future uses?** For example, is there anything that a dataset consumer might need to know to avoid uses that could result in unfair treatment of individuals or groups (e.g., stereotyping, quality of service issues) or other risks or harms (e.g., legal risks, financial harms)? If so, please provide a description. Is there anything a dataset consumer could do to mitigate these risks or harms?

- Yes.

**Are there tasks for which the dataset should not be used?** If so, please provide a description.

- The proposed dataset should not be used to generate offensive data.

## J.6 DISTRIBUTION

**Will the dataset be distributed to third parties outside of the entity (e.g., company, institution, organization) on behalf of which the dataset was created?** If so, please provide a description.

- Yes.

**How will the dataset will be distributed (e.g., tarball on website, API, GitHub)?** Does the dataset have a digital object identifier (DOI)?

- We will open-source our dataset on our GitHub project homepage. At the moment, we do not have a DOI number.

**When will the dataset be distributed?**

- The dataset can be downloaded right now.

**Will the dataset be distributed under a copyright or other intellectual property (IP) license, and/or under applicable terms of use (ToU)?** If so, please describe this license and/or ToU, and provide a link or other access point to, or otherwise reproduce, any relevant licensing terms or ToU, as well as any fees associated with these restrictions.

- The licence of Dysca is "CreativeML Open RAIL++-M", which follows the licence set by the Stable Diffusion XL.

**Have any third parties imposed IP-based or other restrictions on the data associated with the instances?** If so, please describe these restrictions, and provide a link or other access point to, or otherwise reproduce, any relevant licensing terms, as well as any fees associated with these restrictions.

- No.

**Do any export controls or other regulatory restrictions apply to the dataset or to individual instances?** If so, please describe these restrictions, and provide a link or other access point to, or otherwise reproduce, any supporting documentation.

- Not yet.

## J.7 MAINTENANCE

**Who will be supporting/hosting/maintaining the dataset?**

- Jie Zhang and Zhongqi Wang are hosting and maintaining the dataset.

**How can the owner/curator/manager of the dataset be contacted (e.g., email address)?**

- Email: zhangjie@ict.ac.cn

**Is there an erratum?** If so, please provide a link or other access point.

- No.

**Will the dataset be updated (e.g., to correct labeling errors, add new instances, delete instances)?** If so, please describe how often, by whom, and how updates will be communicated to dataset consumers (e.g., mailing list, GitHub)?

- There are no plans at the moment, but if there are updates, they will be announced, and the download source will be updated on the project homepage.

**If the dataset relates to people, are there applicable limits on the retention of the data associated with the instances (e.g., were the individuals in question told that their data would be retained for a fixed period of time and then deleted)?** If so, please describe these limits and explain how they will be enforced.

- No.

**Will older versions of the dataset continue to be supported/hosted/maintained?** If so, please describe how. If not, please describe how its obsolescence will be communicated to dataset consumers.

- Yes. If there are any updates, the previous version of the dataset will also be shared on website for download.

**If others want to extend/augment/build on/contribute to the dataset, is there a mechanism for them to do so?** If so, please provide a description. Will these contributions be validated/verified? If so, please describe how. If not, why not? Is there a process for communicating/distributing these contributions to dataset consumers? If so, please provide a description.

- Yes. We welcome and encourage researchers to extend/augment/build on/contribute to our dataset for non-profit purposes without the need for prior notification.