# OpenReview forum: "Dysca: A Dynamic and Scalable Benchmark for Evaluating Perception Ability of LVLMs"
_ICLR.cc/2025/Conference — ICLR 2025 Poster_

### Official Review · Reviewer_3C5q · 2024-10-21

**Soundness:** 3
**Presentation:** 3
**Contribution:** 3
**Rating:** 6
**Confidence:** 4

**Summary:**

This is a data paper. The authors introduce Dysca, a new benchmark for evaluating Large Vision-Language Models (LVLMs) from the perspective of assessing multi-stylized images and noisy scenarios, as well as test environments involving corruption, print attacks, and adversarial attacks. Dysca uses SDXL images to dynamically synthesize varied image styles and questions, enabling evaluation across 51 styles and 20 subtasks under four scenarios: clean, corrupted, print attacked, and adversarially attacked. They also provide interesting insights into LVLMs' capabilities and performance under diverse conditions.

**Strengths:**

1. The reviewer appreciates the authors' attempt to address a unique aspect of LVLM benchmarks: the evaluation of multi-stylized images and noisy scenarios. This perspective is intriguing and relevant.

2. The paper is well-written with clear motivations and a logical structure. The presentation is visually appealing with sophisticated diagrams. The detailed introduction of the benchmark data facilitates reader comprehension.

3. The experimental analysis provides several valuable insights. Notably, there is a high correlation between their evaluation rankings and those of non-synthetic benchmarks, highlighting deficiencies in current LVLMs when facing various question types, image styles, and scenarios.

4. The overall effort in the work is solid. Using an automated synthesis method to generate an evaluation benchmark is an unconventional but intriguing and worthwhile exploration.

**Weaknesses:**

1. The major concern with this dataset is the use of image generation tool for automatic vision creation. For such images to serve as a gold standard for MLLM evaluation, they must meet a critical assumption: the generated images are perfect and error-free. Unfortunately, even today's state-of-the-art SDXL models, which can produce high-quality images, still blend real and synthetic styles rather than producing wholly realistic images. As humans, we can typically recognize diffusion-synthesized images. SD models excel at creating artistic-style images, but for sure, non-realistic images inevitably introduce bias in evaluations on realistic images. Also, many automatically synthesized images that contain scene texts are often of low quality, i.e., containing inaccurate scene text. Such biases could largely skew model evaluations.
Therefore, the reviewer suggests that the authors add ample discussion regarding this issue. For example, the authors could include some in-depth analytical experiments to validate the concerns raised above.


2. The entire process is fully automated, including the use of CLIP for automated quality assessment of generated QA image-text pairs, without any human inspection, which is problematic.
The reviewer suggests that the authors analyze the impact of all automatically constructed data on quality, or consider using a more reasonable evaluation method for assessment.

3. While using automatically synthesized images significantly reduces annotation costs, as a testing set, it is a dubious claim without (or with less) real value. Conversely, generating a large-scale training set through automated image synthesis is undoubtedly positive.
Therefore, the reviewer hopes that the authors can further explain and clarify this matter.



4. The main focus of this data should be on testing LVLMs' performance in multi-stylized and noisy images and in scenarios involving corruption, print attacks, and adversarial attacks. If a LVLM benchmark is limited to these aspects, its utility and contribution are significantly diminished.
Therefore, the reviewer hopes that the authors can further explain and clarify this matter.


5. "_Although they claim that the questions are re-annotated, previous work (Chen et al., 2024) has demonstrated that these benchmarks have unintentionally leaked into the training data of LLMs and LVLMs_": This claim might largely be inaccurate, as Chen et al. (2024) only discussed very early LVLM benchmarks, which were simplistic and not re-annotated. However, most current benchmarks have re-annotated QA texts, almost solving the data leakage issue.
Please provide further details.

6. The authors claim a large dataset size with 617K visual-language QA pairs. However, for evaluation benchmarks, sheer size is not as important as having a more varied set of subsets assessing different aspects and capabilities [1]. Unfortunately, Dysca only covers 20 image perception subtasks, offering no significant advantage over SEED-Bench2 and BenchLMM. Moreover, it appears to be surpassed by existing work (e.g., [2]) in both data quantity and evaluation scope, which the authors have not compared or mentioned.
The reviewer suggests adding more comparisons for this.


7. As a benchmark, it evaluates too few (comparatively) LVLMs, limited to just a small subset of models (actually 16; the 26 LVLMs are somewhat overclaimed). There are many more state-of-the-art multimodal LLMs that should be included at least in the appendix. the reviewer suggests the authors include evaluations of more models next.
The reviewer strongly recommends adding more MLLMs for experimental comparison.



## Summary

Overall, if the authors had focused solely on evaluating multi-stylized images and noisy scenarios, it would have been a safer claim, even with automated image synthesis. However, by comparing it to current general-purpose LVLM benchmarks, the limitations and significant issues of automated image synthesis are exposed, particularly its inability to replace the importance of realistic images. This is my key suggestion for the authors.
Thus, "_We demonstrate for the first time that evaluating LVLMs using large-scale synthetic data is valid_" might be an overclaim, at least not sufficiently and safely proved in this paper.

Overall, the reviewer is open-minded. If the authors can actively and effectively address the concerns with more discussions, the reviewer would consider raising my rating.

---

[1] Are We on the Right Way for Evaluating Large Vision-Language Models? 2024.

[2] MEGA-Bench: Scaling Multimodal Evaluation to over 500 Real-World Tasks. 2024

-----------

After the reviewer-author interaction, the reviewer decided to increase the evaluation to positive as the authors provided enough detailed responses and clarification.

**Questions:**

1. Can the authors present the complete prompts used for image generation in the appendix? Could some prompts be displayed?

2. Did the authors assess the quality of automatically synthesized images? How is quality ensured when generating such a large scale of images, especially those containing text?

3. How were the QA pairs (text) constructed on such a large scale? Was ChatGPT utilized? or just all by rule-based template?

---

> ### Author Response · Authors · 2024-11-25
> **Response to Reviewer 3C5q (Part 1/4)**
>
> Thank you for your valuable feedback! We really appreciate that you found our work is intriguing and worthwhile exploration.
>
> > The major concern with this dataset is the use of image generation tool for automatic vision creation. For such images to serve as a gold standard for MLLM evaluation, they must meet a critical assumption: the generated images are perfect and error-free. Unfortunately, even today's state-of-the-art SDXL models, which can produce high-quality images, still blend real and synthetic styles rather than producing wholly realistic images. As humans, we can typically recognize diffusion-synthesized images. SD models excel at creating artistic-style images, but for sure, non-realistic images inevitably introduce bias in evaluations on realistic images. Also, many automatically synthesized images that contain scene texts are often of low quality, i.e., containing inaccurate scene text. Such biases could largely skew model evaluations. Therefore, the reviewer suggests that the authors add ample discussion regarding this issue. For example, the authors could include some in-depth analytical experiments to validate the concerns raised above.
>
> Good point. We understand the reviewers' concerns regarding the utility of image generation tool for automatic vision creation. To ensure the quality of our data, four steps are adopt as follows:
>
> 1. First, we manually remove difficult-to-generate foregrounds and attributes, along with backgrounds and styles that could heavily affect image content. We believe this process can serve as a coarse-grained method to eliminate samples that are highly likely to be generated incorrectly.
> 2. Then, we use CLIP for further image filtering. In the experiment, we randomly choose 1000 samples first and apply filtering with thresholds of 0.65, 0.7, 0.75, and 0.8 individually. Our findings indicate that using 0.75 as the threshold achieve a good balance between image correctness and data scale.
> 3. After that, we leverage top 6 LVLMs to eliminate any question-answer pairs where the models either answer incorrectly or indicate that the answer is not included among the options. we observe that nearly 100% of the samples filtered out by these models are incorrect.
> 4. Finally, we analyze the patterns in these incorrect samples, removing the associated vocabulary from our metadata and discarding all related samples.
>
> By meticulously refining the metadata manually and utilizing automated tools to assist in question filtering, Dysca ensures high-quality data synthesis. In the end, we filter out nearly 40% of low quality samples.
>
> Besides, we agree with the reviewer that benchmark creators should aim for a perfect and error-free benchmark. However, in practice, many mainstream benchmarks labelled by human contain significant labeling errors [R1]. For example, the ImageNet [R2] has at least 2,916 errors in its validation set, with an error rate of 6%; the QuickDraw [R3] has at least 5 million errors, with an error rate of approximately 10%. Despite the labeling noise in these benchmarks, it does not affect their validity as widely used evaluation benchmarks. In appendix C, we randomly sample 7.7K images (10% of our data) and manually select images that contain wrong content or potentially lead to incorrect responses. In the end, a total of 167 images (2.2%) are filtered. Compared with these benchmarks [R1], Dysca achieves a lower error rates than these benchmarks.
>
> In addition, Dysca also includes various question-answer pairs generated from prompts with realistic styles, such as “HDR photo” and "iPhone photo". These images are visually indistinguishable from those sampled from the real world. Furthermore, in the section 4.2.2 of the main paper, we find that the distribution between Dysca and real datasets is minimal, indicating that the evaluation results from our VQAs effectively reflect the model’s performance in real-world scenarios. As the reviewer says, *"As humans, we can typically recognize diffusion-synthesized images."* LVLMs also require the ability to recognize diffusion-synthesized images. We believe that Dysca can serve as a diverse and comprehensive evaluation benchmark and contribute to the community in evaluating LVLMs.

---

> > ### Author Response · Authors · 2024-11-25
> > **Response to Reviewer 3C5q (Part 2/4)**
> >
> > > The entire process is fully automated, including the use of CLIP for automated quality assessment of generated QA image-text pairs, without any human inspection, which is problematic. The reviewer suggests that the authors analyze the impact of all automatically constructed data on quality, or consider using a more reasonable evaluation method for assessment.
> >
> > We provide quantitative evaluation results regarding correctness in Table 9 of the Appendix, where we calculate the scores of 26 models on the original data and after removing erroneous data. For clarity, we display the results here.
> >
> > |        **Model**        | Before | After |
> > | :---------------------: | :----: | :---: |
> > |     Xcomposer-VL-7B     | 72.36  | 72.45 |
> > | InstructBLIP-Flan-T5-XL | 67.41  | 67.55 |
> > |          BLIP2          | 66.13  | 66.24 |
> > |      Qwen-VL-Chat       | 62.68  | 62.90 |
> > |         Shikra          | 62.52  | 62.78 |
> > |      LLaVa-1.5-13B      | 59.28  | 60.03 |
> > |       Shikra-VQA        | 58.64  | 58.68 |
> > |          Otter          | 55.49  | 55.73 |
> > |      LLaVa-1.5-7B       | 51.75  | 52.15 |
> > |   MiniGPT4-Vicuna-7B    | 42.64  | 42.43 |
> >
> > As observed, the score difference is negligible, making the automatic evaluation result reliable. We believe that with the development of deep generative models, Dysca will serve as an evolving benchmark with improving quality.
> >
> > > While using automatically synthesized images significantly reduces annotation costs, as a testing set, it is a dubious claim without (or with less) real value. Conversely, generating a large-scale training set through automated image synthesis is undoubtedly positive. Therefore, the reviewer hopes that the authors can further explain and clarify this matter.
> >
> > Good point. We agree with the statement that *"generating a large-scale training set through automated image synthesis is undoubtedly positive"*. In the Appendix B.1, we believe that Dysca also holds potential as a training data synthesis tool for LVLMs. However, exploring the use of synthetic data for training LVLMs is beyond the scope of this paper.
> >
> > But we strongly argue that using automatically synthesized images is not useless in evaluation. Firstly, low cost is only one of advantages of Dysca. We believe our dataset places greater emphasis on avoiding potential data leakage and being dynamically expandable to various image styles, multiple scenarios, and diverse evaluation tasks. Secondly, Dysca can conduct an in-depth analysis of the shortcomings of LVLMs. From the perspective of question formats, we observe inconsistencies in the performance of current LVLMs when handling multiple-choice and true-or-false questions. From the perspective of evaluation tasks, Dysca can specifically identify model performance differences across subtasks as well as within a subtask.
> >
> > We argue that Dysca can serve as a fine-grained probe to reveal vulnerability of current LVLMs and assist the community and model developers in understanding characteristics of models and implementing targeted improvements.
> >
> > > The main focus of this data should be on testing LVLMs' performance in multi-stylized and noisy images and in scenarios involving corruption, print attacks, and adversarial attacks. If a LVLM benchmark is limited to these aspects, its utility and contribution are significantly diminished. Therefore, the reviewer hopes that the authors can further explain and clarify this matter.
> >
> > We argue that our work is not merely a combination of previous efforts. The greatest advantage of our dataset compared to prior work lies in its dynamic scalability. Thus, our dataset can easily incorporate diverse evaluation perspectives and scenarios. This approach significantly reduces annotation costs compared to manually labeling synthetic images (e.g., MME). It also eliminates the risk of hallucinated annotations that may arise when using ChatGPT for labeling based on image prompts (e.g., JourneyDB).

---

> > > ### Author Response · Authors · 2024-11-25
> > > **Response to Reviewer 3C5q (Part 3/4)**
> > >
> > > > "*Although they claim that the questions are re-annotated, previous work (Chen et al., 2024) has demonstrated that these benchmarks have unintentionally leaked into the training data of LLMs and LVLMs*": This claim might largely be inaccurate, as Chen et al. (2024) only discussed very early LVLM benchmarks, which were simplistic and not re-annotated. However, most current benchmarks have re-annotated QA texts, almost solving the data leakage issue. Please provide further details.
> > >
> > > We sincerely appreciate the reviewers for raising concerns about the key statements in this paper. In this paper, the term "re-annotated" refers to VQAs derived by sampling from existing dataset images and annotating the original images to generate new question-answer pairs. We apologize for any confusion caused by unclear statement and will revise it in the final version of the paper.
> > >
> > > Besides, we acknowledge that using entirely new images and question-answer pairs makes data leakage unlikely. However, this does not guarantee that model developers will not illegally acquire validation/test samples for training in the future. In contrast, Dysca's evaluation data is uniquely generated using random seeds chosen by the testers, making it difficult for developers to access the test samples.
> > >
> > > > The authors claim a large dataset size with 617K visual-language QA pairs. However, for evaluation benchmarks, sheer size is not as important as having a more varied set of subsets assessing different aspects and capabilities [1]. Unfortunately, Dysca only covers 20 image perception subtasks, offering no significant advantage over SEED-Bench2 and BenchLMM. Moreover, it appears to be surpassed by existing work (e.g., [2]) in both data quantity and evaluation scope, which the authors have not compared or mentioned. The reviewer suggests adding more comparisons for this.
> > >
> > > We acknowledge that the number of evaluation dimensions is more important than the size of the evaluation dataset. This is also one of our advantages compared to previous evaluation datasets, as it enables us to assess LVLMs' performance from multiple perspectives and across diverse scenarios.
> > >
> > > We appreciate the reviewer for highlighting the latest evaluation work. However, we note that MEGA-Bench [R4] was made publicly available on Arxiv on October 14, 2024, which is after the ICLR submission deadline. Nonetheless, we will include a comparison with MEGA-Bench [R4] in the final version of our paper.
> > >
> > > Beside, in section B.2, we demonstrate that 40% of the Dysca data is sufficient to achieve stable evaluation results. However, Dysca aims to provide more than just stable rankings, but also provide in-depth analysis and valuable hard samples through a sufficient number of examples.
> > >
> > > > As a benchmark, it evaluates too few (comparatively) LVLMs, limited to just a small subset of models (actually 16; the 26 LVLMs are somewhat overclaimed). There are many more state-of-the-art multimodal LLMs that should be included at least in the appendix. the reviewer suggests the authors include evaluations of more models next. The reviewer strongly recommends adding more MLLMs for experimental comparison.
> > >
> > > Good Suggestion.  Due to the limited time available during the rebuttal period, we add two more LVLMs, including LLaVA-OneVision [R5] and Phi-3.5 [R6]. The table below displays the evaluation results of the 2 models in clean scenario.
> > >
> > > |      Model      |  MT  |  TF  |  IC  | Avg  |
> > > | :-------------: | :--: | :--: | :--: | :--: |
> > > | LLaVA-OneVision | 94.4 | 92.2 | 51.6 | 79.4 |
> > > |     Phi-3.5     | 92.6 | 83.3 | 53.2 | 76.3 |
> > >
> > > We promise that our evaluation results will be continuously updated alongside the development of LVLMs. Additionally, we will include the reported evaluation results of the two models in the final version of the paper.

---

> > > > ### Author Response · Authors · 2024-11-25
> > > > **Response to Reviewer 3C5q (Part 4/4)**
> > > >
> > > > > Can the authors present the complete prompts used for image generation in the appendix? Could some prompts be displayed?
> > > >
> > > > We very appreciate the valuable idea for adding complete prompts in the appendix. To better illustrate our generation protocol, we present two examples containing all elements of PQA, where “prompt” and “Negative prompt” are used for image generation.
> > > >
> > > > ```
> > > > {
> > > >     "id": 1,
> > > >     "images": [
> > > >       "images/1.png"
> > > >     ],
> > > >     "prompt": "face portrait of a 25 years old female, Indian race, calm and placid, on Mars, retro arcade style, 8-bit, pixelated, vibrant, classic video game, old school gaming, reminiscent of 80s and 90s arcade games",
> > > >     "negative_prompt": "modern, ultra-high resolution, photorealistic, 3D",
> > > >     "instruction": "Question: What is the background in the picture?\nChoices:\n(A) on the lawn\n(B) under the water background\n(C) sunset background\n(D) on Mars\nHint: Please answer the question and provide the correct option letter, e.g., (A), (B), (C), (D), at the end. Do not contain the analysis progress.\nYour answer is:",
> > > >     "question": "What is the background in the picture?",
> > > >     "answer": "on Mars",
> > > >     "question_type": "multi choice",
> > > >     "task": "recognition",
> > > >     "options": [
> > > >       "on the lawn",
> > > >       "under the water background",
> > > >       "sunset background",
> > > >       "on Mars"
> > > >     ],
> > > >     "question_majority": "background",
> > > >     "granularity": "coarse"
> > > > },
> > > > {
> > > > 	"id": 2,
> > > >     "images": [
> > > >       "images/2.png"
> > > >     ],
> > > >     "prompt": "1 white color cat, on Mars, thick layered papercut art of, deep 3D, volumetric, dimensional, depth, thick paper, high stack, heavy texture, tangible layers",
> > > >     "negative_prompt": "2D, flat, thin paper, low stack, smooth texture, painting, drawing, photo, deformed",
> > > >     "instruction": "Question: Is there a cat in the picture?\nChoices:\n(A) False\n(B) True\nHint: Please answer the question and provide the correct option letter, e.g., (A), (B), (C), (D), at the end. Do not contain the analysis progress.\nYour answer is:",
> > > >     "question": "Is there a cat in the picture?",
> > > >     "answer": "True",
> > > >     "question_type": "true or false",
> > > >     "task": "recognition",
> > > >     "options": [
> > > >       "False",
> > > >       "True"
> > > >     ],
> > > >     "question_majority": "animal",
> > > >     "granularity": "fine"
> > > > }
> > > > ```
> > > >
> > > > For more details, please refer to the image generation code released on the anonymous GitHub repository at https://github.com/Benchmark-Dysca/Dysca/blob/main/image_generate/generate_data.py .
> > > >
> > > > > Did the authors assess the quality of automatically synthesized images? How is quality ensured when generating such a large scale of images, especially those containing text?
> > > >
> > > > As mentioned earlier, we use a four-step iterative process to generate high-quality data. In our experiments, we demonstrate that the evaluation results using our synthetic data are reliable. For synthetic images containing text used in OCR evaluation tasks, we acknowledge that current text-to-image methods cannot accurately generate complex images with complex scene text. To minimize errors in text generation images, we first adopt the advanced image synthesis algorithm TextDiffusion2 [R7], focusing on generating shorter text and avoiding complex scenes. Subsequently, we use PP-OCRv3 [R8] to recognize the text in the images and filter out incorrect synthetic images. We randomly sample 1000 text images from Dysca and upload the images at [Anonymous link](https://terabox.com/s/1ny5Kp-PMHNJn8IMXa3OKsg). We invite the reviewer to review them.
> > > >
> > > > > How were the QA pairs (text) constructed on such a large scale? Was ChatGPT utilized? or just all by rule-based template?
> > > >
> > > > The process of generating QA pairs is based on pre-defined rules. We manually design specific rules for generating questions for each tasks. The reason for not using LLMs (e.g., ChatGPT) to assist in annotation is to eliminate the impact of hallucination during QA generation. For more details, please refer to the QA generation code released on the anonymous GitHub repository at https://github.com/Benchmark-Dysca/Dysca/blob/main/pqa_generate/pqa_generate.py .
> > > >
> > > > Please let us know if you have any remaining concerns, or if you would consider updating your evaluation based on our response.
> > > >
> > > > [R1] G. Northcutt et al. Pervasive Label Errors in Test Sets Destabilize Machine Learning Benchmarks. NeurIPS'21.
> > > >
> > > > [R2] Deng et al. ImageNet: A Large-Scale Hierarchical Image Database. CVPR'09.
> > > >
> > > > [R3] Ha et al.  A neural representation of sketch drawings. ICLR'18.
> > > >
> > > > [R4] Chen et al. MEGA-Bench: Scaling Multimodal Evaluation to over 500 Real-World Tasks. Arxiv'24.
> > > >
> > > > [R5] Li et al. Llava-onevision: Easy visual task transfer. Arxiv'24.
> > > >
> > > > [R6] Abdin et al. Phi-3 technical report: A highly capable language model locally on your phone. Arxiv'24.
> > > >
> > > > [R7] Chen et al. TextDiffuser-2: Unleashing the Power of Language Models for Text Rendering. ECCV'24.
> > > >
> > > > [R8] Li et al. PP-OCRv3: More Attempts for the Improvement of Ultra Lightweight OCR System. Arxiv'22.

---

> ### Author Response · Authors · 2024-12-01
> **New Response to Reviewer 3C5q (Part 1/4)**
>
> Dear Reviewer 3C5q:
>
> We sincerely appreciate your thoughtful and detailed feedback, which has greatly enhanced the clarity and quality of our paper.
>
> >  Q1.1: And I still believe that since the process is fully automated, it will inevitably affect the quality of the evaluation testing set.
>
> We acknowledge that using a generated paradigm may introduce some errors. However, we carefully cleaned the metadata and used various tools to ensure the dataset has very few mistakes. Additionally, when we evaluated the models using both manually cleaned and uncleaned VQA data, we found no significant difference in the scores, suggesting that the automatic evaluation results are reliable.
>
> > Q1.2: I remain unconvinced that SDXL’s inclination toward generating artistic-style images does not result in significant style and domain biases. Such images can never replace realistic images in evaluations.
>
> We understand the the reviewers' concerns on the potential bias on the model would react between synthesized and real images.  In the section 4.2.2 of the main paper, we report the data distribution distance between our dataset and the current existing datasets. We select CCBench, COCO_Val, MMVet, MMBench, MME, MMStar, OCRBench and ScienceQA. The reason why we choose these benchmarks is that they have been widely used in evaluating LVLMs. We use Kernel Maximum Mean Discrepancy (KMMD) to measure the distribution distance. Specifically, we randomly sample 3,000 images from each benchmark (if the scale of the benchmark less than 3000, we use all that data) and utilize CLIP to encode these images. Then, we calculate the KMMD value using an RBF kernel between each pair. The results are shown below:
>
> |             | CCBench | COCO | Dysca (Ours) | MM-Vet | MMBench | MME  | MMStar | OCRBench | ScienceQA |
> | ----------- | ------- | ---- | ------------ | ------ | ------- | ---- | ------ | -------- | --------- |
> | CCBench     | 0.00    | 0.28 | 0.14         | 0.11   | 0.24    | 0.31 | 0.13   | 0.35     | 0.17      |
> | COCO        | 0.28    | 0.00 | 0.32         | 0.21   | 0.43    | 0.53 | 0.19   | 0.32     | 0.16      |
> | Dysca       | 0.14    | 0.32 | 0.00         | 0.15   | 0.31    | 0.44 | 0.11   | 0.38     | 0.21      |
> | MM-Vet      | 0.11    | 0.21 | 0.15         | 0.00   | 0.22    | 0.33 | 0.08   | 0.27     | 0.11      |
> | MMBench     | 0.24    | 0.43 | 0.31         | 0.22   | 0.00    | 0.44 | 0.24   | 0.49     | 0.30      |
> | MME         | 0.31    | 0.53 | 0.44         | 0.33   | 0.44    | 0.00 | 0.33   | 0.59     | 0.39      |
> | MMStar      | 0.13    | 0.19 | 0.11         | 0.08   | 0.24    | 0.33 | 0.00   | 0.24     | 0.08      |
> | OCRBench    | 0.35    | 0.32 | 0.38         | 0.27   | 0.49    | 0.59 | 0.24   | 0.00     | 0.24      |
> | ScienceQA   | 0.17    | 0.16 | 0.21         | 0.11   | 0.30    | 0.39 | 0.08   | 0.24     | 0.00      |
> | **Average** | 0.19    | 0.27 | 0.23         | 0.16   | 0.30    | 0.37 | 0.16   | 0.32     | 0.18      |
>
> Each row represents the value of KMMD between two benchmarks. The value of last row denotes the average value of KMMD. As can be seen, the distribution distance between Dysca and real-image benchmarks ranks in the middle compared to all other benchmarks. This demonstrates that the distribution between our benchmark and real benchmarks is similar, indicating that the evaluation results can effectively reflect the model's performance in real-world scenarios.

---

> ### Author Response · Authors · 2024-12-01
> **New Response to Reviewer 3C5q (Part 2/4)**
>
> > Q3: My doubts about their role in testing seem not to have been addressed by the authors. I would prefer the authors to design some quantitative experiments and analyses in this work to help confirm that “automatically synthesized images are also valuable for testing”.
>
> We appreciate the reviewer's suggestion for design more quantitative experiments and analyses to introduce more insightful observations. Thanks to generative evaluation framework of Dysca, we are able to effectively control variables and conduct a detailed analysis of the model's fine-grained capabilities.
>
> Specifically, we first introduce two metrics to measure the sensitivity of LVLMs on question types and covariate shift. The sensitivity to question type aims to evaluate whether LVLMs exhibit inconsistent performance when facing different types of question types (i.e., multi-choice vs. true-or-false). The sensitivity to covariate shift aims to evaluate whether LVLMs exhibit inconsistent performance when facing the same content and question format, but with variations in image covariates (i.e., image style).
>
> ### Metrics:
>
> **1 - Sensitivity to Question Types:**
>
> - True-or-false:
>   $$
>   TFTS=\frac{S-50}{100-50}*100\%.
>   $$
>
> - Multi-choice:
>   $$
>   MCTS = \frac{S-25}{100-25}*100\%.
>   $$
>
> - Overall:
>   $$
>   TS_{Avg} = \frac{TFTS+MCTS}{2},\\
>   TS_{Var} = \frac{(TFTS-TS_{Avg})^2+(MCTS-TS_{Avg})^2}{2}，
>   $$
>
> where $S$ denotes the score of the models in each question type. $TS_{Var}$ denotes the sensitivity of LVLMs to question types. The lower, the better.
>
> **2 - Sensitivity to Covariate Shift:**
> $$
> SS_{Avg}=\frac{\sum_{i=1}^{N}S}{N},\\
> SS_{Var} = \frac{\sum_{i=1}^{N} (S_i-SS_{Avg})^2}{N},
> $$
>
> where $S$ denotes the score of the models in each scenario. Since there are 51 image styles in Dysca, we set $N=51 $. $SS_{Var}$ denotes the sensitivity of LVLMs to covariate shift. The lower, the better.
>
> Table below shows the performance of each LVLM on the two metrics, where 'Score' represents the model's perception score across all samples.
>
> |     Model      |        LLM         |  Visual Encoder   | Score | $TS_{Var}$ | $SS_{Var}$ |
> | :------------: | :----------------: | :---------------: | :---: | :--------: | :--------: |
> |   MiniGPT-4    |     Vicuna-7B      |  EVA-CLIP ViT-G   | 41.4  |   193.6    |    1.7     |
> |   MiniGPT-4    |     Vicuna-13B     |  EVA-CLIP ViT-G   | 50.2  |    758     |    2.5     |
> |   MiniGPT-4    |       LLaMA2       |  EVA-CLIP ViT-G   | 56.6  |   1344.5   |    3.0     |
> |   MiniGPT-2    |       LLaMA2       |  EVA-CLIP ViT-G   | 58.5  |   1512.1   |    4.3     |
> |     BLIP2      |     Flan-T5-XL     |  EVA-CLIP ViT-G   | 65.3  |   165.5    |    8.5     |
> |     BLIP2      |       OPT-3B       |  EVA-CLIP ViT-G   | 39.5  |   110.7    |    2.3     |
> |     BLIP2      |       OPT-7B       |  EVA-CLIP ViT-G   | 39.5  |    50.2    |    1.8     |
> |  InstructBLIP  |     Vicuna-7B      |  EVA-CLIP ViT-G   | 67.5  |   1287.1   |    2.8     |
> |  InstructBLIP  |     Vicuna-13B     |  EVA-CLIP ViT-G   | 64.9  |   1252.4   |    2.5     |
> |  InstructBLIP  |     Flan-T5-XL     |  EVA-CLIP ViT-G   | 66.5  |   271.3    |    6.3     |
> |  InstructBLIP  |    Flan-T5-XXL     |  EVA-CLIP ViT-G   | 68.7  |    215     |    6.8     |
> |   LLava-1.5    |     Vicuna-7B      |    CLIP ViT-L     | 51.3  |   788.1    |    3.2     |
> |   LLava-1.5    |     Vicuna-13B     |    CLIP ViT-L     | 59.2  |   912.5    |    7.7     |
> |     Otter      |      LLaMA-7B      |    CLIP ViT-L     | 54.9  |   427.4    |    6.0     |
> |     Shikra     |      LLaMA-7B      |    CLIP ViT-L     | 62.2  |   440.6    |    5.0     |
> |  Xcomposer-VL  |    InternLM-7B     |  EVA-CLIP ViT-G   | 71.4  |   147.9    |    6.1     |
> | Xcomposer2-VL  |    InternLM2-7B    |    CLIP ViT-L     | 79.1  |    20.9    |    1.9     |
> |  Qwen-VL-Chat  |      Qwen-7B       | OpenClip ViT-bigG | 62.2  |   1885.4   |    6.7     |
> |   Emu2-Chat    |     LLaMA-33B      |    EVA2-CLIP-E    | 63.6  |   2497.9   |    1.6     |
> |     GLM-4V     |   GLM-4-9B-Chat    |    EVA2-CLIP-E    | 82.1  |    25.5    |    0.8     |
> |  MiniCPM-V2.5  | Llama3-Instruct 8B | SigLIP SoViT-400m | 78.8  |    38.3    |    1.7     |
> |     Yi-VL      |     Yi-6B-Chat     |  OpenClip ViT-H   | 75.7  |   233.1    |    1.9     |
> |  mPLUG-Owl-2   |      LLaMA-7B      |    CLIP ViT-L     | 74.1  |   180.9    |    2.2     |
> |  Phi-3-Vision  |     Phi-3 Mini     |    CLIP ViT-L     | 73.2  |   292.7    |    1.5     |
> |     GPT-4o     |         /          |         /         | 75.7  |    67.4    |    1.9     |
> | Gemini-1.5-Pro |         /          |         /         | 77.8  |   439.6    |    1.6     |

---

> ### Author Response · Authors · 2024-12-01
> **New Response to Reviewer 3C5q (Part 3/4)**
>
> Key observations:
>
> 1. **For the Sensitivity to Question Type :** Xcomposer2-VL achieves the best result with a score of 20.9. However, two phenomena are observed: First, the perception ability of LVLM does not show a direct positive correlation with sensitivity to question types. For instance, while the GLM-4V model achieves the highest performance in evaluation tasks, it exhibits higher sensitivity to question types than Xcomposer2-VL. Second, one of the factors influencing the sensitivity to question types may be the inherent biases of the language model. Using the same base language model may result in similar outcomes for this metric.  Additionally, it is noted that the Gemini model does not show a significant advantage in this metric, revealing its preference for certain question types.
> 2. **For the Sensitivity to Covariate Shift:** GLM-4V achieves the best result with a score of 0.8. However, we also observe that the perception ability of LVLM does not show a direct positive correlation with covariate shift. For example, InstructBLIP-Flan-T5-XXL outperforms InstructBLIP-Flan-T5-XL in terms of performance but shows higher sensitivity to covariate shift.
>
> ------
>
> In addition, since Dysca is able to maintain consistency in both image style and question type, we assess the performance of LVLMs in terms of inter-dimension and intra-dimension differences, where "dimension" denotes the 20 sub-tasks we proposed in Dysca.
>
> ### In-depth analysis
>
> - **Analysis on Inter-dimension:**
>
> Here, we perform hierarchical clustering based on the Euclidean distance of the scores across 20 evaluation dimensions for 26 models. Based on the consistency of their performance across different tasks, we categorize the current LVLMs into four levels (first level indicates the best):
>
> | Level |                            Models                            |                         Description                          |
> | :---: | :----------------------------------------------------------: | :----------------------------------------------------------: |
> |   1   | phi-3-vision, GPT-4o, Gemini-1.5-pro,Yi-VL, mPLUG-Owl-2, MiniCPM-v2.5,Xcompose2-VL, GLM-4V | Models that perform consistently well across all dimensions |
> |   2   | Emu2-chat, InstructBlip-Vicuna-7B,  InstructBlip-Vicuna-13B,InstructBlip-Flan-T5-XXL, InstructBlip-Flan-T5-XL, BLIP2-Flan-T5-XL, Xcomposer-VL |   Models that perform well in 60% of the dimensions    |
> |   3   | MiniGPT4-LLaMA-2, MiniGPT2-LLaMA-2, Shikra-LLaMA-7B, Qwen-VL-Chat, LLava-1.5-Vicuna-13B, Otter |   Models that perform well in 30% of the dimensions    |
> |   4   | MiniGPT4-Vicuna-13B, BLIP2-OPT-3B,BLIP-OPT-7B, MiniGPT4-Vicuna-13B, MiniGPT4-Vicuna-7B | Models that perform consistently poorly across all dimensions |
>
> We observe that:
>
> 1. LVLMs tend to perform better in dimensions that involve well-defined image perception, such as landmark recognition and object recognition. However, dimensions likes style recognition and movie recognition exhibit greater challenges for LVLMs, which may be attributed to the limited training resources in these specific domains.
> 2. Commercial models exhibit poor performance in tasks related to "race recognition." This is likely due to the additional safety training incorporated into closed-source models, which causes these models to refuse to answer questions related to race-related issues.
>
> **Analysis on Intra-dimension:**
>
> Thanks to the diverse metadata design, Dysca enables highly granular analysis. Taking animal categories as an example, we analyze the performance of various models within each category. Here, we present the evaluation results of six advanced LVLMs across nine animal categories on multiple-choice questions.
>
> |     Model     |  dog  | rabbit | seagull | salmon | shrimp | shellfish |  ant  |
> | :-----------: | :---: | :----: | :-----: | :----: | :----: | :-------: | :---: |
> |    gemini     | 100.0 | 100.0  |  100.0  |  87.5  | 66.67  |   87.5    | 100.0 |
> |    glm-4v     | 100.0 | 100.0  |  100.0  | 100.0  | 83.33  |   100.0   | 91.67 |
> |    gpt-4o     | 100.0 | 100.0  |  100.0  |  75.0  | 83.33  |   75.0    | 91.67 |
> | xcomposer2-vl | 100.0 | 100.0  |  94.44  | 100.0  |  50.0  |   87.5    | 91.67 |
> | minicpm-v2.5  | 100.0 | 100.0  |  94.44  |  87.5  | 100.0  |   75.0    | 100.0 |
> |     yi-vl     | 100.0 | 100.0  |  94.44  |  62.5  | 83.33  |   87.5    | 83.33 |
>
> The metadata covers 51 animal categories, and we observed that model performance varies significantly across these categories. For animals like seagull and shrimp, models tend to perform worse compared to other animal categories. This is likely because these animals are harder to collect and incorporate into model training, highlighting the need to direct the model's focus to corresponding domains.

---

> > ### Author Response · Authors · 2024-12-01
> > **New Response to Reviewer 3C5q (Part 4/4)**
> >
> > > For Q4 and Q6
> >
> > We appreciate the reviewer’s suggestion for improving the clarity of our paper. We have revised the relevant statements in the revised version of the paper accordingly.
> >
> >
> >
> > In summary,  we believe that the advantage of using synthetic data for model evaluation lies in the following aspects:
> >
> > 1. **Diverse Evaluation Dimensions and In-Depth Analysis**: By synthesizing different types of images and questions, our evaluation scenarios can cover a broader range of tasks and contexts. Additionally, Dysca effectively controls variables, allowing for in-depth analysis of model performance in specific dimensions.
> > 2. **Cost-Effectiveness and Low Data Leakage Risk**: Compared to traditional data collection and annotation methods, synthetic data significantly reduces the costs associated with data generation and labeling, while improving the speed and accuracy of evaluations. Moreover, synthetic data helps mitigate the risk of data leakage, ensuring fairness and reliability in the evaluation process.
> > 3. **Scalability and dynamic Nature**: Dysca can be dynamically updated and expanded to meet evolving model and task requirements, ensuring that the dataset remains aligned with the latest research and technological advancements. This makes our dataset not only scalable but also adaptable for future evaluation needs, providing long-term utility and relevance.

---

### Official Review · Reviewer_DRDs · 2024-10-29

**Soundness:** 3
**Presentation:** 3
**Contribution:** 3
**Rating:** 6
**Confidence:** 5

**Summary:**

This paper proposes Dysca, a dynamic and scalable benchmark that uses large language models (LLMs) and text-to-image diffusion models (T2I) to generate synthetic images, questions, and answers. They filter the low-quality examples by leveraging off-the-shelf models. It is dynamic and scalable. Prior benchmark might have potential data leakage issues and scalability limitations. Evaluation experiments show their findings, such as inconsistency in question types and the vulnerability of attacks. It shows the limitations of current vision language models (VLMs).

**Strengths:**

Their strengths come from the idea of automatically synthesizing dataset curation. The metadata contains information to generate prompt, image, and question-answer pairs.
1. Scaling is easy because of adopting the synthetic data.
2. They propose the findings, the importance of the language model, the inconsistent performance to different question types, the difference perceptual performance on different sub-tasks, and the degradation in attack.
3. They demonstrate the validity of Dysca in the section 4.2.2.

**Weaknesses:**

Their weaknesses come from the idea of simply scaling data. Note that "simple" does not mean "easy." Papers such as [R1], [R2], and [R3] consider the specific points of cultural understanding, Humor, and hallucination. It would be useful to have specific evaluation key points like these benchmarks.
1. How does the proposed benchmark differ from unifying existing benchmarks into one benchmark? Does it evaluate the model behavior that existing benchmarks fail to cover? Combining the existing benchmarks proposed in Table 1 creates a benchmark covering a wide range of perceptual tasks and question types. Therefore, it should clearly show something that existing benchmarks have not been able to do. For example, it would be valuable to show researchers the characteristics of the samples that all VLMs get wrong.
4. What specific insights can the proposed benchmark provide that are not observable through existing benchmarks? It would be interesting to propose a new metric that is only available in the proposed benchmark. This is a weakness that can be linked to Weakness 1. Metadata is an important role in sample generation. Similar to [R4], an explainable metric using that metadata could be an example. Figure 7 may be good for fine-grained analysis, but showing 51 styles or considering other meta data at once is difficult to understand at once.
5. With the growing body of research on training with synthetic data, suppose we follow the training approach: (1) generate data using the same process with Dysca, and (2) train VLMs. Afterward, how should we handle potential data leakage due to the similar data distribution compared to Dysca? It can be useful to know the sub-populations where all VLMs underperform, even when trained on synthetic images.
7. T2I diffusion models have the limitation of creating all possible images corresponding to question-answer pairs. Could authors discuss this limitation? For example, benchmarks that consider specific vulnerabilities of VLMs [R1, R2, R3, R5, R6] seem to be hard to generate automatically.

[R1] Do Androids Laugh at Electric Sheep? Humor “Understanding” Benchmarks from The New Yorker Caption Contest, ACL'23

[R2] Benchmarking Vision Language Models for Cultural Understanding, EMNLP'24

[R3] BEAF: Observing BEfore-AFter Changes to Evaluate Hallucination in Vision-language Models, ECCV'24

[R4] Attribute Based Interpretable Evaluation Metrics for Generative Models, ICML'24

[R5] NaturalBench: Evaluating Vision-Language Models on Natural Adversarial Samples, NeurIPS'24

[R6] Vision language models are blind, arxiv'24

**Questions:**

1. Would authors change "" to the right double quotes?
2. Would authors fix the image captions to be below the image and the table captions to be above the table? i.e., Table 2, Figure 6.
3. In Section 4.2.2, is there any reference that a correlation is high if it is higher than 0.6?

---

> ### Author Response · Authors · 2024-11-25
> **Response to Reviewer DRDs (Part 1/3)**
>
> We sincerely thanks for your positive and constructive feedback. We address your concerns as follows.
>
> > How does the proposed benchmark differ from unifying existing benchmarks into one benchmark? Does it evaluate the model behavior that existing benchmarks fail to cover? Combining the existing benchmarks proposed in Table 1 creates a benchmark covering a wide range of perceptual tasks and question types. Therefore, it should clearly show something that existing benchmarks have not been able to do. For example, it would be valuable to show researchers the characteristics of the samples that all VLMs get wrong.
>
>  Compared to previous datasets, Dysca provide insightful analyses that others don't:
>
> - The vary subtasks provide the fine-grained weakness detection for LVLMs. For example, in the case of Qwen-VL-Chat , it achieves a score of 96.96% accuracy in landmark recognition task, but obtains a score of 53.48% accuracy in age recognition task. The result suggests that Qwen-VL-Chat may require more fine-tuning in age perception data. The performance variability of models across different sub-tasks enable developers to make targeted improvements.
> - Dysca includes diverse questioning types (e.g., Multi-choices, True-or-false, and Image Caption), highlighting the biases models exhibit in response to different types of questions. For example, Otter achieves an accuracy of 47.31% in the multi-choices question type but obtains an accuracy of 82.51% in the true-or-false question type. The problem exposes the bias in the training dataset of LVLMs towards particular question types.
>
> We acknowledge that our benchmark can be seen as the union of perceptual tasks from all existing benchmarks. However, we believe this highlights our advantage: Dysca, as a single benchmark, can complete the evaluation tasks of many different benchmarks do. Although there already exist many benchmarks for LVLMs (as we discussed in Table 1), we provide an end-to-end process from image to Vision-QA generation. Our approach significantly reduces annotation costs compared to manually labeling images (e.g., MME). It also avoids the risk of hallucinate annotations that may occur when using ChatGPT for labeling based on image prompts (e.g., JourneyDB). This novel pipeline enables us to create a benchmark that is easily scalable and adaptable for incorporating new subtasks and scenarios.
>
> We appreciate the valuable suggestion to show more samples that all VLMs fail to answer correctly. In Figure 4 of the main paper, we present the responses of four advanced models to a sample with a print attacking. As shown, when the word "False" is added to the image, all models, including GPT-4o, Gemini-1.5-pro, GLM4-V, and Qwen-VL-Chat, produce incorrect answers. This result underscores the vulnerability of these models in print attacking scenarios. We commit to including additional samples that multiple VLMs fail to answer correctly in the final version of our paper.

---

> > ### Author Response · Authors · 2024-11-25
> > **Response to Reviewer DRDs (Part 2/3)**
> >
> > > What specific insights can the proposed benchmark provide that are not observable through existing benchmarks? It would be interesting to propose a new metric that is only available in the proposed benchmark. This is a weakness that can be linked to Weakness 1. Metadata is an important role in sample generation. Similar to [R4], an explainable metric using that metadata could be an example. Figure 7 may be good for fine-grained analysis, but showing 51 styles or considering other meta data at once is difficult to understand at once.
> >
> > Good suggestion! We very appreciate the valuable idea for using explainable metric. In this work, we construct 20 evaluation dimensions to assess LVLMs. After reviewing the materials provided by the reviewer [R1], we acknowledge that we can further refine our analysis based on metadata. However, unlike [R1], we do not have access to the knowledge about the training datasets of LVLMs, making it challenging to measure the divergence of LVLMs. Nonetheless, inspired by this work, we provide the potential possibility to discuss our evaluation results at a finer granularity. Taking the "animal" dimension as an example, our metadata includes 67 different animal categories. Conducting fine-grained evaluations of various models' perceptual performance across these animal categories will help the community better understand the characteristics of the models.
> >
> > Here, we present the evaluation results of six LVLMs across six animal categories on multiple-choice questions.
> >
> > |           name            |  dog  | rabbit | seagull | salmon | shrimp | shellfish |
> > | :-----------------------: | :---: | :----: | :-----: | :----: | :----: | :-------: |
> > |      gemini-1.5-pro       | 100.0 | 100.0  |  100.0  |  87.5  | 66.67  |   87.5    |
> > |         glm-4v-9b         | 100.0 | 100.0  |  100.0  | 100.0  | 83.33  |   100.0   |
> > |          gpt-4o           | 100.0 | 100.0  |  100.0  |  75.0  | 83.33  |   75.0    |
> > | internlm-xcomposer2-vl-7b | 100.0 | 100.0  |  94.44  | 100.0  |  50.0  |   87.5    |
> > |    minicpm-llama2-v2.5    | 100.0 | 100.0  |  94.44  |  87.5  | 100.0  |   75.0    |
> > |           yi-vl           | 100.0 | 100.0  |  94.44  |  62.5  | 83.33  |   87.5    |
> >
> > We observe that models indeed exhibit inconsistent performance across different animal categories. Overall, as shown in the table above, we find that the models demonstrate weaker performance on seafood-related animals. Specifically, models perform well on commonly seen animals in datasets, such as dogs and rabbits, but their performance significantly degrades on less commonly seen seafood species. For example, XComperVL2 answers correctly only 50% of the questions related to shrimp, compared to 100% accuracy for dogs. The results indicate that Dysca can serve as a fine-grained probe to present the detailed vulnerability of current LVLMs.
> >
> > > With the growing body of research on training with synthetic data, suppose we follow the training approach: (1) generate data using the same process with Dysca, and (2) train VLMs. Afterward, how should we handle potential data leakage due to the similar data distribution compared to Dysca? It can be useful to know the sub-populations where all VLMs underperform, even when trained on synthetic images.
> >
> > Good point. Dysca employs a dynamic evaluation framework that can synthesize novel test data across various dimensions and scenarios based on the needs of evaluators. We believe that even if model developers use the same strategy as Dysca to synthesize training data, as a continuously evolving benchmark, Dysca also possesses new capabilities to detect model shortcomings.

---

> > > ### Author Response · Authors · 2024-11-25
> > > **Response to Reviewer DRDs (Part 3/3)**
> > >
> > > > T2I diffusion models have the limitation of creating all possible images corresponding to question-answer pairs. Could authors discuss this limitation? For example, benchmarks that consider specific vulnerabilities of VLMs [R1, R2, R3, R5, R6] seem to be hard to generate automatically.
> > >
> > > We acknowledge that the T2I diffusion models have the limitation of creating all possible VQAs. In appendix B.1, we discuss the limitation of Dysca in generating data for evaluating cognition abilities (e.g., commonsense reasoning). Nonetheless, it is important to highlight that all existing LVLMs still struggle to provide accurate responses to questions formulated by Dysca. It is evident that for the latest large models, their scores remain below 83. Besides, compared to previous benchmarks, we offer a greater number of evaluation dimensions and a larger scale of evaluation data.
> > >
> > > However, for [R2] provided by reviewer, We believe that Dysca holds the potential to extend towards [R2]. In [R2], the authors leverage stable diffusion model to erase specific objects (i.e., by inpainting) in the original image to evaluate hallucination. Dysca can be extended by transforming metadata into image collections and generating scalable evaluation question-answer pairs through editing techniques.
> > >
> > > We hope that our work serves as a cornerstone for using synthetic images in model evaluation and drives future research into leveraging synthetic images as a testing framework. We believe that with the advancement of deep generative models, Dysca will serve as a dynamic and highly scalable benchmark, gradually expanding its evaluation dimensions.
> > >
> > > > Would authors change "" to the right double quotes?
> > > >
> > > > Would authors fix the image captions to be below the image and the table captions to be above the table? i.e., Table 2, Figure 6.
> > >
> > > We apologize for the mistake of wrong quotes and wrong caption positions. we promise to change it in the final version of our paper.
> > >
> > > > In Section 4.2.2, is there any reference that a correlation is high if it is higher than 0.6?
> > >
> > > The strength of the correlation is shown in the table below.
> > >
> > > | absolute value of correlation | **Strength of correlation** |
> > > | :---------------------------: | :-------------------------: |
> > > |           0.0 < 0.1           |       no correlation        |
> > > |           0.1 < 0.3           |       low correlation       |
> > > |           0.3 < 0.5           |     medium correlation      |
> > > |           0.5 < 0.7           |      high correlation       |
> > > |            0.7 < 1            |    very high correlation    |
> > >
> > > From [Kuckartz et al.: Statistik, Eine verständliche Einführung, 2013, p. 213](https://www.amazon.de/gp/product/3531198890/ref=as_li_qf_asin_il_tl?ie=UTF8&tag=uq3mdsck-21&creative=6742&linkCode=as2&creativeASIN=3531198890&linkId=dbcc42c8569686fa3f927f978c453955). More detailed explanation can be found at https://datatab.net/tutorial/spearman-correlation.
> > >
> > > We apologize for any confusion about this point. We will add the related references in the final version of our paper.
> > >
> > >
> > >
> > > [R1] Kim et al. Attribute Based Interpretable Evaluation Metrics for Generative Models. ICML'24.
> > >
> > > [R2] Bin et al. BEAF: Observing BEfore-AFter Changes to Evaluate Hallucination in Vision-language Models, ECCV'24.
> > >
> > > Please let us know if you have any remaining concerns, or if you would consider updating your evaluation based on our response.

---

> > > > ### Comment · Reviewer_DRDs · 2024-11-27
> > > >
> > > > I thank the authors for their response to the comment. I was wondering if any revisions have been made, as I wasn’t able to find them in the revision history.

---

> > > > > ### Author Response · Authors · 2024-11-27
> > > > > **Paper Updates**
> > > > >
> > > > > Dear Reviewer DRDs,
> > > > >
> > > > > We apologize for the delay in updating the paper.  We have uploaded a revised paper based on the review in ["Paper Updates"](https://openreview.net/forum?id=bU1JOvdXXK&noteId=g6pEgOg2q4) at the top of this page.
> > > > >
> > > > > Best regards!

---

> ### Comment · Reviewer_DRDs · 2024-11-28
>
> It is worth noting that VLMs tend to perform poorly when applied to marine life. This could be caused by the challenges in collecting ocean-related data or possibly to the fact that human understanding of land environments is better than that of the ocean. This observation may indicate a need for more specialized VLMs tailored to oceanic domains.
>
> Regarding reviewer 3C5q's feedback on the Q3 answer, reviewer 3C5q mentioned: _"I would prefer the authors to design some quantitative experiments and analyses in this work to help confirm that ‘automatically synthesized images are also valuable for testing.’”_ I'm in agreement with this response. As such, I have commented on proposing new metrics or presenting results that reveal insights not covered by existing benchmarks. Including these aspects in the main paper would strengthen the work.
>
> Lastly, it is somewhat challenging to identify the changes made in the revised version. It might be helpful to highlight the revisions in a different color.

---

> > ### Author Response · Authors · 2024-12-01
> > **New Response to Reviewer DRDs (Part 1/2)**
> >
> > Dear Reviewer DRDs:
> >
> > We sincerely appreciate your thoughtful and detailed feedback, which has greatly enhanced the clarity and quality of our paper.
> >
> > > I'm in agreement with this response. As such, I have commented on proposing new metrics or presenting results that reveal insights not covered by existing benchmarks. Including these aspects in the main paper would strengthen the work.
> >
> > We appreciate the reviewer's suggestion for design more quantitative experiments and analyses to introduce more insightful observations. Thanks to generative evaluation framework of Dysca, we are able to effectively control variables and conduct a detailed analysis of the model's fine-grained capabilities.
> >
> > Specifically, we first introduce two metrics to measure the sensitivity of LVLMs on question types and covariate shift. The sensitivity to question type aims to evaluate whether LVLMs exhibit inconsistent performance when facing different types of question types (i.e., multi-choice vs. true-or-false). The sensitivity to covariate shift aims to evaluate whether LVLMs exhibit inconsistent performance when facing the same content and question format, but with variations in image covariates (i.e., image style).
> >
> > ### Metrics:
> >
> > **1 - Sensitivity to Question Types:**
> >
> > - True-or-false:
> >   $$
> >   TFTS=\frac{S-50}{100-50}*100\%.
> >   $$
> >
> > - Multi-choice:
> >   $$
> >   MCTS = \frac{S-25}{100-25}*100\%.
> >   $$
> >
> > - Overall:
> >   $$
> >   TS_{Avg} = \frac{TFTS+MCTS}{2},\\
> >   TS_{Var} = \frac{(TFTS-TS_{Avg})^2+(MCTS-TS_{Avg})^2}{2}，
> >   $$
> >
> > where $S$ denotes the score of the models in each question type. $TS_{Var}$ denotes the sensitivity of LVLMs to question types. The lower, the better.
> >
> > **2 - Sensitivity to Covariate Shift:**
> > $$
> > SS_{Avg}=\frac{\sum_{i=1}^{N}S}{N},\\
> > SS_{Var} = \frac{\sum_{i=1}^{N} (S_i-SS_{Avg})^2}{N},
> > $$
> >
> > where $S$ denotes the score of the models in each scenario. Since there are 51 image styles in Dysca, we set $N=51 $. $SS_{Var}$ denotes the sensitivity of LVLMs to covariate shift. The lower, the better.
> >
> > Table below shows the performance of each LVLM on the two metrics, where 'Score' represents the model's perception score across all samples.
> >
> > |     Model      |        LLM         |  Visual Encoder   | Score | $TS_{Var}$ | $SS_{Var}$ |
> > | :------------: | :----------------: | :---------------: | :---: | :--------: | :--------: |
> > |   MiniGPT-4    |     Vicuna-7B      |  EVA-CLIP ViT-G   | 41.4  |   193.6    |    1.7     |
> > |   MiniGPT-4    |     Vicuna-13B     |  EVA-CLIP ViT-G   | 50.2  |    758     |    2.5     |
> > |   MiniGPT-4    |       LLaMA2       |  EVA-CLIP ViT-G   | 56.6  |   1344.5   |    3.0     |
> > |   MiniGPT-2    |       LLaMA2       |  EVA-CLIP ViT-G   | 58.5  |   1512.1   |    4.3     |
> > |     BLIP2      |     Flan-T5-XL     |  EVA-CLIP ViT-G   | 65.3  |   165.5    |    8.5     |
> > |     BLIP2      |       OPT-3B       |  EVA-CLIP ViT-G   | 39.5  |   110.7    |    2.3     |
> > |     BLIP2      |       OPT-7B       |  EVA-CLIP ViT-G   | 39.5  |    50.2    |    1.8     |
> > |  InstructBLIP  |     Vicuna-7B      |  EVA-CLIP ViT-G   | 67.5  |   1287.1   |    2.8     |
> > |  InstructBLIP  |     Vicuna-13B     |  EVA-CLIP ViT-G   | 64.9  |   1252.4   |    2.5     |
> > |  InstructBLIP  |     Flan-T5-XL     |  EVA-CLIP ViT-G   | 66.5  |   271.3    |    6.3     |
> > |  InstructBLIP  |    Flan-T5-XXL     |  EVA-CLIP ViT-G   | 68.7  |    215     |    6.8     |
> > |   LLava-1.5    |     Vicuna-7B      |    CLIP ViT-L     | 51.3  |   788.1    |    3.2     |
> > |   LLava-1.5    |     Vicuna-13B     |    CLIP ViT-L     | 59.2  |   912.5    |    7.7     |
> > |     Otter      |      LLaMA-7B      |    CLIP ViT-L     | 54.9  |   427.4    |    6.0     |
> > |     Shikra     |      LLaMA-7B      |    CLIP ViT-L     | 62.2  |   440.6    |    5.0     |
> > |  Xcomposer-VL  |    InternLM-7B     |  EVA-CLIP ViT-G   | 71.4  |   147.9    |    6.1     |
> > | Xcomposer2-VL  |    InternLM2-7B    |    CLIP ViT-L     | 79.1  |    20.9    |    1.9     |
> > |  Qwen-VL-Chat  |      Qwen-7B       | OpenClip ViT-bigG | 62.2  |   1885.4   |    6.7     |
> > |   Emu2-Chat    |     LLaMA-33B      |    EVA2-CLIP-E    | 63.6  |   2497.9   |    1.6     |
> > |     GLM-4V     |   GLM-4-9B-Chat    |    EVA2-CLIP-E    | 82.1  |    25.5    |    0.8     |
> > |  MiniCPM-V2.5  | Llama3-Instruct 8B | SigLIP SoViT-400m | 78.8  |    38.3    |    1.7     |
> > |     Yi-VL      |     Yi-6B-Chat     |  OpenClip ViT-H   | 75.7  |   233.1    |    1.9     |
> > |  mPLUG-Owl-2   |      LLaMA-7B      |    CLIP ViT-L     | 74.1  |   180.9    |    2.2     |
> > |  Phi-3-Vision  |     Phi-3 Mini     |    CLIP ViT-L     | 73.2  |   292.7    |    1.5     |
> > |     GPT-4o     |         /          |         /         | 75.7  |    67.4    |    1.9     |
> > | Gemini-1.5-Pro |         /          |         /         | 77.8  |   439.6    |    1.6     |

---

> ### Author Response · Authors · 2024-12-01
> **New Response to Reviewer DRDs (Part 2/2)**
>
> Key observations:
>
> 1. **For the Sensitivity to Question Type :** Xcomposer2-VL achieves the best result with a score of 20.9. However, two phenomena are observed: First, the perception ability of LVLM does not show a direct positive correlation with sensitivity to question types. For instance, while the GLM-4V model achieves the highest performance in evaluation tasks, it exhibits higher sensitivity to question types than Xcomposer2-VL. Second, one of the factors influencing the sensitivity to question types may be the inherent biases of the language model. Using the same base language model may result in similar outcomes for this metric.  Additionally, it is noted that the Gemini model does not show a significant advantage in this metric, revealing its preference for certain question types.
> 2. **For the Sensitivity to Covariate Shift:** GLM-4V achieves the best result with a score of 0.8. However, we also observe that the perception ability of LVLM does not show a direct positive correlation with covariate shift. For example, InstructBLIP-Flan-T5-XXL outperforms InstructBLIP-Flan-T5-XL in terms of performance but shows higher sensitivity to covariate shift.
>
> ------
>
> In addition, since Dysca is able to maintain consistency in both image style and question type, we assess the performance of LVLMs in terms of inter-dimension and intra-dimension differences, where "dimension" denotes the 20 sub-tasks we proposed in Dysca.
>
> ### In-depth analysis
>
> - **Analysis on Inter-dimension:**
>
> Here, we perform hierarchical clustering based on the Euclidean distance of the scores across 20 evaluation dimensions for 26 models. Based on the consistency of their performance across different tasks, we categorize the current LVLMs into four levels (first level indicates the best):
>
> | Level |                            Models                            |                         Description                          |
> | :---: | :----------------------------------------------------------: | :----------------------------------------------------------: |
> |   1   | phi-3-vision, GPT-4o, Gemini-1.5-pro, Yi-VL, mPLUG-Owl-2, MiniCPM-v2.5,Xcompose2-VL, GLM-4V | Models that perform consistently well across all dimensions |
> |   2   | Emu2-chat, InstructBlip-Vicuna-7B,  InstructBlip-Vicuna-13B, InstructBlip-Flan-T5-XXL, InstructBlip-Flan-T5-XL, BLIP2-Flan-T5-XL, Xcomposer-VL |   Models that perform well in 60% of the dimensions    |
> |   3   | MiniGPT4-LLaMA-2, MiniGPT2-LLaMA-2, Shikra-LLaMA-7B,Qwen-VL-Chat, LLava-1.5-Vicuna-13B, Otter |   Models that perform well in 30% of the dimensions    |
> |   4   | MiniGPT4-Vicuna-13B, BLIP2-OPT-3B, BLIP-OPT-7B, MiniGPT4-Vicuna-13B, MiniGPT4-Vicuna-7B | Models that perform consistently poorly across all dimensions |
>
> We observe that:
>
> 1. LVLMs tend to perform better in dimensions that involve well-defined image perception, such as landmark recognition and object recognition. However, dimensions likes style recognition and movie recognition exhibit greater challenges for LVLMs, which may be attributed to the limited training resources in these specific domains.
> 2. Commercial models exhibit poor performance in tasks related to "race recognition." This is likely due to the additional safety training incorporated into closed-source models, which causes these models to refuse to answer questions related to race-related issues.
>
> **Analysis on Intra-dimension:**
>
> Thanks to the diverse metadata design, Dysca enables highly granular analysis. Taking animal categories as an example, we analyze the performance of various models within each category. Here, we present the evaluation results of six advanced LVLMs across nine animal categories on multiple-choice questions.
>
> |     Model     |  dog  | rabbit | seagull | salmon | shrimp | shellfish |  ant  |
> | :-----------: | :---: | :----: | :-----: | :----: | :----: | :-------: | :---: |
> |    gemini     | 100.0 | 100.0  |  100.0  |  87.5  | 66.67  |   87.5    | 100.0 |
> |    glm-4v     | 100.0 | 100.0  |  100.0  | 100.0  | 83.33  |   100.0   | 91.67 |
> |    gpt-4o     | 100.0 | 100.0  |  100.0  |  75.0  | 83.33  |   75.0    | 91.67 |
> | xcomposer2-vl | 100.0 | 100.0  |  94.44  | 100.0  |  50.0  |   87.5    | 91.67 |
> | minicpm-v2.5  | 100.0 | 100.0  |  94.44  |  87.5  | 100.0  |   75.0    | 100.0 |
> |     yi-vl     | 100.0 | 100.0  |  94.44  |  62.5  | 83.33  |   87.5    | 83.33 |
>
> The metadata covers 51 animal categories, and we observed that model performance varies significantly across these categories. For animals like seagull and shrimp, models tend to perform worse compared to other animal categories. This is likely because these animals are harder to collect and incorporate into model training, highlighting the need to direct the model's focus to corresponding domains.

---

> > ### Comment · Reviewer_DRDs · 2024-12-03
> >
> > I thank the authors for their thoughtful comments and effort. I have a few follow-up questions and suggestions:
> >
> > 1. In lines 134–136, the text is crossed out. Could you clarify what that means?
> > 2. The Appendix reference is currently displayed as "Appendix ??" in several lines, such as 406 and 434. Could you kindly correct these references?
> > 3. It seems that the authors’ comments mentioned above have not yet been included in either the main paper or the supplementary material. I would appreciate it if these points could be reflected in both.
> >
> > I thank the authors again for their time and effort.

---

> > > ### Author Response · Authors · 2024-12-03
> > > **Response to reviewer DRDs**
> > >
> > > Dear Reviewer DRDs:
> > >
> > > > Q1: In lines 134–136, the text is crossed out. Could you clarify what that means?
> > >
> > > Since the reviewer 3C5q mentioned that *“Instead, I hope the authors can reposition the advantages of their benchmark. Blindly claiming the size of the dataset is meaningless.”*  As suggested, we delete the corresponding claim in lines 134-136 and the change is highlighted with cross out text in the revised pdf. We promise to remove this part in the final version of the paper.
> > >
> > > > Q2: The Appendix reference is currently displayed as "Appendix ??"
> > >
> > > We sincerely apologize for the mistake regarding the lost Appendix number. However, as the deadline for paper revision is November 27th, we are unable to make this change at the moment. We will address this issue in the final version of the paper.
> > >
> > > > Q3: It seems that the authors’ comments mentioned above have not yet been included in either the main paper or the supplementary material. I would appreciate it if these points could be reflected in both.
> > >
> > > We would like to express our sincere gratitude to the reviewer again for the valuable suggestions regarding the use of new metrics and for providing deeper insights. Since the deadline for paper revisions was November 27th, we were unable to update the paper after providing our response on December 3rd. However, we promise that the comments on new metrics and in-depth findings will be fully incorporated into the final version of the paper.

---

> > > > ### Comment · Reviewer_DRDs · 2024-12-03
> > > >
> > > > I've increased the score and really appreciate the comments.

---

### Official Review · Reviewer_ZY88 · 2024-11-03

**Soundness:** 3
**Presentation:** 2
**Contribution:** 2
**Rating:** 6
**Confidence:** 4

**Summary:**

This work propose a large-scale dataset, i.e., Dysca, for evaluating the perceptual capability of LVLMs. To build such a large-scale dataset, the author propose a flexible pipeline which defines the metadata and a rule-based approach for generating the text-to-image prompt, image and question-answer (QA) pairs. In summary, Dysca consists of 617K Vision-language QA pairs, covering 20 perceptual subtasks, 4 image scenarios and 3 question types. The author conduct comprehensive evaluation for 24 existing LVLMs on Dysca, demonstrating the weakness of existing LVLMs.

**Strengths:**

1. The proposed dataset is comprehensive and large in scale. Dysca not only considered different perceptual scenarios, it also provided a testbed for evaluating the perceptual capability of LVLMs under both corruption and printing attack situations.
2. The data construction pipeline is flexible to be extended for assembling the samples of other sub-tasks. Benifit from the powerful generation capability of T2I models, the pipeline also demonstrated the good quality of generative large-scale datasets.
3. The experiments are comprehensive and proved the claimed merits of the proposed dataset.

**Weaknesses:**

There are two main concerns:
1. No data leakage is one of the main claimed contribution for the dataset, but there is no comprehensive statistical comparison of Dysca with existing benchmarks. Only the blind experiment in Appendix D, which should be moved to the main content considering its importance, trying to explain this point which is not enough from my perspective.
2. The setup of adversarial attack, i.e., only PGD, is too simple. Compare to the corruption and printing attack setup, only PGD is considered for adversarial attack is not convincible, especially considering there are bunch of emerging studies on the adversarial attack of LVLMs.


There are also several unclear points,
1. There is no description of the rule-based approach adopted when getting the prompt and QA from the metadata.
2. No explanatio of the metadata itself. Why the metadata is combined with the four aspects? Is there any dataset- or task-specific relationship when considering this?
3. It is unclear that how many examples for each type of corruption and printing attack in the dataset.

**Questions:**

It is very common that the text and image cannot be aligned well for T2I generation. As far as i know, only the data clean part in the pipeline is related to this problem. How do you ensure the threshold and biding (top 6 models) way can work for it?

---

> ### Author Response · Authors · 2024-11-25
> **Response to Reviewer ZY88 (Part 1/2)**
>
> Thank you for your valuable feedback! We really appreciate that you found our work is good-quality and comprehensive.
>
>
> > No data leakage is one of the main claimed contribution for the dataset, but there is no comprehensive statistical comparison of Dysca with existing benchmarks. Only the blind experiment in Appendix D, which should be moved to the main content considering its importance, trying to explain this point which is not enough from my perspective.
>
> Good suggestion! We appreciate the valuable suggestion to move the blind experiment in Appendix D and add more statistical comparison of Dysca with existing benchmarks. In MMStar [R1], the authors report widely-used benchmarks enable LVLMs to answer text-only questions with an accuracy higher than random selection, including MMMU [R2], MMB [R3], ScienceQA [R4], AI2D [R5], Seed [R6] and MathVista [R7]. Here, we present the evaluation results of LLAVA-1.5-7B [R8] on Dysca and the six benchmarks, showing the accuracy of responses when only textual questions are provided. Here, we present the evaluation results of LLAVA-1.5-7B on Dysca and the six datasets, along with a comparison to random selection.
>
> |     Model     | MMMU | MMB  | ScienceQA | AI2D | SEED | MathVista | Dysca |
> | :-----------: | :--: | :--: | :-------: | :--: | :--: | :-------: | :---: |
> | Random Choice | 22.1 |  0   |   24.2    | 23.8 | 24.3 |   17.9    | 37.5  |
> | LLaVA-1.5-7B  | 29.9 | 19.5 |   64.1    | 48.7 | 37.5 |   20.3    | 38.7  |
>
> As can be seen, LLAVA significantly outperforms random selection when only text questions are provided in the other 6 benchmarks. In contrast, Dysca achieves results closest to random selection, indicating that our work has a limited data leakage issue.
>
> > The setup of adversarial attack, i.e., only PGD, is too simple. Compare to the corruption and printing attack setup, only PGD is considered for adversarial attack is not convincible, especially considering there are bunch of emerging studies on the adversarial attack of LVLMs.
>
> We acknowledge that PGD is a simple attack method targeting LVLMs. However, simplicity does not equate to "ineffectiveness." As demonstrated, when using the PGD algorithm for attacks, models experience performance drops exceeding 50% in some cases. For instance, Xcomposer-VL achieves a score of 71.40 on benign samples but drops to a score of 30.28 when evaluated on adversarial samples, a decrease of approximately 58%. We believe that the severe performance degradation caused by even the simplest adversarial attacks is a critical issue that model developers need to address. As a gradually evolving dataset, Dysca will incorporate more diverse attack methods in the future to enrich our evaluation scenarios.
>
> > There is no description of the rule-based approach adopted when getting the prompt and QA from the metadata.
>
> The process of generating the prompt and QA is shown in Figure 3. We first construct four lists containing style, attribute, foreground, and background, respectively. The prompt is combined by randomly selecting a word from each of the four lists. Each word is donated as the answer and then classified into one of 20 perceptual tasks. We design specific rules for generating questions for each tasks. For more details, please refer to the QA generation code released on the anonymous GitHub repository at https://github.com/Benchmark-Dysca/Dysca/blob/main/pqa_generate/pqa_generate.py .
>
> > No explanation of the metadata itself. Why the metadata is combined with the four aspects? Is there any dataset- or task-specific relationship when considering this?
>
> The combination of the four aspects (including their order) is designed based on the experiment. We find that the combination effectively generates prompts that produce accurate images for text-to-image diffusion models. Additionally, these four components cover enough dimensions for the evaluation tasks and serve as a flexible framework to add new evaluation dimensions. In the future, we will also consider adding more aspects if new task dimensions are required.

---

> > ### Author Response · Authors · 2024-11-25
> > **Response to Reviewer ZY88 (Part 2/2)**
> >
> > > It is unclear that how many examples for each type of corruption and printing attack in the dataset.
> >
> > We provide the key statistics in Table 2 of the main paper. The number of corruption questions and printing attack is 156K and 149K, respectively. Since the OCR subtask does not involve print attacking scenario as misidentifying adversarial text does not indicate poor OCR robustness of the LVLMs. Therefore, there are 7K fewer questions in the printing attack scenario.
> >
> > > It is very common that the text and image cannot be aligned well for T2I generation. As far as i know, only the data clean part in the pipeline is related to this problem. How do you ensure the threshold and biding (top 6 models) way can work for it?
> >
> > Good point. We acknowledge that the text and image cannot be perfectly aligned for T2I generation, even with carefully designed prompts and the use of state-of-the-art image synthesis models. To ensure the quality of our data, four steps are adopt as follows:
> >
> > 1. First, we manually remove difficult-to-generate foregrounds and attributes, along with backgrounds and styles that could heavily affect image content. We believe this process can serve as a coarse-grained method to eliminate samples that are highly likely to be generated incorrectly.
> > 2. Then, we use CLIP for further image filtering. In the experiment, we randomly choose 1000 samples first and apply filtering with thresholds of 0.65, 0.7, 0.75, and 0.8 individually. Our findings indicate that using 0.75 as the threshold achieve a good balance between image correctness and data scale.
> > 3. After that, we leverage top 6 LVLMs to eliminate any question-answer pairs where the models either answer incorrectly or indicate that the answer is not included among the options. we observe that nearly 100% of the samples filtered out by these models are incorrect.
> > 4. Finally, we analyze the patterns in these incorrect samples, removing the associated vocabulary from our metadata and discarding all related samples.
> >
> > By meticulously refining the metadata manually and utilizing automated tools to assist in question filtering, Dysca ensures high-quality data synthesis. In the end, we filter out nearly 40% of low quality samples. To further validate the quality of our data cleaning, we also provide a comparison between the evaluation results of manually cleaned data and those of the automated cleaning process in the appendix C. The results show that our data contains very limited incorrect samples compared to mainstream datasets.
> >
> > [R1] Chen et al. Are we on the right way for evaluating large vision-language models? NeurIPS'24.
> >
> > [R2] Yue et al. A massive multi-discipline multimodal understanding and reasoning benchmark for expert agi. CVPR'24.
> >
> > [R3] Liu et al. MMbench:Is your multi-modal model an all-around player? ECCV'24.
> >
> > [R4] Lu et al. Learn to explain: Multimodal reasoning via thought chains for science question answering. NeurIPS'22.
> >
> > [R5] Kembhavi et al. A diagram is worth a dozen images. ECCV'16.
> >
> > [R6] Li et al. Seed-bench: Benchmarking multimodal llms with generative comprehension. CVPR'24.
> >
> > [R7] Lu et al. Mathvista: Evaluating mathematical reasoning of foundation models in visual contexts. NeurIPS'23.
> >
> > [R8] Liu et al. Visual instruction tuning. NeurIPS'23.
> >
> > Please let us know if you have any remaining concerns, or if you would consider updating your evaluation based on our response.

---

> > > ### Comment · Reviewer_ZY88 · 2024-12-03
> > >
> > > Thanks for the authors response, i've  updated my score after reading all the responses to reviewers' comments.

---

### Author Response · Authors · 2024-11-27
**Paper Updates**

Dear Reviewers and Area Chair,

Thanks for your precious and careful review. We have uploaded a revised paper based on your feedback, with updates below:

- Section 3.3: add *'Data Clean'* part to better highlight the efforts we made in ensuring data quality. [Reviewer ZY88 and 3C5q]

- Table 3 & Section 4.1 & Appendix B: move the "blind" results to the main paper and add the comparison results to previous benchmarks. [Reviewer ZY88]
- Table 2 & Figure 6: fix the wrong captions position. [Reviewer DRDs]
- Double quotes problem: fix the wrong format of double quotes in the paper. [Reviewer DRDs]
- Section 4.2.2: add a reference to support the claim that a correlation value exceeding 0.6 indicates a strong correlation. [Reviewer DRDs]
- Appendix G: add hard samples that LVLMs tends to get wrong. [Reviewer DRDs]
- Section Introduction: given a more precise description of "re-annotate" to avoid ambiguity, i.e., *"conduct Vision-language QAs by selecting images from existing dataset and annotate the textual questions."*
- Appendix I: add the JSON structure of data to better assist readers in understanding MPQI, i.e., Metadata, Prompts, Questions and Images. [Reviewer 3C5q]
- Section Introduction: we reclaim our contribution to provide a more appropriate description of our evaluation scope.  [Reviewer 3C5q]
- Section Introduction:  we delete the statement that compares the scale of Dysca with that of other benchmarks.  [Reviewer 3C5q]

We have highlighted the modified part in the revised PDF and invite the reviewers to review them!

Best regards!

Authors of Dysca

---

### Meta-Review · Area_Chair_3YE5 · 2024-12-16

**Metareview:**

This work introduces Dysca, a large-scale dataset designed to evaluate the perceptual capabilities of LVLMs. To construct this extensive dataset, the authors developed a versatile pipeline that defines metadata and employs a rule-based approach to generate text-to-image prompts, images, and question-answer (QA) pairs. The authors also conduct comprehensive evaluation for 24 existing LVLMs on Dysca, demonstrating the weakness of existing LVLMs.

This submission received one positive review and two negative reviews before rebuttal. The main concerns lie on fully automated pipeline and presentation issues. After rebuttal, all the issues are solved. All reviewers have raised their score to boradline accept. In particular, the authors have presented a very detailed response to all reviewers. The main draft has been largely modified according to the reviewers' suggestions and questions.

AC checks the discussions and reviews, finds this work is valuable to current LVLMs benchmarks.

AC recommends accepting this work.

**Additional Comments On Reviewer Discussion:**

No

---

### Decision · Program_Chairs · 2025-01-22

Accept (Poster)